# Revisiting Energy Based Models as Policies: Ranking Noise Contrastive Estimation and Interpolating Energy Models

**Sumeet Singh**                                          *ssumeet@google.com*
*Google DeepMind*

**Stephen Tu**[*]                                          *stephen.tu@usc.edu*
*Ming Hsieh Department of Electrical and Computer Engineering*
*University of Southern California*

**Vikas Sindhwani**                                          *sindhwani@google.com*
*Google DeepMind*

**Reviewed on OpenReview:** *https://openreview.net/forum?id=JmKAYb7IOO*

## Abstract

A crucial design decision for any robot learning pipeline is the choice of policy representation: what type of model should be used to generate the next set of robot actions? Owing to the inherent multi-modal nature of many robotic tasks, combined with the recent successes in generative modeling, researchers have turned to state-of-the-art probabilistic models such as diffusion models for policy representation. In this work, we revisit the choice of energy-based models (EBM) as a policy class. We show that the prevailing folklore—that energy models in high dimensional continuous spaces are impractical to train—is false. We develop a practical training objective and algorithm for energy models which combines several key ingredients: (i) ranking noise contrastive estimation (R-NCE), (ii) learnable negative samplers, and (iii) non-adversarial joint training. We prove that our proposed objective function is asymptotically consistent and quantify its limiting variance. On the other hand, we show that the Implicit Behavior Cloning (IBC) objective is actually biased even at the population level, providing a mathematical explanation for the poor performance of IBC trained energy policies in several independent follow-up works. We further extend our algorithm to learn a continuous stochastic process that bridges noise and data, modeling this process with a family of EBMs indexed by scale variable. In doing so, we demonstrate that the core idea behind recent progress in generative modeling is actually compatible with EBMs. Altogether, our proposed training algorithms enable us to train energy-based models as policies which compete with—and even outperform—diffusion models and other state-of-the-art approaches in several challenging multi-modal benchmarks: obstacle avoidance path planning and contact-rich block pushing.

## 1 Introduction

Many robotic tasks—e.g., grasping, manipulation, and trajectory planning—are inherently *multi-modal*: at any point during task execution, there may be multiple actions which yield task-optimal behavior. Thus, a fundamental question in policy design for robotics is how to capture such multi-modal behaviors, especially when learning from optimal demonstrations. A natural starting point is to treat policy learning as a distribution learning problem: instead of representing a policy as a deterministic map $\pi_\theta(x)$, utilize a conditional

---

[*]Work performed while author was affiliated with Google DeepMind.

generative model of the form $p_\theta(y \mid x)$ to model the entire distribution of actions conditioned on the current robot state.

This approach of modeling the entire action distribution, while very powerful, leaves open a key design question: which type of generative model should one use? Due to the recent advances in diffusion models across a wide variety of domains (Dhariwal & Nichol, 2021; Blattmann et al., 2023; Kong et al., 2021; Saharia et al., 2023; Song et al., 2022; Yang et al., 2022), it is natural to directly apply a diffusion model for the policy representation (Chi et al., 2023; Janner et al., 2022; Reuss et al., 2023). In this work, however, we revisit the use of *energy-based models* (EBMs) as policy representations (Florence et al., 2022). Energy-based models have a number of appealing properties in the context of robotics. First, by modeling a scalar potential field without any normalization constraints, EBMs yield compact representations; model size is an important consideration in robotics due to real-time inference requirements. Second, action selection for many robotics tasks (particularly with strict safety and performance considerations) can naturally be expressed as finding a minimizing solution of a particular loss surface (Adams et al., 2022); this implicit structure is succinctly captured through the sampling procedure of EBMs. Furthermore, reasoning in the density space is more amenable to capturing prior task information in the model (Urain et al., 2022b), and allows for straightforward composition which is not possible with score-based models (Du et al., 2023).

Unfortunately, while the advantages of EBMs are clear, EBMs for continuous, high-dimensional data can be difficult to train due to the intractable computation of the partition function. Indeed, designing efficient algorithms for training EBMs in general is still an active area of research (see e.g., Du & Mordatch (2019); Dai et al. (2019); Song & Kingma (2021); Arbel et al. (2021)). Within the context of imitation learning, recent work on implicit behavior cloning (IBC) (Florence et al., 2022) proposed the use of an InfoNCE (van den Oord et al., 2018) inspired objective, which we refer to as the IBC objective. While IBC is considered state of the art for EBM behavioral cloning, follow up works have found training with the IBC objective to be quite unstable (Ta et al., 2022; Reuss et al., 2023; Chi et al., 2023; Pearce et al., 2023), and hence the practicality of training and using EBMs as policies has remained an open question.

In this work we resolve this open question by designing and analyzing new algorithms based on ranking noise constrastive estimation (R-NCE) (Ma & Collins, 2018), focusing on the behavioral cloning setting. Our main theoretical and algorithmic contributions are summarized as follows:

- **The population level IBC objective is biased:** We show that even in the limit of infinite data, the population level solutions of the IBC objective are in general not correct. This provides a mathematical explanation as to why policies learned using the IBC objective often exhibit poor performance (Ta et al., 2022).

- **Ranking noise contrastive estimation with a learned sampler is consistent:** We utilize the ranking noise contrastive estimation (R-NCE) objective of Ma & Collins (2018) to address the shortcomings of the IBC objective. We further show that jointly learning the negative sampling distribution is compatible with R-NCE, preserving the asymptotic normality properties of R-NCE. This joint training turns out to be quite necessary in practice, as without it the optimization landscape of noise contrastive estimation objectives can be quite ill-conditioned (Liu et al., 2022; Lee et al., 2023).

- **EBMs are compatible with multiple noise resolutions:** A key observation behind recent generative models such as diffusion (Ho et al., 2020; Song et al., 2021b) and stochastic interpolants (Albergo et al., 2023; Lipman et al., 2023; Liu et al., 2023) is that learning the data distribution at multiple noise scales is critical. We show that this concept is actually compatible with EBMs and contrastive training, by introducing a framework that models a *continuum* of EBMs which we term *interpolating EBMs*, trained jointly using R-NCE. We believe this contribution to be of independent interest to the generative modeling community.

- **EBMs trained with R-NCE yield high quality policies:** We show empirically on several multi-modal benchmarks, including an obstacle avoidance path planning benchmark and a contact-rich block pushing task, that EBMs trained with R-NCE are competitive with—*and can even outperform*—IBC and diffusion-based policies. To the best of our knowledge, this is the first result in the

literature which shows that EBMs provide a viable alternative to other state-of-the-art generative models for robot policy learning, and can be stably trained even with modern transformer-based backbones.

## 2 Related work

**Generative Models for Controls and RL.**   Recent advances in generative modeling have inspired many applications in reinforcement learning (RL) (Heess et al., 2013; Haarnoja et al., 2017; 2018; Levine, 2018; Liu et al., 2021; Ho & Ermon, 2016), trajectory planning (Du et al., 2020; Urain et al., 2022b; Ajay et al., 2022; Janner et al., 2022), robotic policy design (Florence et al., 2022; Pearce et al., 2023; Chi et al., 2023; Reuss et al., 2023), and robotic grasp and motion generation (Urain et al., 2022a). In this paper, we focus specifically on generative policies trained with behavior cloning, although many of our technical contributions apply more broadly to learning energy-based models. Most related to our work is the influential implicit behavior cloning (IBC) work of Florence et al. (2022), which advocates for using a noise contrastive estimation objective (which we refer to as the IBC objective) based on InfoNCE (van den Oord et al., 2018) in order to train energy-based policies. As discussed previously, many works (Ta et al., 2022; Reuss et al., 2023; Chi et al., 2023; Pearce et al., 2023) have independently found that the IBC objective is numerically unstable and does not consistently yield high quality policies. One of our main contributions is a theoretical explanation of the limitations of the IBC objective, and an alternative training procedure based on ranking noise contrastive estimation. Another closely related work is Chi et al. (2023), which shows that score-based diffusion models (Sohl-Dickstein et al., 2015; Ho et al., 2020; Song et al., 2020; 2021b) yield state-of-the-art generative policies for many multi-modal robotics tasks. In light of this work and the poor empirical performance of the IBC objective, it is natural to conclude that score-based diffusion methods are now the de facto standard for learning generative policies to solve complex robotics tasks. One of our contributions is to show that this conclusion is false; energy-based policies trained via our proposed R-NCE algorithms are actually competitive with diffusion-based policies in terms of performance.

We focus the rest of the related work discussion on the training of EBMs, as this is where most of our technical contributions apply. Given the considerable scope of this topic and the extensive literature (see e.g., Song & Kingma (2021); Lecun et al. (2006) for thorough literature reviews), we concentrate our discussion on the two most common frameworks: maximum likelihood estimation (MLE) and noise contrastive estimation (NCE).

**Training EBMs via Maximum Likelihood.**   A key challenge in training EBMs via MLE is in computing the gradient of the MLE objective, which involves an expectation of the gradient of the energy function w.r.t. samples drawn from the EBM itself (cf. Section 3). This necessitates the use of expensive Markov Chain Monte Carlo (MCMC) techniques to estimate the gradient, resulting in potentially significant truncation bias that can cause training instability. A common heuristic involves using *persistent chains* (Tieleman, 2008; Du & Mordatch, 2019; Du et al., 2021) over the course of training to better improve the quality of the MCMC samples. However, such techniques cannot be straightforwardly applied to the conditional distribution setting, as one would require storing *context-conditioned* chains, which becomes infeasible when the context space is continuous.

Another common technique for scalable MLE optimization is to approximate the EBM model's samples using an additional learned generative model that easier to sample from. Re-writing the partition function in the MLE objective using a change-of-measure argument and applying Jensen's inequality yields a variational lower bound and thus, a max-min optimization problem where the sampler is optimized to tighten the bound while the EBM is optimized to maximize likelihood (Dai et al., 2019; Grathwohl et al., 2021). For example, Dai et al. (2019) propose to learn a base distribution followed by using a finite number of Hamiltonian Monte Carlo (HMC) leapfrog integration steps to sample from the EBM. Grathwohl et al. (2021) use a Gaussian sampler with a learnable mean function along with variational approximations for the inner optimization. More examples of this max-min optimization approach can be found in e.g., Song & Kingma (2021); Bond-Taylor et al. (2022). Unfortunately, adversarial optimization is notoriously challenging and requires several stabilization tricks to prevent mode-collapse (Kumar et al., 2019).

Alternatively, one may instead leverage an EBM to correct a backbone latent variable-based generative model, commonly referred to as *exponential tilting*. For instance, this has been accomplished by using VAEs (Xiao et al., 2021) and normalizing flows (Arbel et al., 2021; Nijkamp et al., 2022) as the backbone. Leveraging the inverse of this model, one may perform MCMC sampling within the latent space, using the pullback of the exponentially tilted distribution. Ideally, this pullback distribution is more unimodal, thereby allowing faster mixing for MCMC. Pushing forward the samples from the latent space via the backbone model yields the necessary samples for defining the MLE objective. Thus, the EBM acts as an implicit "generative correction" of the backbone model, as opposed to a standalone generative model. The challenge however is that the overall energy function that needs to be differentiated for MCMC sampling now involves the transport map as well, which, for non-trivial backbone models (e.g., continuous normalizing flows (Chen et al., 2018)) yields a non-negligible computational overhead.

We next outline NCE as a form of "discriminative correction", drawing a natural analogy with Generative Adversarial Networks (GANs).

**Training EBMs via Noise Contrastive Estimation.** Instead of directly trying to maximize the likelihood of the observed samples, NCE leverages contrastive learning (Le-Khac et al., 2020) and thus, is composed of two primary parts: (i) the contrastive sample generator, and (ii) the classification-based critic, where the latter serves as the optimization objective. The general formulation was introduced by Gutmann & Hyvärinen (2012) for unconditional generative models, whereby for each data sample, one samples $K$ "contrast" (also referred to as "negative") examples from the noise distribution, and formulates a *binary* classification problem based upon the posterior probability of distinguishing the true sample from the synthesized contrast samples.

For conditional distributions, Ma & Collins (2018) found that the binary classification approach is severely limited: consistency requires the EBM model class to be self-normalized. Instead, they advocate for the ranking NCE (R-NCE) variant (Jozefowicz et al., 2016). R-NCE posits a multi-class classification objective as the critic, and overcomes the consistency issues with the binary classification objective in the conditional setting (Ma & Collins, 2018), at least for discrete probability spaces. In this work, we focus on the R-NCE formulation, and extend the analysis of Ma & Collins (2018) to include jointly optimized noise distribution models over arbitrary probability spaces (i.e., beyond the discrete setting). In particular we study various properties such as: (i) asymptotic convergence, (ii) the pitfalls of weak contrastive distributions, (iii) jointly training the contrastive generative model, either via an independent objective or adversarially, (iv) extension to a multiple noise scale framework by defining a suitable time-indexed family of models and contrastive losses, and (v) sampling from the combined contrastive generative model and EBM.

Despite leveraging a similar classification-based objective to GANs, NCE does not need adversarial training in order to guarantee convergence. In particular, the contrastive model in NCE is not the primary generative model being learned; it is merely used to provide the counterexamples needed to score the EBM-based classifier. Furthermore, our asymptotic analysis shows that a fixed noise distribution suffices for consistency and asymptotic normality. In practice, however, the design of the noise distribution is the most critical factor in the success of NCE methods (Gutmann & Hirayama, 2011). An overly simple fixed noise distribution will lead to a trivial classification problem, which is problematic from an optimization perspective due to exponentially flat loss landscapes (Liu et al., 2022; Lee et al., 2023). Thus, a key part of our algorithmic contributions is in how the negative sampler is learned. We propose simultaneously training a simpler normalizing flow model as the sampler, jointly with the EBM. This is in contrast with adversarial approaches (Gao et al., 2020; Bose et al., 2018) which require max-min training.[1] Without a need for adversarial optimization, our joint training algorithm is numerically stable and yields a pair of *complementary* generative models, whereby the more expressive model (EBM) bootstraps off the simpler model (normalizing flow) for both training and inference-time sampling.

---

[1] Gao et al. (2020) actually consider a similar approach in the binary NCE setting, but ultimately dismiss it in favor of adversarial training. We delve further by analyzing the statistical properties of adversarial training and question its necessity in regards to the R-NCE objective.

## 3 Ranking Noise Contrastive Estimation

### 3.1 Notation

Let the context space $(\mathsf{X}, \mu_X)$ and event space $(\mathsf{Y}, \mu_Y)$ be compact measure spaces (for simplicity, we assume that the event space is not a function of the context). Equip the product space $\mathsf{X} \times \mathsf{Y}$ with a probability measure $\mathsf{P}_{X,Y} = \mathsf{P}_X \times \mathsf{P}_{Y|X}$, and assume that this probability measure is absolutely continuous w.r.t. the base product measure $\mu_X \times \mu_Y$. Suppose furthermore that the conditional distribution $\mathsf{P}_{Y|X}$ is regular. Let $p(x, y)$, $p(x)$, and $p(y \mid x)$ denote the joint density, the marginal density, and the conditional density, respectively (all densities are with respect to the their respective base measures).

Let $\Theta$ be a compact subset of Euclidean space, and consider the function class of conditional energy models:

$$\mathscr{F} := \{\mathcal{E}_\theta(x, y) : \mathsf{X} \times \mathsf{Y} \to \mathbb{R} \mid \theta \in \Theta\}. \tag{3.1}$$

This family of energy functions induces conditional densities in the following way:[2]

$$p_\theta(y \mid x) = \frac{\exp(\mathcal{E}_\theta(x, y))}{Z_\theta(x)}, \quad Z_\theta(x) := \int_\mathsf{Y} \exp(\mathcal{E}_\theta(x, y)) \, d\mu_Y.$$

Next, let $\Xi$ also be a compact subset of Euclidean space, and consider the parametric class of conditional densities for the contrastive model:

$$\mathscr{F}_\mathrm{n} := \{p_\xi(y \mid x) \mid \xi \in \Xi\}. \tag{3.2}$$

While our theory will be written for general parametric density classes, for our proposed algorithm to be practical we require that both computing and sampling from $p_\xi(y \mid x)$ is efficient. Some examples of models which satisfy these requirements include normalizing flows (Grathwohl et al., 2019) and stochastic interpolants (Albergo & Vanden-Eijnden, 2023; Albergo et al., 2023).

We globally fix a positive integer $K \in \mathbb{N}_+$. For a given $x$, let $\mathsf{P}^K_{\mathbf{y}|x;\xi}$ denote the product conditional sampling distribution over $\mathsf{Y}^K$ where $\mathbf{y} \sim \mathsf{P}^K_{\mathbf{y}|x;\xi}$ denotes a random vector $\mathbf{y} = (y_k)_{k=1}^K \in \mathsf{Y}^K$ with $y_k \sim p_\xi(\cdot \mid x)$ for $k = 1, \ldots, K$. Now, for a given $(x, y) \sim \mathsf{P}_{X,Y}$, $\mathbf{y} \sim \mathsf{P}^K_{\mathbf{y}|x;\xi}$, and parameters $\theta \in \Theta$, $\xi \in \Xi$, we define:

$$\ell_{\theta,\xi}(x, y \mid \mathbf{y}) := \log \left[ \frac{\exp(\mathcal{E}_\theta(x, y) - \log p_\xi(y \mid x))}{\sum_{y' \in \{y\} \cup \mathbf{y}} \exp(\mathcal{E}_\theta(x, y') - \log p_\xi(y' \mid x))} \right]. \tag{3.3}$$

With this notation, the Ranking Noise Contrastive Estimation (R-NCE) population *maximization* objective $L(\theta, \xi)$ is:

$$L(\theta, \xi) := \mathbb{E}_{(x,y) \sim \mathsf{P}_{X,Y}} \mathbb{E}_{\mathbf{y} \sim \mathsf{P}^K_{\mathbf{y}|x;\xi}} \ell_{\theta,\xi}(x, y \mid \mathbf{y}). \tag{3.4}$$

For notational brevity, we will write this as:

$$L(\theta, \xi) := \mathbb{E}_{x,y} \mathbb{E}_{\mathbf{y}|x;\xi} \, \ell_{\theta,\xi}(x, y \mid \mathbf{y}),$$

where the length $K$ of the vector $\mathbf{y}$ is implied from context.

**Intuition.** Let us build some intuition for the ranking objective (3.3). For any $(x, y) \sim \mathsf{P}_{X,Y}$ and $\mathbf{y} \sim \mathsf{P}^K_{\mathbf{y}|x;\xi}$, let $\bar{\mathbf{y}} := \{y\} \bigcup \mathbf{y} = (y_j)_{j=1}^{K+1} \in \mathsf{Y}^{K+1}$. Let the random variable $D \in \{1, \ldots, K+1\}$ denote the index of the true sample $y$ within $\bar{\mathbf{y}}$. We will construct a Bayes classifier for $D$ using the EBM. In particular, leveraging the EBM as the generative model for the true distribution $\mathsf{P}_{Y|X}$, we can define the "class-conditional" distribution over $\bar{\mathbf{y}}$ as:

$$\bar{p}_{\theta,\xi}(\bar{\mathbf{y}} \mid x, D = k) = p_\theta(y_k \mid x) \prod_{j=1, j \neq k}^{K+1} p_\xi(y_j \mid x).$$

---

[2] While an EBM would traditionally be represented as $p_\theta(y \mid x) \propto \exp(-\mathcal{E}_\theta(x, y))$ (see e.g., Lecun et al. (2006)), we utilize the non-negated version for notational convenience and consistency with referenced works such as Gutmann & Hyvärinen (2012); Ma & Collins (2018), upon which we base our theoretical development.

Then, the posterior $p(D = k \mid x, \bar{\mathbf{y}}) \triangleq q_{\theta,\xi}(k \mid x, \bar{\mathbf{y}})$ may be derived via Bayes rule as:

$$q_{\theta,\xi}(k \mid x, \bar{\mathbf{y}}) = \frac{\bar{p}_{\theta,\xi}(\bar{\mathbf{y}} \mid x, D = k)p(D = k \mid x)}{\sum_{j=1}^{K+1} \bar{p}_{\theta,\xi}(\bar{\mathbf{y}} \mid x, D = j)p(D = j \mid x)}.$$

Now, it remains to specify a prior distribution $p(D = k \mid x)$. A simple, yet natural, choice reflecting the desire that no index is to be preferred over any other indices is to choose the uniform prior distribution $p(D = k \mid x) = 1/(K + 1)$ for all $k \in \{1, \ldots, K + 1\}$. With this prior, the posterior probability simplifies as:

$$q_{\theta,\xi}(k \mid x, \bar{\mathbf{y}}) = \frac{p_\theta(y_k \mid x) \prod_{j \neq k} p_\xi(y_j \mid x)}{\sum_{j=1}^{K+1} p_\theta(y_j \mid x) \prod_{l \neq j} p_\xi(y_l \mid x)} \tag{3.5}$$

$$= \frac{p_\theta(y_k \mid x)/p_\xi(y_k \mid x)}{\sum_{j=1}^{K+1} p_\theta(y_j \mid x)/p_\xi(y_j \mid x)}. \tag{3.6}$$

Substituting $p_\theta(y \mid x) = \exp(\mathcal{E}_\theta(x, y))/Z_\theta(x)$, and noting that the partition function $Z_\theta$ cancels:

$$q_{\theta,\xi}(k \mid x, \bar{\mathbf{y}}) = \frac{\exp(\mathcal{E}_\theta(x, y_k) - \log p_\xi(y_k \mid x))}{\sum_{y \in \bar{\mathbf{y}}} \exp(\mathcal{E}_\theta(x, y) - \log p_\xi(y \mid x))}. \tag{3.7}$$

Thus, the objective in (3.3) denotes the posterior log-probability (conditioned on both the data $(x, y)$ and the contrastive samples $\mathbf{y}$) that the sample $y$ is drawn from the energy model $\mathcal{E}_\theta$ and the remaining samples $K$ samples in $\mathbf{y}$ are all drawn iid from $p_\xi(\cdot \mid x)$. Based on this interpretation, we adopt the terminology that $y$ is the *positive example*, $\mathbf{y}$ contains the *negative examples*, and $p_\xi(\cdot \mid x)$ is the *negative proposal distribution*.

Given this posterior log-probability interpretation, it will prove useful to re-state the R-NCE objective as follows. For $k \in \{1, \ldots, K + 1\}$, let $\bar{\mathsf{P}}_{\bar{\mathbf{y}}|x;\xi,k}$ denote the true "class-conditional" distribution over $\mathsf{Y}^{K+1}$. That is, for a given $\bar{\mathbf{y}} = (y_j)_{j=1}^{K+1} \sim \bar{\mathsf{P}}_{\bar{\mathbf{y}}|x;\xi,k}$, we have $y_k \sim p(\cdot \mid x)$ and $y_j \sim p_\xi(\cdot \mid x)$ for $j \neq k$. When the index $k$ is omitted, this refers to the setting of $k = 1$: $\bar{\mathsf{P}}_{\bar{\mathbf{y}}|x;\xi} \equiv \bar{\mathsf{P}}_{\bar{\mathbf{y}}|x;\xi,1}$. By symmetry then, for any $k \in \{1, \ldots, K + 1\}$, the population objective in (3.4) can be equivalently written as:

$$L(\theta, \xi) := \mathbb{E}_{x \sim \mathsf{P}_X} \mathbb{E}_{\bar{\mathbf{y}} \sim \bar{\mathsf{P}}_{\bar{\mathbf{y}}|x;\xi,k}} \log q_{\theta,\xi}(k \mid x, \bar{\mathbf{y}}). \tag{3.8}$$

For notational brevity, we will write this as:

$$L(\theta, \xi) := \mathbb{E}_x \mathbb{E}_{\bar{\mathbf{y}}|x;\xi,k} \ \log q_{\theta,\xi}(k \mid x, \bar{\mathbf{y}}).$$

This notation will prove useful for subsequent analysis.

**Comparison to Maximum Likelihood Estimation.** Let us take a moment to compare the R-NCE objective to the standard maximum likelihood estimation (MLE) objective:

$$L_{\mathrm{mle}}(\theta) := \mathbb{E}_{x,y} \log p_\theta(y \mid x). \tag{3.9}$$

Letting $\bar{\mathbf{y}} = \{y\} \cup \mathbf{y}$, and taking the gradient of both $L_{\mathrm{mle}}(\theta)$ and $L(\theta, \xi)$ (assume for now the validity of exchanging the order of expectation and derivative),

$$\nabla_\theta L_{\mathrm{mle}}(\theta) = \mathbb{E}_{x,y} \nabla_\theta \mathcal{E}_\theta(x, y) - \mathbb{E}_x \mathbb{E}_{y' \sim p_\theta(\cdot|x)} \nabla_\theta \mathcal{E}_\theta(x, y'), \qquad \textbf{(MLE)}$$

$$\nabla_\theta L(\theta, \xi) = \mathbb{E}_{x,y} \nabla_\theta \mathcal{E}_\theta(x, y) - \mathbb{E}_x \mathbb{E}_{\bar{\mathbf{y}}|x;\xi} \left[ \sum_{i=1}^{K+1} q_{\theta,\xi}(i \mid x, \bar{\mathbf{y}}) \nabla_\theta \mathcal{E}_\theta(x, y_i) \right]. \qquad \textbf{(R-NCE)}$$

Comparing the two expressions, the main difference is in the gradient correction term on the right. For MLE, this gradient computation requires sampling from the energy model $p_\theta(y \mid x)$, which is prohibitively expensive during training. On the other hand, the R-NCE gradient is computationally efficient to compute, provided the negative sampler is efficient to both sample from and compute log-probabilities.

---

**Algorithm 1:** Ranking noise-contrastive estimation with learnable negative sampler

---

**Data:** Dataset $\mathcal{D} = \{(x_i, y_i)\}_{i=1}^n$, number of outer steps $T_{\text{outer}}$, number of sampler steps per outer step $T_{\text{samp}}$, number of R-NCE steps per outer step $T_{\text{rnce}}$, number of negative samples $K$.

**Result:** Energy model parameters $\theta$, negative sampler parameters $\xi$.

**1** Initialize $\theta$ and $\xi$.

**2** Set $\hat{L}_{\text{mle}}(\xi; \mathcal{B}) = -\sum_{(x_i, y_i) \in \mathcal{B}} \log p_\xi(y_i \mid x_i)$.

**3** Set $\hat{L}_{\text{rnce}}(\theta, \xi; \bar{\mathcal{B}}) = -\sum_{(x_i, y_i, \mathbf{y}_i) \in \bar{\mathcal{B}}} \ell_{\theta, \xi}(x_i, y_i \mid \mathbf{y}_i)$.

**4 for** $T_{\text{outer}}$ *iterations* **do**

**5**     **for** $T_{\text{samp}}$ *iterations* **do**

**6**         $\mathcal{B} \leftarrow$ batch of $\mathcal{D}$.

**7**         $\xi \leftarrow$ minimize using $\nabla_\xi \hat{L}_{\text{mle}}(\xi; \mathcal{B})$. `// Or use auxiliary loss, see Remark` 3.1.

**8**     **end**

**9**     **for** $T_{\text{rnce}}$ *iterations* **do**

**10**         $\mathcal{B} \leftarrow$ batch of $\mathcal{D}$.

            `// Augment batch with negative samples.`

**11**         $\bar{\mathcal{B}} \leftarrow \{(x_i, y_i, \mathbf{y}_i) \mid (x_i, y_i) \in \mathcal{B}\}$, with $\mathbf{y}_i \sim \mathsf{P}_{\mathbf{y}|x_i; \xi}^K$.

**12**         $\theta \leftarrow$ minimize using $\nabla_\theta \hat{L}_{\text{rnce}}(\theta, \xi; \bar{\mathcal{B}})$.

**13**     **end**

**14**     **return** $(\theta, \xi)$

**15 end**

---

**Algorithm 2:** Sample from $p_\theta(y \mid x)$ for Euclidean $\mathsf{Y}$

---

**Data:** Context $x \in \mathsf{X}$, energy model parameters $\theta$, negative sampler parameters $\xi$, Langevin steps $T_{\text{mcmc}}$, Langevin step size $\eta$.

**Result:** Sample $y \sim p_\theta(y \mid x)$.

**1** Set $y_0 \sim p_\xi(y \mid x)$.

   `// Run Langevin sampling, could alternatively run HMC sampling.`

**2 for** $t = 0, \ldots, T_{\text{mcmc}} - 1$ **do**

**3**    Set $y_{t+1} = y_t + \eta \nabla_y \mathcal{E}_\theta(x, y_t) + \sqrt{2\eta} w_t$, with $w_t \sim \mathcal{N}(0, I)$.

**4 end**

**5 return** $y_T$

---

## 3.2 Algorithm

Our main proposed R-NCE algorithm with a learnable negative sampler is presented in Algorithm 1. We remark that one could alternatively pre-train the negative sampler distribution $p_\xi$ (i.e., move Lines 5–8 after Line 3 and set $T_{\text{samp}}$ large enough so that the negative sampler converges), as opposed to jointly training the negative sampler concurrently with the energy model. Using a pre-trained negative sampler distribution was previously explored in Gao et al. (2020), where it was empirically shown to be an ineffective idea for binary NCE. We will also revisit this choice within our experiments.

**Remark 3.1.** *As mentioned earlier, any family of densities $\mathscr{F}_n$ suffices in Algorithm 1. For specific families such as stochastic interpolants (Albergo & Vanden-Eijnden, 2023) where an auxiliary loss is used instead of negative log-likelihood, Algorithm 1, Line 7 is replaced with the corresponding auxiliary loss.*

**Remark 3.2.** *Note that in Algorithm 1, the proposal model is optimized independently of the EBM; the latter is optimized purely via the R-NCE objective. Crucially, the negative samples $\mathbf{y}$ do not depend upon the EBM's parameters $\theta$. Hence, no re-parameterization tricks are required since the batch augmentation in Line 11 does not introduce any gradient dependencies on $\theta$ for the optimization step in Line 12.*

Algorithm 2 outlines how to sample from the combined proposal $p_\xi$ and EBM $p_\theta$ models. In particular, we leverage $p_\xi$ to generate an initial sample in Line 1 and warm-start a finite-step MCMC chain with the

EBM in Lines 2-4. We outline the simplest possible MCMC implementation (Langevin sampling), however one may leverage any additional variations and tricks, including Hamiltonian Monte Carlo (HMC) and Metropolis-Hastings adjustments.

## 4 Analysis

We now present our analysis of ranking noise contrastive estimation. First, we will require the following regularity assumptions:

**Assumption 4.1** (Continuity). *Assume the following assumptions are satisfied:*

1. *For a.e. $(x, y)$, the map $\theta \mapsto \mathcal{E}_\theta(x, y)$ is continuous, and for all $\theta$, the map $(x, y) \mapsto \mathcal{E}_\theta(x, y)$ is measurable.*

2. *For a.e. $(x, y)$, the map $\xi \mapsto p_\xi(y \mid x)$ is continuous, and for all $\xi$, the map $(x, y) \mapsto p_\xi(y \mid x)$ is measurable.*

We additionally require the following boundedness assumption:

**Assumption 4.2** (Uniform Integrability). *Assume that:*

$$\exists \theta_0, \xi_0 \in \Theta \times \Xi \quad \text{s.t.} \quad \mathbb{E}_{x,y} \mathbb{E}_{\mathbf{y} \mid x; \xi} \sup_{\substack{\theta \in \Theta, \\ \xi \in \Xi}} |\ell_{\theta, \xi}(x, y \mid \mathbf{y}) - \ell_{\theta_0, \xi_0}(x, y \mid \mathbf{y})| < \infty.$$

Furthermore, define the random variable $\bar{\ell}_{\theta, \xi}(x, y)$, measurable on $\mathsf{X} \times \mathsf{Y}$:

$$\bar{\ell}_{\theta, \xi}(x, y) := \mathbb{E}_{\mathbf{y} \mid x; \xi} \, \ell_{\theta, \xi}(x, y \mid \mathbf{y}).$$

Then, by the continuity of $(\theta, \xi) \mapsto \ell_{\theta, \xi}(x, y \mid \mathbf{y})$ for a.e. $(x, y, \mathbf{y})$ and the Lebesgue Dominated Convergence theorem, it follows that: (i) for a.e. $(x, y)$, the map $(\theta, \xi) \mapsto \bar{\ell}_{\theta, \xi}(x, y)$ is continuous, and (ii) the map $(\theta, \xi) \mapsto L(\theta, \xi)$ is continuous. For all results in this section, the proofs are provided in Appendix C.

### 4.1 Optimality

The first step in analyzing the R-NCE objective is to characterize the set of optimal solutions. To do so, we introduce the following key definition:

**Definition 4.3** (Realizability). *We say that $\mathscr{F}$ is* realizable *if there exists a $\theta_\star \in \Theta$ such that for almost every $(x, y)$:*

$$p_{\theta_\star}(y \mid x) = \frac{\exp(\mathcal{E}_{\theta_\star}(x, y))}{Z_{\theta_\star}(x)} = p(y \mid x). \tag{4.1}$$

*We let $\Theta_\star$ denote the set of $\theta_\star \in \Theta$ such that the above condition (4.1) holds. If $\Theta_\star$ is a singleton, we say that $\mathscr{F}$ is* identifiable.

Realizability stipulates that the energy model $\mathcal{E}_\theta$ is sufficiently expressive to capture the true underlying distribution. We now establish the following optimality result:

**Theorem 4.4** (Optimality). *Suppose that $\mathscr{F}$ is realizable. Then, for any $\xi \in \Xi$, we have that:*

$$\Theta_\star = \operatorname*{argmax}_{\theta \in \Theta} L(\theta, \xi). \tag{4.2}$$

We note that the proof of Theorem 4.4 mostly follows that of Ma & Collins (2018, Theorem 4.1), but includes the measure-theoretic arguments necessary to extend it to our more general setting.

**Remark 4.5.** *Notice that the optimality of $\Theta_\star$ is independent of the negative proposal distribution, thereby allowing us to use any generative model for negative samples $\mathbf{y}$, provided it is easily samplable and the computation of $\log p_\xi(\cdot \mid x)$ is tractable. The question of which negative proposal distribution to use is addressed in Section 4.3.*

### 4.1.1  Comparison to the IBC Objective

In Florence et al. (2022), the following implicit behavior cloning (IBC) objective is proposed:

$$L_{\mathrm{ibc}}(\theta, \xi) = \mathbb{E}_{x,y}\mathbb{E}_{\mathbf{y}|x;\xi} \log \left[ \frac{\exp(\mathcal{E}_\theta(x,y))}{\sum_{y' \in \{y\} \cup \mathbf{y}} \exp(\mathcal{E}_\theta(x,y'))} \right]. \tag{4.3}$$

The IBC objective can be seen as a special case of the R-NCE objective, with the particular choice of negative sampling distribution $p_\xi(y \mid x)$ as the uniform distribution on $\mathsf{Y}$ for a.e. $x \in \mathsf{X}$. Note that the uniform distribution is *the only choice* which renders the objective unbiased (this observation was also made in Ta et al. (2022)). Figure 1 illustrates a toy one dimensional setting with a Gaussian proposal distribution, illustrating the bias inherent in the population objective when a non-uniform proposal distribution is used. Shortly, we will characterize the optimizers of the IBC objective, quantifying the bias.

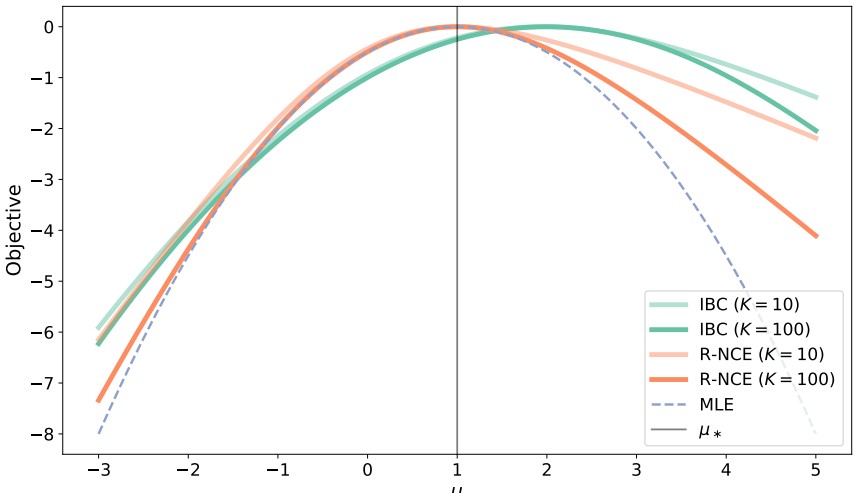

**Figure 1:** The population objective landscape for the IBC objective (4.3) versus the R-NCE objective (3.4), for $K \in \{10, 100\}$. The true distribution $p(y \mid x)$ is a $\mathcal{N}(1,1)$ distribution (for simplicity, the context $x$ is not relevant), and the proposal distribution is $\mathcal{N}(0,1)$. The energy model function class is comprised of unit variance Gaussians, i.e., $\mathscr{F} = \{(x,y) \mapsto -\frac{1}{2}(y-\mu)^2 \mid \mu \in \mathbb{R}\}$. Note that for ease of comparison, all objectives have been shifted so that the maximum values are zero. Furthermore, the maximum likelihood objective (3.9) has also been plotted for reference.

The fact that the IBC objective forces the use of a uniform proposal distribution substantially limits its applicability in high dimension. Indeed, in Section 4.3 we will see that a proposal distribution which is non-informative causes the gradient to vanish, and hence training to stall. This issue has been empirically observed in many follow up works (Ta et al., 2022; Reuss et al., 2023; Chi et al., 2023; Pearce et al., 2023).

**Optimizers of the IBC Objective.**  The optimizers of the IBC objective can be studied using the optimality of the R-NCE objective (Theorem 4.4) in conjunction with a change of variables. We first introduce the notion of $\xi$-realizability, which posits that the density *ratio $p(y \mid x)/p_\xi(y \mid x)$* is representable by the function class $\mathscr{F}$.

**Definition 4.6** ($\xi$-realizability). *Let $\xi \in \Xi$. We say that $\mathscr{F}$ is $\xi$-realizable if there exists $\theta_\star \in \Theta$ such that for a.e. $(x,y)$:*

$$\frac{\exp(\mathcal{E}_{\theta_\star}(x,y))}{\int_{\mathsf{Y}} \exp(\mathcal{E}_{\theta_\star}(x,y))p_\xi(y \mid x)\,\mathrm{d}\mu_Y} = \frac{p(y \mid x)}{p_\xi(y \mid x)}.$$

*We let $\Theta_\xi$ denote the subset of $\Theta$ such that the above condition holds.*

With this definition, we can now characterize the optimizers of the IBC objective.

**Proposition 4.7.** *Let $\xi \in \Xi$, and suppose that $\mathscr{F}$ is $\xi$-realizable (Definition 4.6). We have:*

$$\Theta_\xi = \underset{\theta \in \Theta}{\operatorname{argmax}}\, L_{\mathrm{ibc}}(\theta, \xi).$$

*Proof.* The proof follows immediately from Theorem 4.4 after a simple transformation. In particular, we note that:

$$L_{\mathrm{ibc}}(\theta, \xi) = \mathbb{E}_{x,y}\mathbb{E}_{\mathbf{y}|x;\xi} \log \left[ \frac{\exp(\mathcal{E}_\theta(x, y) + \log p_\xi(y \mid x) - \log p_\xi(y \mid x))}{\sum_{y' \in \{y\} \cup \mathbf{y}} \exp(\mathcal{E}_\theta(x, y') + \log p_\xi(y' \mid x) - \log p_\xi(y' \mid x))} \right].$$

This is equivalent to the R-NCE objective over the shifted function class:

$$\mathscr{F}_\xi := \{\mathcal{E}_\theta(x, y) + \log p_\xi(y \mid x) \mid \theta \in \Theta\}.$$

Note that realizability of $\mathscr{F}_\xi$ (Definition 4.3) is equivalent to $\xi$-realizability of $\mathscr{F}$ (Definition 4.6), from which the claim now follows by Theorem 4.4. □

Proposition 4.7 illustrates that optimizing the IBC objective results in an energy model which represents the density ratio $p(y \mid x)/p_\xi(y \mid x)$, explaining the bias in the IBC objective when the negative sampling distribution is non-uniform. The reason for this is because the IBC objective is based on an incorrect application of InfoNCE (van den Oord et al., 2018), which is designed to maximize a lower bound on mutual information between contexts $x$ and events $y$, rather than extract an energy model $\mathcal{E}_\theta(x, y)$ to model $p(y \mid x)$ (Ta et al., 2022).

### 4.2 Asymptotic Convergence

In this section, we study the asymptotics of the R-NCE joint optimization algorithm. We begin by defining the finite-sample version of the R-NCE population objective:

$$L_n(\theta, \xi) := \hat{\mathbb{E}}_{x,y}\, \bar{\ell}_{\theta,\xi}(x, y) := \frac{1}{n} \sum_{i=1}^{n} \bar{\ell}_{\theta,\xi}(x_i, y_i). \tag{4.4}$$

We consider a sequence of arbitrary negative sampler parameters $\{\hat{\xi}_n\} \subset \Xi$ which are random variables depending on the corresponding prefix of data points $\{(x_i, y_i)\}$. From these negative sampler parameters, we select energy model parameters via the optimization:

$$\hat{\theta}_n \in \underset{\theta \in \Theta}{\operatorname{argmax}}\, L_n(\theta, \hat{\xi}_n). \tag{4.5}$$

We note that the definition of $\hat{\theta}_n$ in (4.5) is a stylized version of Algorithm 1, where the training procedure (e.g., gradient-based optimization) is assumed to succeed. We now establish consistency of the sequence of estimators $\{\hat{\theta}_n\}$ to the set $\Theta_\star$. Let $d(z, \mathcal{A})$ denote the set distance function between the set $\mathcal{A}$ and the point $z$, i.e.,[3]

$$d(z, \mathcal{A}) := \inf_{z' \in \mathcal{A}} \|z - z'\|.$$

**Theorem 4.8** (General Consistency). *Suppose that $\mathscr{F}$ is realizable, $\Theta_\star$ is contained in the interior of $\Theta$, and Assumptions 4.1 and 4.2 hold. Let $\hat{\theta}_n \in \operatorname{argmax}_{\theta \in \Theta} L_n(\theta, \hat{\xi}_n)$ denote an arbitrary empirical risk maximizer from $n$ samples. Then, $d(\hat{\theta}_n, \Theta_\star) \xrightarrow{n \to \infty} 0$ almost surely.*

---

[3]Throughout this work, $\|\cdot\|$ denotes the $\ell_2$-norm.

The proof of Theorem 4.8 requires some care to deal with the fact that even for a fixed $\theta$, the empirical objective $L_n(\theta, \hat{\xi}_n)$ is *not* an unbiased estimate of the population objective $L(\theta, \hat{\xi}_n)$, since the negative sampler parameters $\hat{\xi}_n$ in general depend on the data points $\{(x_i, y_i)\}_{i=1}^n$. The key, however, lies in Theorem 4.4, which states that *for any $\xi$, the map $\theta \mapsto L(\theta, \xi)$ has the same set of maximizers $\Theta_\star$*. With this observation, we show that it suffices to establish uniform convergence jointly over $\Theta \times \Xi$, i.e., $\sup_{\theta \in \Theta, \xi \in \Xi} |L_n(\theta, \xi) - L(\theta, \xi)| \to 0$ a.s. as $n \to \infty$.

As with the optimality result in Theorem 4.4, the asymptotic consistency of the R-NCE estimator is independent of the sequence of noise distributions, parameterized by $\{\hat{\xi}_n\}$. In Section 4.3, we study the algorithmic implications of the design of the noise distribution and its concurrent optimization. We now characterize the asymptotic normality of $\hat{\theta}_n$, under certain regularity assumptions:

**Assumption 4.9** ($\mathcal{C}^2$-Continuity). *Assume the following conditions are satisfied:*

1. *For a.e. $(x, y)$, the maps $\theta \mapsto \mathcal{E}_\theta(x, y)$ and $\xi \mapsto p_\xi(y \mid x)$ are $\mathcal{C}^2$-continuous.*

2. *For a.e. $(x, y)$ and all $(\theta, \xi) \in \Theta \times \Xi$:[4]*

$$\exists \varepsilon > 0 \quad \text{s.t.} \quad \mathbb{E}_{\mathbf{y}|x;\xi} \sup_{\|\theta' - \theta\| \leqslant \varepsilon, \|\xi' - \xi\| \leqslant \varepsilon} \|\nabla^2_{(\theta, \xi)} \ell_{\theta', \xi'}(x, y \mid \mathbf{y})\|_{\text{op}} < \infty. \tag{4.6}$$

*Then, by the Lebesgue Dominated Convergence theorem, the map $(\theta, \xi) \mapsto \bar{\ell}_{\theta, \xi}(x, y)$ is $\mathcal{C}^2$-continuous for a.e. $(x, y)$. Additionally, assume that for all $(\theta, \xi) \in \Theta \times \Xi$:*

$$\exists \varepsilon > 0 \quad \text{s.t.} \quad \mathbb{E}_{x, y} \sup_{\|\theta' - \theta\| \leqslant \varepsilon, \|\xi' - \xi\| \leqslant \varepsilon} \|\nabla^2_{(\theta, \xi)} \bar{\ell}_{\theta', \xi'}(x, y)\|_{\text{op}} < \infty. \tag{4.7}$$

*It follows then that the map $(\theta, \xi) \mapsto L(\theta, \xi)$ is $\mathcal{C}^2$-continuous.*

We are now ready to formalize the asymptotic distribution of $\hat{\theta}_n$.

**Theorem 4.10** (Asymptotic Normality). *Suppose that $\mathscr{F}$ is realizable and Assumptions 4.1, 4.2, and 4.9 hold. Let $\{\hat{\theta}_n\}$ be a sequence of optimizers as defined via (4.5). Denote the joint parameter $\gamma := (\theta, \xi)$ and joint r.v. $z := (x, y)$, and assume the following properties hold:*

1. *The negative proposal distribution parameters $\{\xi_n\}$ are $\sqrt{n}$-consistent about a fixed point $\xi_\star$, i.e., $\sqrt{n}(\hat{\xi}_n - \xi_\star) = O_p(1)$.*

2. *The set $\Theta_\star = \{\theta_\star\}$ is a singleton, and $\gamma_\star := (\theta_\star, \xi_\star)$ lies in the interior of $\Theta \times \Xi$.*

3. *The Hessian of $\bar{\ell}_\gamma$ is Lipschitz-continuous, i.e.,*

$$\|\nabla^2_\gamma \bar{\ell}_{\gamma_1}(z) - \nabla^2_\gamma \bar{\ell}_{\gamma_2}(z)\|_{\text{op}} \leqslant M(z)\|\gamma_1 - \gamma_2\|, \quad \mathbb{E}_z[M(z)] < \infty, \ \forall \gamma_1, \gamma_2 \in \Theta \times \Xi.$$

4. *The $(\theta, \theta)$ block of the Hessian $\nabla^2_\gamma L(\theta_\star, \xi_\star)$, denoted $\nabla^2_\theta L(\theta_\star, \xi_\star)$, satisfies $\nabla^2_\theta L(\theta_\star, \xi_\star) \prec 0$.*

*Then, $\sqrt{n}(\hat{\theta}_n - \theta_\star) \rightsquigarrow \mathcal{N}(0, V_\theta)$, where*

$$V_\theta = -\nabla^2_\theta L(\theta_\star, \xi_\star)^{-1} = -\mathbb{E}_z \left[\nabla^2_\theta \bar{\ell}_{\gamma_\star}(z)\right]^{-1}.$$

A few remarks are in order regarding this result. First, note that the fixed point for the negative distribution parameters $\xi_\star$ need not be optimal in any sense, as long as it is approached by the sequence $\hat{\xi}_n$ sufficiently fast. Second, the proof itself begins by using the standard machinery for proving asymptotic normality of $M$-estimators for the joint set of parameters $\gamma := (\theta, \xi)$ (see e.g. Ferguson, 2017). In order to extract the marginal asymptotics for $\hat{\theta}_n$, while employing minimal assumptions regarding the convergence of $\hat{\xi}_n$, again the key step is to leverage Theorem 4.4, which states that $\theta_\star$ is optimal for the population objective $L(\cdot, \xi)$ *for any $\xi \in \Xi$*. This implies that the joint Hessian $\nabla^2_\gamma L(\theta_\star, \xi_\star)$ is block-diagonal, yielding the necessary simplification. Finally, we remark that the result in Ma & Collins (2018, Theorem 4.6) is a special case of Theorem 4.10 where (i) the negative distribution is held fixed, i.e., $\hat{\xi}_n = \xi_\star$ for all $n$, and (ii) the event spaces $\mathsf{X}$ and $\mathsf{Y}$ are finite.

---

[4]The matrix norm $\|\cdot\|_{\text{op}}$ in this work refers to the induced $\ell_2 \mapsto \ell_2$ operator norm, i.e., the largest singular value.

### 4.3 Learning the Proposal Distribution

Given that the consistency and asymptotic normality for the R-NCE estimator are guaranteed under the very mild condition that the negative proposal distribution parameters converge to a fixed point (cf. Theorem 4.10), there is considerable algorithmic flexibility in generating the sequence of proposal models, indexed by $\{\hat{\xi}_n\}$. We begin our discussion with first establishing the importance of having a learnable proposal distribution. Using the posterior probability notation (3.7), we can write the R-NCE gradient as:

$$\nabla_\theta L(\theta, \xi) = \mathbb{E}_{x,y} \nabla_\theta \mathcal{E}_\theta(x, y) - \mathbb{E}_x \mathbb{E}_{\bar{\mathbf{y}}|x;\xi} \left[ \sum_{i=1}^{K+1} q_{\theta,\xi}(i \mid x, \bar{\mathbf{y}}) \nabla_\theta \mathcal{E}_\theta(x, y_i) \right].$$

Further examination gives us the following key intuition about training: *the proposal distribution $p_\xi(y \mid x)$ should not be significantly less informative than the energy model $\mathcal{E}(x, y)$.* If $p_\xi(y \mid x)$ is uninformative, then training will stall. This is because the posterior distribution $q_{\theta,\xi}(\cdot \mid x, \bar{\mathbf{y}})$ will concentrate nearly all its mass on the positive example, as the energy of the true sample will be significantly higher than the energy of the negative samples. As a consequence, the R-NCE gradient approximately vanishes without necessarily having $\mathcal{E}(x, y) \propto \log p(y \mid x)$, stalling the training procedure. We term this phenomenon *posterior collapse*. Indeed, this is consistent with the findings in Liu et al. (2022); Lee et al. (2023) in the context of binary NCE, and motivates having a *learnable* (cf. Algorithm 1) negative sampling distribution $p_\xi(y \mid x)$ during training, to avoid early gradient collapses that result in a sub-optimal energy model. We next attempt to characterize the properties of an "optimal" negative sampler.

#### 4.3.1 Nearly Asymptotically Efficient Negative Sampler

In principle, the asymptotic normality result Theorem 4.10 gives us a recipe to characterize an optimal proposal model: simply select the distribution $p_\xi$ which minimizes the trace of the asymptotic variance matrix $V_\theta$ (which itself is a function of $p_\xi$). However, this is an intractable optimization problem in general (see e.g., Gutmann & Hyvärinen, 2012, Section 2.4). Nevertheless, we will show that considerable insight may be gained by setting the negative distribution $p_\xi(y \mid x)$ equal to the conditional data distribution $p(y \mid x)$.

To begin, we first define the conditional Fisher information matrix, which generalizes the classical Fisher information matrix from asymptotic statistics.

**Definition 4.11** (Fisher information matrix). *The conditional Fisher information matrix of a conditional density $p_\theta(y \mid x)$ at a point $x \in \mathsf{X}$ is defined as:*

$$I(\theta \mid x) := \mathbb{E}_{y \sim p_\theta(\cdot \mid x)}[\nabla_\theta \log p_\theta(y \mid x) \nabla_\theta \log p_\theta(y \mid x)^\mathsf{T}].$$

*Note that for our parameterization $\mathscr{F}$, we have:*

$$I(\theta \mid x) = \mathbb{E}_{y \sim p_\theta(\cdot \mid x)}[\nabla_\theta \mathcal{E}_\theta(x, y) \nabla_\theta \mathcal{E}_\theta(x, y)^\mathsf{T}] - \mathbb{E}_{y \sim p_\theta(\cdot \mid x)}[\nabla_\theta \mathcal{E}_\theta(x, y)] \mathbb{E}_{y \sim p_\theta(\cdot \mid x)}[\nabla_\theta \mathcal{E}_\theta(x, y)^\mathsf{T}].$$

Under suitable conditions similar to those in Theorem 4.10, it is well known that the inverse Fisher information matrix governs the asymptotic behavior of the maximum likelihood estimator:

$$\sqrt{n}(\hat{\theta}_n^{\text{mle}} - \theta_\star) \rightsquigarrow N(0, \mathbb{E}_x[I(\theta_\star \mid x)]^{-1}), \quad \text{where} \quad \hat{\theta}_n^{\text{mle}} \in \underset{\theta \in \Theta}{\text{argmax}} \frac{1}{n} \sum_{i=1}^n \log p_\theta(y_i \mid x_i).$$

Our next result shows that by selecting $p_\xi(y \mid x) = p(y \mid x)$, the R-NCE estimator has similar asymptotic efficiency as the MLE.

**Theorem 4.12.** *Consider the setting of Theorem 4.10, and suppose that $p_{\xi_\star}(y \mid x) = p(y \mid x)$. Then, the asymptotic variance matrix $V_\theta$ is:*

$$V_\theta = \left(1 + \frac{1}{K}\right) \mathbb{E}_x[I(\theta_\star \mid x)]^{-1}.$$

Theorem 4.12 extends a similar result for binary NCE in Gutmann & Hyvärinen (2012, Corollary 7) to the ranking NCE setting. As an aside, we note that $p_\xi(y \mid x) = p(y \mid x)$ is actually *not* the optimal choice for minimizing asymptotic variance (Chehab et al., 2022). Nevertheless, even for reasonable values of $K$, the sub-optimality factor is fairly well controlled.

We now compare Theorem 4.12 to Ma & Collins (2018, Theorem 4.6). In Ma & Collins (2018, Theorem 4.6), it is shown that regardless of the (fixed) proposal distribution $p_\xi$, one has that $\|V_\theta - \mathbb{E}_x[I(\theta_\star \mid x)]^{-1}\|_{\mathrm{op}} \leqslant O(1/\sqrt{K})$, where the $O(\cdot)$ hides constants depending on the proposal distribution $p_\xi$. These hidden constants are not made explicit, and are likely quite pessimistic. On the other hand, in Theorem 4.12, we show that under the idealized setting when $p_\xi(y \mid x) = p(y \mid x)$, we have a sharper multiplicative bound w.r.t. the maximum likelihood estimator. Furthermore, since our analysis and algorithm allows for learnable proposal distributions, $p_\xi(y \mid x) = p(y \mid x)$ is much more of a reasonable assumption for theoretical analysis.

The near-asymptotically efficient result of Theorem 4.12 strongly motivates a learnable proposal model, but still leaves undetermined the algorithm by which such a result can be attained. We next investigate adversarial training for partial clues.

**Is Adversarial Training the Answer?** Recently, several papers which study the binary version of the NCE objective have advocated for an adversarial "GAN-like" approach where $\xi$ is optimized via concurrently minimizing the NCE objective (Gao et al., 2020; Bose et al., 2018). This approach is justified from the viewpoint of "hard negative" sampling when, similar to our earlier discussion on posterior collapse, the proposal distribution is easily distinguishable by a sub-optimal EBM. The following result establishes a close connection between the near-optimal asymptotic variance in Theorem 4.12 and adversarial optimization, under a realizability assumption for the set of proposal distributions.

**Proposition 4.13.** *Suppose that $\mathscr{F}$ is realizable, and furthermore, the family of proposal distributions $\mathscr{F}_n$ is realizable. That is, there exists some non-empty $\Xi_\star \subset \Xi$ s.t. for every $\xi_\star \in \Xi_\star$, $p_{\xi_\star}(y \mid x) = p(y \mid x)$ for a.e. $(x, y)$. Then:*

$$\max_{\theta \in \Theta} \min_{\xi \in \Xi} L(\theta, \xi) = L(\theta_\star, \xi_\star) = \log\left(\frac{1}{K+1}\right), \tag{4.8}$$

*where $(\theta_\star, \xi_\star) \in \Theta_\star \times \Xi_\star$.*

Recall from Theorem 4.12 that if the fixed point for the proposal distribution $p_{\xi_\star}(\cdot \mid x)$ equals $p(\cdot \mid x)$, the asymptotic variance for $\hat{\theta}_n$ equals a $1 + 1/K$ multiplicative factor of the Cramér-Rao optimal variance. The result above corresponds precisely to such a scenario: given a realizable family of proposal distributions, the adversarial saddle-point for R-NCE is characterized by the equalities $p_{\theta_\star}(\cdot \mid x) = p_{\xi_\star}(\cdot \mid x) = p(\cdot \mid x)$. Thus, when a sufficiently rich family of proposal distributions (i.e., $\mathscr{F}_n$ is realizable) is paired with a tractable convergent algorithm that ensures $d(\hat{\xi}_n, \Xi_\star) \overset{n \to \infty}{\longrightarrow} 0$, there is *no additional benefit* to using adversarial R-NCE: the asymptotic distribution of $\hat{\theta}_n$ is unaffected.

Notwithstanding, our preferred setup assumes, for both computational and conceptual reasons, that $\mathscr{F}_n$ is *not* realizable. This is motivated by the fact that the proposal distribution is used both for sampling and likelihood evaluation within the R-NCE objective, and is thus a significantly simpler model for computational reasons. In the following, we investigate the asymptotic properties of adversarial R-NCE under such conditions.

### 4.3.2 Equilibria and Convergence for Adversarial R-NCE

Let us define the finite-sample maxmin objective as follows:

$$\max_{\theta \in \Theta} \min_{\xi \in \Xi} \ L_n(\theta, \xi) := \hat{\mathbb{E}}_{x,y} \ \bar{\ell}_{\theta,\xi}(x, y) \tag{4.9}$$

In order to study the statistical properties of this game, we first define two families of solutions: Nash and Stackelberg equilibria.

**Definition 4.14** (Maxmin Nash equilibria)**.** *Let $f : \Theta \times \Xi \to \mathbb{R}$. A pair $(\theta_\star, \xi_\star) \in \Theta \times \Xi$ is a* Nash equilibrium *w.r.t. $f$ if:*

$$f(\theta, \xi_\star) \leqslant f(\theta_\star, \xi_\star) \leqslant f(\theta_\star, \xi) \quad \forall (\theta, \xi) \in \Theta \times \Xi.$$

A Nash equilibrium is characterized as follows: taking the other "player" as fixed, each player has no incentive to change their "action". It is straightforward to verify from the proof of Proposition 4.13 that any pair $(\theta_\star, \xi_\star) \in \Theta_\star \times \Xi_\star$ is in fact a Nash equilibrium w.r.t. the population objective $L$. However, for general nonconcave-nonconvex objectives, e.g., the finite-sample game defined in (4.9), the existence of a global or local Nash equilibrium is not guaranteed (Jin et al., 2020). Thus, we need an alternative definition of optimality, which comes from considering games as sequential.

**Definition 4.15** (Maxmin Stackelberg equilibria)**.** *Let $f : \Theta \times \Xi \to \mathbb{R}$. A pair $(\bar{\theta}, \bar{\xi}) \in \Theta \times \Xi$ is a* Stackelberg equilibrium *w.r.t. $f$ if:*

$$\inf_{\xi' \in \Xi} f(\theta, \xi') \leqslant f(\bar{\theta}, \bar{\xi}) \leqslant f(\bar{\theta}, \xi) \quad \forall (\theta, \xi) \in \Theta \times \Xi.$$

That is, $\bar{\xi}$ minimizes $f(\bar{\theta}, \cdot)$, whereas $\bar{\theta}$ maximizes $\theta \mapsto \inf_{\xi' \in \Xi} f(\theta, \xi')$. In contrast to global Nash equilibrium, global Stackelberg equilibrium points are guaranteed to exist provided continuity of $f$ and compactness of $\Theta \times \Xi$ (Jin et al., 2020).

For what follows, we will study the finite-sample Stackelberg equilibrium points $(\hat{\theta}_n, \hat{\xi}_n)$:

$$\inf_{\xi \in \Xi} L_n(\theta, \xi) \leqslant L_n(\hat{\theta}_n, \hat{\xi}_n) \leqslant L_n(\hat{\theta}_n, \xi) \quad \forall \theta \in \Theta, \xi \in \Xi. \tag{4.10}$$

Prior to proving a consistency statement about the convergence of $(\hat{\theta}_n, \hat{\xi}_n)$, we first establish some basic facts regarding Stackelberg optimal pairs for the population game.

**Proposition 4.16.** *For any Stackelberg optimal pair $(\bar{\theta}, \bar{\xi})$ to the population game (4.8), it holds that $\bar{\theta} \in \Theta_\star$. Conversely, for any $\theta_\star \in \Theta_\star$, there exists an $\bar{\xi} \in \Xi$ such that $(\theta_\star, \bar{\xi})$ is Stackelberg optimal for the population game (4.8).*

We now establish convergence of the finite-sample Stackelberg optimal pairs $(\hat{\theta}_n, \hat{\xi}_n)$ to a Stackelberg optimal pair for the population objective.

**Theorem 4.17** (Adversarial Consistency)**.** *Suppose that Assumption 4.1 and Assumption 4.2 hold. Let $(\hat{\theta}_n, \hat{\xi}_n) \in \Theta \times \Xi$ be a sequence of Stackelberg optimal pairs, as defined in (4.10), for the finite sample game (4.9). Then, the following results hold:*

1. *Suppose that $\Theta_\star$ is contained in the interior of $\Theta$. We have that $d(\hat{\theta}_n, \Theta_\star) \overset{n \to \infty}{\longrightarrow} 0$ a.s.*

2. *Suppose furthermore that Assumption 4.9 holds. Let $\bar{\Xi} := \{\xi \in \Xi \mid L(\theta_\star, \xi) = \inf_{\xi' \in \Xi} L(\theta_\star, \xi')\}$ for an arbitrary choice of $\theta_\star \in \Theta_\star$ (the set definition is independent of this choice), and suppose $\bar{\Xi}$ is contained in the interior of $\Xi$. We have that $d(\hat{\xi}_n, \bar{\Xi}) \overset{n \to \infty}{\longrightarrow} 0$ a.s.*

We note that part (1) of the consistency result is perhaps not surprising in light of Theorem 4.8, which merely assumed that $\hat{\theta}_n$ maximizes the finite-sample objective $L_n$ for any particular $\hat{\xi}_n$. In particular, no assumptions are placed upon the negative distribution parameter sequence $\{\hat{\xi}_n\}$. Under a mild regularity assumption however, we are additionally able to show convergence of the noise distribution parameters to a population Stackelberg adversary. In the setting where the family of proposal distributions $\mathscr{F}_n$ is realizable, the set $\bar{\Xi}$ is indeed the Nash optimal set $\Xi_\star$, as defined in Proposition 4.13. Next, we characterize the asymptotic normality of the sequence of finite-sample Stackelberg-optimal pairs $(\hat{\theta}_n, \hat{\xi}_n)$.

**Theorem 4.18** (Adversarial Normality)**.** *Suppose that $\mathscr{F}$ is realizable and Assumptions 4.1, 4.2, and 4.9 hold. Let $\hat{\gamma}_n := (\hat{\theta}_n, \hat{\xi}_n) \in \Theta \times \Xi$ be a sequence of Stackelberg optimal pairs, as defined in (4.10), for the finite sample game (4.9), and define $\bar{\Xi}$ as in Theorem 4.17. Assume then that the following properties hold:*

1. *The sets $\Theta_\star, \bar{\bar{\Xi}}$ are singletons, i.e., $\Theta_\star = \{\theta_\star\}$, and $\bar{\bar{\Xi}} = \{\bar{\xi}\}$, and $\gamma_\star := (\theta_\star, \bar{\xi})$ lies in the interior of $\Theta \times \Xi$.*

2. *The Hessian of $\bar{\ell}_\gamma$ is Lipschitz-continuous, i.e.,*

$$\|\nabla_\gamma^2 \bar{\ell}_{\gamma_1}(z) - \nabla_\gamma^2 \bar{\ell}_{\gamma_2}(z)\|_{\mathrm{op}} \leqslant M(z)\|\gamma_1 - \gamma_2\|, \quad \mathbb{E}_z\left[M(z)\right] < \infty, \ \forall \gamma_1, \gamma_2 \in \Theta \times \Xi.$$

3. *The $(\theta, \theta)$ block of the Hessian $\nabla_\gamma^2 L(\theta_\star, \bar{\xi})$, denoted $\nabla_\theta^2 L(\theta_\star, \bar{\xi})$, satisfies $\nabla_\theta^2 L(\theta_\star, \bar{\xi}) \prec 0$, and the $(\xi, \xi)$ block, denoted $\nabla_\xi^2 L(\theta_\star, \bar{\xi})$, satisfies $\nabla_\xi^2 L(\theta_\star, \bar{\xi}) \succ 0$.*

*Then, $\sqrt{n}(\hat{\gamma}_n - \gamma_\star) \rightsquigarrow \mathcal{N}(0, V_\gamma^a)$, where*

$$V_\gamma^a = \mathbb{E}_z\left[\nabla_\gamma^2 \bar{\ell}_{\gamma_\star}(z)\right]^{-1} \mathrm{Var}_z\left[\nabla_\gamma \bar{\ell}_{\gamma_\star}(z)\right] \mathbb{E}_z\left[\nabla_\gamma^2 \bar{\ell}_{\gamma_\star}(z)\right]^{-1}. \tag{4.11}$$

A particularly important corollary of this result is as follows:

**Corollary 4.19** (Marginal Normality)**.** *Under the assumptions of Theorem 4.18, the parameters $\{\theta_n\}$ satisfy $\sqrt{n}(\hat{\theta}_n - \theta_\star) \rightsquigarrow \mathcal{N}(0, V_\theta^a)$, where*

$$V_\theta^a = -\nabla_\theta^2 L(\theta_\star, \bar{\xi})^{-1}$$

The variance $V_\theta^a$ is similar to $V_\theta$ in Theorem 4.10, with the difference being the specification of a particular fixed-point for the proposal distribution parameters: the Stackelberg adversary of $\theta_\star$. In the special case when the family of proposal distributions is realizable, since the fixed-point involves the Nash adversary $\xi_\star$ (cf. Proposition 4.13), these two variances coincide. However, when we lack realizability of the proposal distributions, we are left to compare the asymptotic variance $V_\theta$ from Theorem 4.10 with the new asymptotic variance $V_\theta^a$. Given the quite non-trivial relationship between the proposal parameters that a non-adversarial negative sampler converges to versus the Stackelberg adversary, it is not clear that a general statement can be made comparing these two variances. Therefore, the theoretical benefits of adversarial training are quite unclear.

Furthermore, actually finding Stackelberg solutions comes with its own set of challenges in practice. Computationally, differentiating the R-NCE objective with respect to the negative proposal parameters would involve differentiating both the negative samples $\mathbf{y}$ as well as the likelihoods $\log p_\xi(\cdot \mid x)$, a non-trivial computational bottleneck. In contrast, differentiating the negative samples $\mathbf{y}$ with respect to the proposal model parameters $\xi$ is *not required* in the non-adversarial setting (cf. Remark 3.2). For this work, we assume that the independent optimization of $\xi$ via a suitable objective for the generative model $p_\xi$ is sufficient to optimize the EBM, without assuming realizability for $\mathscr{F}_n$. We leave the explicit minimization of some function of $V_\theta$ to future work.

Finally, we conclude by noting that the proofs of Theorem 4.17 (adversarial consistency) and Theorem 4.18 (adversarial normality) are based off of ideas from Biau et al. (2020), who study the asymptotics of GAN training. The main difference is that we are able to exploit specific properties of the R-NCE objective to derive simpler expressions for the resulting asymptotic variances.

## 4.4 Typical Training Profiles

The results of Section 4.2 and Section 4.3 focus on the asymptotic properties of R-NCE training, and do not consider the optimization aspects of the problem. Here, in Figure 2, we characterize typical training curves that one can expect when jointly training an EBM and negative proposal distribution with Algorithm 1.

## 4.5 Convergence of Divergences

To conclude this section, we translate asymptotic normality in the parameter space to convergence of both KL-divergence and relative Fisher information. For two measures $\mu$ and $\nu$ over the same measure space, the KL-divergence and the relative Fisher information of $\mu$ w.r.t. $\nu$ are defined as:

$$\mathrm{KL}(\mu \parallel \nu) := \mathbb{E}_\mu\left[\log\left(\frac{\mu}{\nu}\right)\right], \quad \mathrm{FI}(\mu \parallel \nu) := \mathbb{E}_\mu\left[\left\|\nabla \log\left(\frac{\mu}{\nu}\right)\right\|^2\right].$$

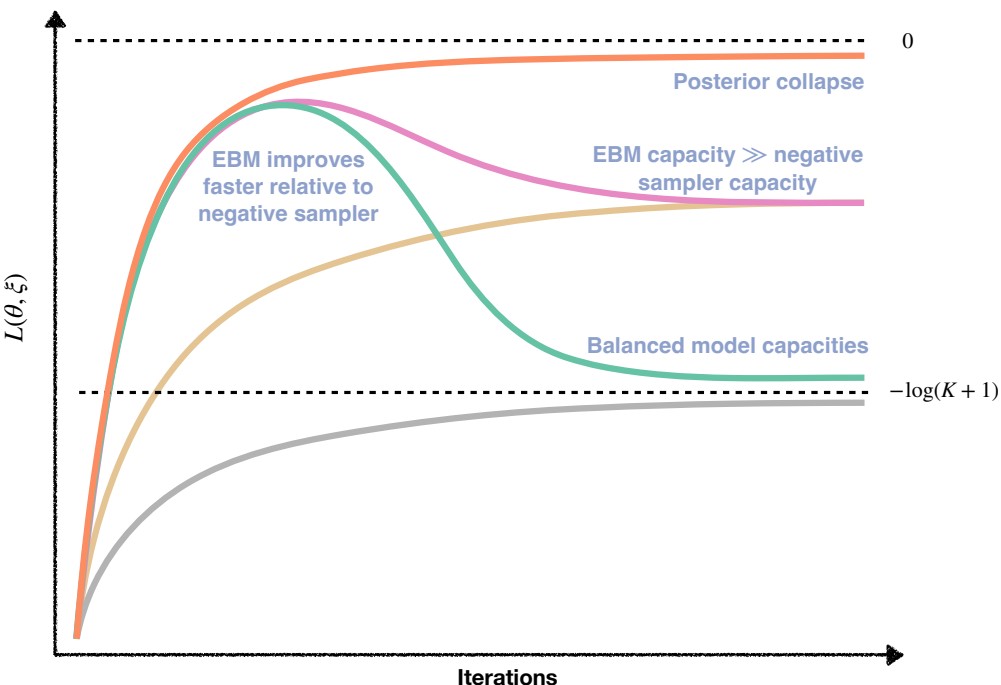

**Figure 2:** An illustration of the different qualitative behaviors that R-NCE training curves can exhibit. The 🟧 curve illustrates an example of posterior collapse, where the negative sampler is completely un-informative, leading to a trivial classification problem. Both the 🟩 and 🟪 curves illustrate a phase transition in the training: at first the EBM learns quicker relative to the negative sampler (and hence the classification accuracy increases), but then the negative sampler catches up in learning, resulting in a drop in the R-NCE objective. The main difference between the 🟩 and 🟪 curves is the final value each curve converges towards. The 🟩 curve tends toward $-\log(K+1)$, which indicates that both the EBM and negative sampler have sufficient capacity to represent the true distribution $p(y \mid x)$ (cf. Equation (3.6) in the case where $p_{\theta_\star}(\cdot \mid x) = p_{\xi_\star}(\cdot \mid x) = p(\cdot \mid x)$). On the other hand, the 🟪 curve tends towards a final value larger than $-\log(K+1)$ but bounded away from zero, indicating that the EBM capacity exceeds that of the negative sampler. The remaining 🟨 and ⬜ curves represent when the negative sampler learns faster than the EBM; this occurs when, for example, a pre-trained negative sampler is used. Here, the training curve approaches its final value from below. As before, the difference between the final value of both the 🟨 and ⬜ curves is due to negative sampler model capacity, with the 🟨 curve denoting when the EBM has much larger capacity than the negative sampler, and the ⬜ curve denoting when the model capacities are relatively balanced.

Here, we have (a) overloaded the same notation for measures and densities, and (b) in the definition of relative Fisher information we have assumed smoothness of the densities. The following propositions are straightforward applications of the second-order delta method.

**Proposition 4.20.** *Suppose that $\theta_\star \in \Theta_\star$, and that $\{\hat{\theta}_n\}$ is a sequence of estimators satisfying $\sqrt{n}(\hat{\theta}_n - \theta_\star) \rightsquigarrow \mathcal{N}(0, \Sigma_\star)$. Then:*

$$n \cdot \mathbb{E}_x[\mathrm{KL}(p(y \mid x) \| p_{\hat{\theta}_n}(y \mid x))] \rightsquigarrow \frac{1}{2}\|Z\|^2, \quad Z \sim \mathcal{N}(0, \Sigma_\star^{1/2}\mathbb{E}_x[I(\theta_\star \mid x)]\Sigma_\star^{1/2}).$$

Hence, in the setting of Theorem 4.10, if $p_{\xi_\star}(y \mid x) = p(y \mid x)$ and if $\Theta \subset \mathbb{R}^p$, then we have:

$$n \cdot \mathbb{E}_x[\mathrm{KL}(p(y \mid x) \| p_{\hat{\theta}_n}(y \mid x))] \rightsquigarrow \frac{1}{2}\left(1 + \frac{1}{K}\right)\chi^2(p), \tag{4.12}$$

where $\chi^2(p)$ is a Chi-squared distribution with $p$ degrees of freedom. Now turning to the relative Fisher information, we have the following result.

**Proposition 4.21.** *Suppose that $\theta_\star \in \Theta_\star$, and that $\{\hat{\theta}_n\}$ is a sequence of estimators satisfying $\sqrt{n}(\hat{\theta}_n - \theta_\star) \rightsquigarrow \mathcal{N}(0, \Sigma_\star)$. Then:*

$$n \cdot \mathbb{E}_x[\mathrm{FI}(p(y \mid x) \parallel p_{\hat{\theta}_n}(y \mid x))] \rightsquigarrow \|Z\|^2, \quad Z \sim \mathcal{N}(0, \Sigma_\star^{1/2} \mathbb{E}_x[J(\theta_\star \mid x)]\Sigma_\star^{1/2}),$$

*where $J(\theta \mid x) := \mathbb{E}_{y \sim p_\theta(\cdot \mid x)}[\partial_\theta \nabla_y \mathcal{E}_\theta(x, y)^\mathsf{T} \partial_\theta \nabla_y \mathcal{E}_\theta(x, y)]$.*

The asymptotic variance from Proposition 4.21 is more difficult to interpret than the corresponding asymptotic variance from Proposition 4.20. Unfortunately in general, there is no relationship between the two, as we will see shortly. In order to aid our interpretation, suppose the energy model follows the form of an exponential family, i.e., $\mathcal{E}_\theta(x, y) = \langle \psi(x, y), \theta \rangle$, for some fixed sufficient statistic $\psi$. Then, a straightforward computation yields the following:

$$I(\theta \mid x) = \mathrm{Cov}_{p_\theta(y \mid x)}(\psi(x, y)), \quad J(\theta \mid x) = \mathbb{E}_{p_\theta(y \mid x)}[\partial_y \psi(x, y)\partial_y \psi(x, y)^\mathsf{T}].$$

Hence, assuming as in (4.12) that $p_{\xi_\star}(y \mid x) = p(y \mid x)$, then the asymptotic variance is determined by the variance of the smoothness of the sufficient statistic (in the event variable $y$) relative to the covariance of the sufficient statistic itself. Let us now consider a specific example. Suppose that $p_\theta(y \mid x) = \mathcal{N}(\mu, \sigma^2)$ (so the context variable $x$ is not used). Here, the sufficient statistic is $\psi(x, y) = (y, y^2)$. A straightforward computation yields that:

$$\mathrm{tr}(\mathbb{E}_x[I(\theta \mid x)]^{-1}\mathbb{E}_x[J(\theta \mid x)]) = \frac{3}{\sigma^2} + \frac{4\mu^2}{\sigma^4}.$$

Thus, depending on the specific values of the parameters $(\mu, \sigma^2)$, the asymptotic variance quantity above can be either much smaller or much larger than 2.

## 5 Interpolating EBMs

Recent methods such as diffusion models (Sohl-Dickstein et al., 2015; Ho et al., 2020; Song et al., 2020; 2021b) and stochastic interpolants (Albergo & Vanden-Eijnden, 2023; Albergo et al., 2023) advocate for modeling a *family* of distributions with a single shared network, indexed by a single scale parameter lying in a fixed interval. At one end of this interval, the distribution corresponds to standard Gaussian noise, while the other end models the true data distribution. Sampling from this generative model begins with sampling a "latent" vector $z \in \mathsf{Y}$ from the base Gaussian distribution, and then propagating this sample through a sequence of pushforwards over the interval. This propagation can either refer to a fixed discrete sequence of Gaussian models, or solving a continuous-time SDE or ODE. The strength of such a framework has been attributed to an implicit annealing of the data distribution over the interval (Song & Ermon, 2019), easing the generative process by stepping through a sequence of distributions instead of jumping directly from pure noise to the data.

### 5.1 Stochastic Interpolants

A useful instantiation of this framework that subsequently allows generalizing all other interval-based models as special cases, corresponds to the stochastic interpolants formulation (Albergo & Vanden-Eijnden, 2023). In this setup, a continuous stochastic process over the interval $[0, 1]$ is implicitly constructed by defining an *interpolant function* $I_t : \mathsf{Y} \times \mathsf{Y} \to \mathsf{Y}$, $t \in [0, 1]$, where $I_0(z, y) = z$ and $I_1(z, y) = y$. For a given $x$, define the continuous-time stochastic *process* $Y_t \mid x$ as:

$$y_t \mid x = I_t(z, y), \quad z \sim \mathcal{N}(0, I), \quad y \sim \mathsf{P}_{Y \mid x}, \quad z \perp y.$$

By construction, the law of $Y_t \mid x$ at $t = 1$ is exactly $\mathsf{P}_{Y \mid x}$. An example interpolant function from Albergo & Vanden-Eijnden (2023) is given as $I_t(z, y) := \cos(\frac{1}{2}\pi t)z + \sin(\frac{1}{2}\pi t)y$. The first key result from Albergo & Vanden-Eijnden (2023) is that *law* of the process $Y_t \mid x$, denoted as $\mathsf{P}_{Y_t \mid x}$ with associated time-varying density $p^t(y_t \mid x)$, satisfies the following continuity equation:

$$\partial_t p^t + \mathrm{div}_y(v_t p^t) = 0, \tag{5.1}$$

where $v_t : [0,1] \times \mathsf{X} \times \mathsf{Y} \to \mathbb{R}^{\dim(\mathsf{Y})}$ is a particular time-varying velocity field (described subsequently), yielding the following continuous normalizing flow (CNF) generative model:

$$\frac{\mathrm{d}y_{t|x}(z)}{\mathrm{d}t} = v_t(x, y_{t|x}(z)), \quad y_{t=0|x}(z) = z \sim \mathcal{N}(0, I). \tag{5.2}$$

The distribution of the samples from this model at $t = 1$ correspond to the desired distribution $\mathsf{P}_{Y|x}$. One can now parameterize any function approximator $\hat{v}_{t,\xi}$ to approximate this vector field.

While CNFs as a generative modeling framework are certainly not new, their training has previously relied on leveraging the associated log-probability ODE (Chen et al., 2018; Grathwohl et al., 2019) (cf. Section 6), corresponding to the change-of-variables formula, and using maximum likelihood estimation as the optimization objective. Unfortunately, solving this ODE involves computing the Jacobian of the parameterized vector field $\hat{v}_{t,\xi}$ with respect to $y_t$; gradients of the objective w.r.t. $\xi$ therefore require 2nd-order gradients of $\hat{v}_{t,\xi}$. As a result of these computational bottlenecks, CNFs have been largely overtaken as a framework for generative models. However, the second key result from Albergo & Vanden-Eijnden (2023) is that the true vector field $v_t$ is given as the minimum of a simple quadratic objective:

$$v_t(x, \cdot) := \underset{\hat{v}_t(x, \cdot)}{\operatorname{argmin}} \quad \mathbb{E}_z \mathbb{E}_{y \sim \mathsf{P}_{Y|x}} \left[ \|\hat{v}_t(x, I_t(z, y))\|^2 - 2\partial_t I_t(z, y) \cdot \hat{v}_t(x, I_t(z, y)) \right] \quad \forall x, t.$$

This now enables the use of expressive models for $\hat{v}_{t,\xi}$, paired with the following population optimization objective:

$$\min_{\xi} \quad \mathbb{E}_{(x,y) \sim \mathsf{P}_{X,Y}} \mathbb{E}_{t \sim \Lambda} \mathbb{E}_z \quad \left[ \|\hat{v}_{t,\xi}(x, I_t(z, y))\|^2 - 2\partial_t I_t(z, y) \cdot \hat{v}_{t,\xi}(x, I_t(z, y)) \right], \tag{5.3}$$

where $\Lambda$ is a fixed distribution over the interval $[0, 1]$ (e.g., uniform).

## 5.2 Building Interpolating EBMs

A natural way of using the Interpolant-CNF model described previously is to use the pushforward distribution at $t = 1$ as the proposal generative model $p_\xi(y \mid x)$, with a relatively simple parameterized vector field $\hat{v}_{t,\xi}$. Paired with a parameterized energy function $\mathcal{E}_\theta(x, y)$, one can now proceed with optimizing both models as outlined in Algorithm 1.

However, by truncating the flow at any intermediate time $t \in (0, 1)$, the CNF gives an approximation $p_\xi^t(y_t \mid x)$ of the distribution $p^t(y_t \mid x)$. Thus, we can define a time-indexed EBM $\mathcal{E}_\theta(x, y, t)$ to approximate this same distribution as:

$$p^t(y_t \mid x) \propto \exp(\mathcal{E}_\theta(x, y_t, t)).$$

We refer to the above model as an Interpolating-EBM (I-EBM). Notice that a single shared network $\mathcal{E}_\theta$ is used to represent the distribution across all $t \in [0, 1]$.

By pairing the EBM at time $t$ with the *truncated* CNF at time $t$ as the proposal model, we can straightforwardly define a time-indexed R-NCE objective; see Figure 3 for an illustration.

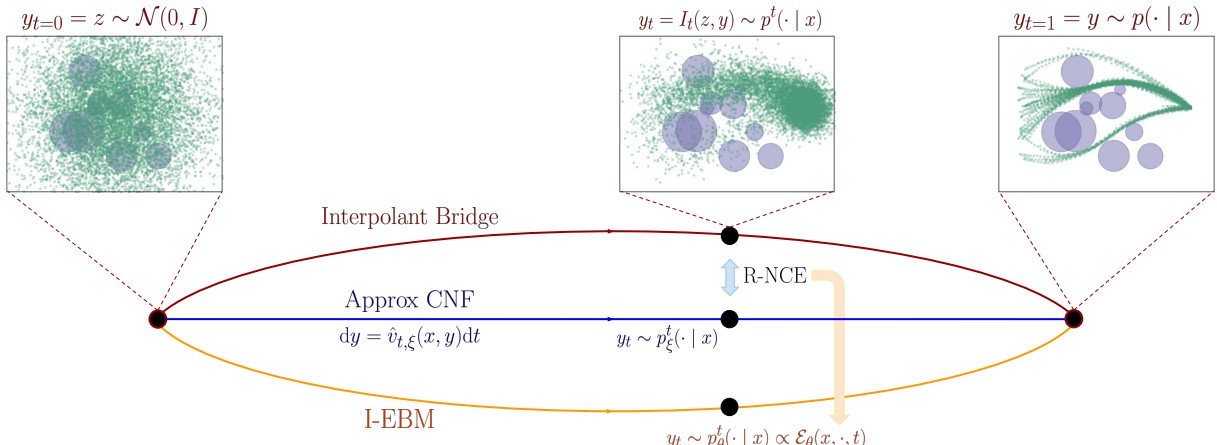

**Figure 3:** An illustration of the Interpolating-EBM (I-EBM) framework applied to a motion planning problem. Here, the event space corresponds to the waypoint trajectory (green circles) of the robot in the obstacle environment. The interpolant function $I_t$ defines a continuous time stochastic bridge process over $t \in [0, 1]$ between Gaussian noise at $t = 0$ and the true conditional distribution at $t = 1$; the time-varying density of this process is notated as $p^t(\cdot \mid x)$. The CNF with (a relatively simply parameterized) vector field $\hat{v}_{t,\xi}$ is an approximation of this continuous-time process, and is independently trained using (5.3); the truncation of the flow of the CNF at time $t$ yields samples from the distribution $p_\xi^t(\cdot \mid x)$. The I-EBM defines a family of (more expressively parameterized) EBMs $p_\theta^t(\cdot \mid x)$, indexed by $t$, to better approximate $p^t$. At a given time $t$, we pair $K$ contrastive samples from the CNF and a true sample from the interpolant bridge to define a time-indexed R-NCE objective for $\mathcal{E}_\theta$.

In particular, let $K \in \mathbb{N}_+$, the number of negative samples defining the R-NCE objective be fixed as before. For a given $x$, let $\mathsf{P}_{\mathbf{y}_t|x;\xi}^K$ denote the product conditional sampling distribution over $\mathsf{Y}^K$ where $\mathbf{y}_t \sim \mathsf{P}_{\mathbf{y}_t|x;\xi}^K$ denotes a random vector $\mathbf{y}_t = (y_{t_k})_{k=1}^K \in \mathsf{Y}^K$ with $y_{t_k} \sim p_\xi^t(\cdot \mid x)$ for $k = 1, \ldots, K$.

Now, for a given $x \sim \mathsf{P}_X$, $t \in (0, 1]$, *positive* sample $y_t \sim \mathsf{P}_{Y_t|x}$, *negative* samples $\mathbf{y}_t \sim \mathsf{P}_{\mathbf{y}_t|x;\xi}^K$, and parameters $\theta \in \Theta$, $\xi \in \Xi$, define:

$$\ell_{\theta,\xi}(x, y_t, t \mid \mathbf{y}_t) := \log \left[ \frac{\exp(\mathcal{E}_\theta(x, y_t, t) - \log p_\xi^t(y_t \mid x))}{\sum_{y' \in \{y_t\} \cup \mathbf{y}_t} \exp(\mathcal{E}_\theta(x, y', t) - \log p_\xi^t(y' \mid x))} \right], \tag{5.4}$$

as the R-NCE loss at time $t$. We can thus define the overall population objective as:

$$L(\theta, \xi) := \mathbb{E}_{t \sim \Lambda} \mathbb{E}_{x \sim \mathsf{P}_X} \mathbb{E}_{y_t \sim \mathsf{P}_{Y_t|x}} \mathbb{E}_{\mathbf{y}_t \sim \mathsf{P}_{\mathbf{y}_t|x;\xi}^K} [\ell_{\theta,\xi}(x, y_t, t \mid \mathbf{y}_t)]. \tag{5.5}$$

Notice that this is simply an expectation of a time-indexed population R-NCE objective. We refer to the objective above as the Interpolating-R-NCE (I-R-NCE) objective. Provided that the function class of time-indexed conditional energy models:

$$\mathscr{F}_t := \{\mathcal{E}_\theta(x, y, t) : \mathsf{X} \times \mathsf{Y} \to \mathbb{R} \mid \theta \in \Theta\},$$

is realizable (cf. Definition 4.3) for *all* $t \in (0, 1]$, i.e., there exists a $\theta_\star \in \Theta$ s.t. for all $t \in (0, 1]$ and a.e. $(x, y_t)$:

$$p_{\theta_\star}^t(y_t \mid x) := \frac{\exp(\mathcal{E}_{\theta_\star}(x, y_t, t))}{\int_\mathsf{Y} \exp(\mathcal{E}_{\theta_\star}(x, y, t)) \, \mathrm{d}\mu_Y} = p^t(y_t \mid x),$$

then the population objective in (5.5) is maximized by $\theta_\star$ for each $t$.

### 5.2.1 Training

The adjustment to Algorithm 1 for training is straightforward for I-EBMs. In particular, for each $(x_i, y_i) \in \mathcal{B}$, we sample $m$ values of $t$, i.e., $\{t_{ij}\}_{j=1}^m \sim \Lambda$, latents $\{z_{ij}\}_{j=1}^m \sim \mathcal{N}(0, I)$, and negative-sample sets $\{\mathbf{y}_{ij}\}_{j=1}^m$

where $\mathbf{y}_{ij} \sim \mathsf{P}^K_{\mathbf{y}_{t_{ij}}|x_i;\xi}$. The net R-NCE loss for $(x_i, y_i)$ is then defined as the average of the R-NCE loss over $\{(x_i, y_{ij}, t_{ij}, \mathbf{y}_{ij})\}^m_{j=1}$, where $y_{ij} = I_{t_{ij}}(z_{ij}, y_i)$. The training pseudocode is presented in Algorithm 3, with the modifications to Algorithm 1 highlighted in teal. We note that choosing values of $m > 1$ serves as variance reduction, and we also apply a similar variance reduction trick to the stochastic interpolant loss in Line 8.

---

**Algorithm 3:** Interpolating-R-NCE with learnable negative sampler

**Data:** Dataset $\mathcal{D} = \{(x_i, y_i)\}^n_{i=1}$, number of outer steps $T_{\mathrm{outer}}$, number of sampler steps per outer step $T_{\mathrm{samp}}$, number of R-NCE steps per outer step $T_{\mathrm{rnce}}$, number of negative samples $K$, interpolating time distribution $\Lambda$, number of interpolating times $m \in \mathbb{N}_{\geqslant 1}$.

**Result:** Energy model parameters $\theta$, negative sampler parameters $\xi$.

**1** Initialize $\theta$ and $\xi$.

**2** Set $\ell_{\mathrm{vf},\xi}(x, z, y, t) = \|v_{t,\xi}(x, I_t(z, y))\|^2 - 2\partial_t I_t(z, y) \cdot v_{t,\xi}(x, I_t(z, y))$.

**3** Set $\hat{L}_{\mathrm{interp}}(\xi; \mathcal{B}) = \sum_{(x_i, y_i, \{(z_{ij}, t_{ij})\}^m_{j=1}) \in \mathcal{B}} \frac{1}{m} \sum^m_{j=1} \ell_{\mathrm{vf},\xi}(x_i, z_{ij}, y_i, t_{ij})$.

**4** Set $\hat{L}_{\mathrm{irnce}}(\theta, \xi; \bar{\mathcal{B}}) = -\sum_{(x_i, \{(y_{ij}, t_{ij}, \mathbf{y}_{ij})\}^m_{j=1}) \in \bar{\mathcal{B}}} \frac{1}{m} \sum^m_{j=1} \ell_{\theta,\xi}(x_i, y_{ij}, t_{ij} \mid \mathbf{y}_{ij})$.

**5** **for** $T_{\mathrm{outer}}$ *iterations* **do**

**6**      **for** $T_{\mathrm{samp}}$ *iterations* **do**

**7**          $\mathcal{B} \leftarrow$ batch of $\mathcal{D}$.

         // Augment batch with latent variables and times.

**8**          $\mathcal{B} \leftarrow \{(x_i, y_i, \{(z_{ij}, t_{ij})\}^m_{j=1}) \mid (x_i, y_i) \in \mathcal{B}\}$, with $t_{ij} \sim \Lambda$, $z_{ij} \sim \mathcal{N}(0, I)$.

**9**          $\xi \leftarrow$ minimize using $\nabla_\xi \hat{L}_{\mathrm{interp}}(\xi; \mathcal{B})$.

**10**      **end**

**11**      **for** $T_{\mathrm{rnce}}$ *iterations* **do**

**12**          $\mathcal{B} \leftarrow$ batch of $\mathcal{D}$.

         // Convert batch to time-indexed positive and negative samples.

**13**          $\bar{\mathcal{B}} \leftarrow \{(x_i, \{(y_{ij}, t_{ij}, \mathbf{y}_{ij})\}^m_{j=1}) \mid (x_i, y_i) \in \mathcal{B}\}$, with $t_{ij} \sim \Lambda$, $z_{ij} \sim \mathcal{N}(0, I)$, $y_{ij} = I_{t_{ij}}(z_{ij}, y_i)$,
         $\mathbf{y}_{ij} \sim \mathsf{P}^K_{\mathbf{y}_{t_{ij}}|x_i;\xi}$.

**14**          $\theta \leftarrow$ minimize using $\nabla_\theta \hat{L}_{\mathrm{irnce}}(\theta, \xi; \bar{\mathcal{B}})$.

**15**      **end**

**16**      **return** $(\theta, \xi)$

**17** **end**

---

**Remark 5.1.** *A naïve implementation of variance reduction for the R-NCE loss in Algorithm 3 would generate $m$ independent sets of $K$ negative samples, i.e., $\{\mathbf{y}_{ij}\}^m_{j=1}$, corresponding to the $m$ intermediate times $\{t_{ij}\}^m_{j=1}$. To make this more efficient, we allow the negative sample sets to be correlated across time by generating a single set of $K$ trajectories, spanning the interval $(0, \max_j t_{ij}]$, and sub-sampling these trajectories at $m$ intermediate times.*

### 5.2.2 Sampling

In follow-up work (Albergo et al., 2023), the stochastic equivalent of the transport PDE (5.1) is derived using the Fokker-Planck equation, which yields the following, *equivalent in law*, SDE-based generative model:

$$\mathrm{d}y_{t|x}(z) = \Big[\underbrace{v_t(x, y_{t|x}(z)) + \eta \nabla_y \log p^t(y_{t|x}(z) \mid x)}_{:=\boldsymbol{v}^\eta_t(x, y_{t|x}(z))}\Big]\mathrm{d}t + \sqrt{2\eta}\,\mathrm{d}W_t, \quad y_{t=0|x}(z) = z \sim \mathcal{N}(0, I), \qquad (5.6)$$

where $\eta \geqslant 0$ and $W_t$ is the standard Wiener process. Given the simplicity of the former ODE-based model in (5.2), a natural question is: why is the SDE-based model necessary? The answer lies in studying the divergence between the true conditional distribution $p(y \mid x)$ and the approximate distribution $p^{t=1}_\xi(y \mid x)$ induced by the learned vector field $\hat{v}_{t,\xi}$. In particular, from Albergo et al. (2023, Lemma 2.19), for a CNF

parameterized by the approximate vector field $\hat{v}_{t,\xi}$:

$$\text{KL}(p(y \mid x) \parallel p_\xi^{t=1}(y \mid x)) =$$
$$\int_0^1 \int_{\mathsf{Y}} (\nabla_y \log p_\xi^t(y \mid x) - \nabla_y \log p^t(y \mid x)) \cdot (\hat{v}_{t,\xi}(x,y) - v_t(x,y)) \, p^t(y \mid x) \, \mathrm{d}\mu_Y \mathrm{d}t.$$

It follows that control over the vector field error, i.e., the objective minimized in (5.3), does not provide any direct control over the Fisher divergence, i.e., error in the learned score function $\nabla_y \log p_\xi^t(y \mid x)$, thus preventing any control over the target KL divergence.

In contrast, let $\hat{\boldsymbol{v}}_t^\eta(x,y)$ be an approximation of the drift term $\boldsymbol{v}_t^\eta(x,y)$ in (5.6). Define the following approximate generative model and associated Fokker-Planck transport PDE for the induced density $\hat{p}^t(y_t \mid x)$:

$$\mathrm{d}y_{t|x}(z) = \hat{\boldsymbol{v}}_t^\eta(x, y_{t|x}(z)) \, \mathrm{d}t + \sqrt{2\eta} \, \mathrm{d}W_t, \quad y_{t|x}(z) = z \sim \mathcal{N}(0, I) \tag{5.7}$$
$$\partial_t \hat{p}^t + \mathrm{div}_y(\hat{\boldsymbol{v}}_t^\eta \hat{p}^t) = \eta \Delta \hat{p}^t. \tag{5.8}$$

Then, Albergo et al. (2023, Lemma 2.20) states that:

$$\text{KL}(p(y \mid x) \parallel \hat{p}^{t=1}(y \mid x)) \leqslant \frac{1}{4\eta} \int_0^1 \int_{\mathsf{Y}} \|\boldsymbol{v}_t^\eta(x,y) - \hat{\boldsymbol{v}}_t^\eta(x,y)\|^2 \, p^t(y \mid x) \, \mathrm{d}\mu_Y \mathrm{d}t.$$

Thus, if one can control the net approximation error of the SDE drift, then the SDE-based generative model should yield samples which are closer to the target distribution in KL. While error in the velocity field $\|v_t - \hat{v}_{t,\xi}\|$ is controllable via the minimization of (5.3), the I-EBM framework gives us a handle on the other component to the SDE drift: an approximation of the time-varying score function,[5] i.e., $\nabla_y \mathcal{E}_\theta(x, y_t, t) \approx \nabla_y \log p^t(y_t \mid x)$. Indeed, from Proposition 4.21, convergence to the global maximizer $\theta_\star$ yields controllable convergence (in distribution) on the relative Fisher information between $p^t(y_t \mid x)$ and $p_{\theta_\star}^t(y_t \mid x)$. Algorithm 4 outlines a three-stage procedure for sampling with I-EBM.

---

**Algorithm 4:** I-EBM: Sample from $p_\theta(y \mid x)$ for Euclidean $\mathsf{Y}$

**Data:** Context $x \in \mathsf{X}$, energy model parameters $\theta$, negative sampler $\xi$, initial sample time $\underline{t} \in (0,1)$, MCMC steps $T_{\text{mcmc}}$, MCMC step size $\eta$.

**Result:** Sample $y \sim p_\theta(y \mid x)$.

**1** Set $y_{\underline{t}} \sim p_\xi^{\underline{t}}(\cdot \mid x)$.
   // Leverage the SDE in (5.7) to transport $y_{\underline{t}}$ from $t = \underline{t}$ to $t = 1$.
**2** $y' \leftarrow \texttt{SDESolve}(\hat{\boldsymbol{v}}_t^\eta(x,y) = \hat{v}_{t,\xi}(x,y) + \eta \nabla_y \mathcal{E}_\theta(x, y, t); \, [\underline{t}, 1], y_{\underline{t}})$.
   // Finite-step MCMC with $\mathcal{E}_\theta(x, y, 1)$ starting at $y'$ (cf. Algorithm 2).
**3** $y \leftarrow \texttt{MCMC}(T_{\text{mcmc}}, \eta, y')$.
**4** **return** $y$

---

Notice in Line 1, we use the CNF parameterized by vector field $\hat{v}_{t,\xi}$, with the flow truncated at time $\underline{t}$ to form the initial sample. This is reasonable since for $t \ll 1$, the proposal model $p_\xi^t$ becomes more accurate with the true process $Y_t \mid x$ tending towards an isotropic Gaussian, dispelling the need for an EBM. In Line 2, we leverage the SDE generative model from (5.7) to transport the sample to $t = 1$, making use of both the proposal model vector field $\hat{v}_{t,\xi}$ and the I-EBM's score function $\nabla_y \mathcal{E}_\theta$ to form the approximation of the drift term. Finally, in Line 3, we run $T_{\text{mcmc}}$ steps of MCMC (either Langevin or HMC, cf. Algorithm 2) at a fixed (or annealed) step-size of $\eta$ with the EBM at $t = 1$ to generate the final sample.

---

[5]We note that in Albergo et al. (2023), the authors introduce a stochastic extension to the interpolant framework from Albergo & Vanden-Eijnden (2023) which elicits a quadratic loss function for the time-varying score function, similar in spirit to the canonical denoising loss used for training diffusion models. Our framework however, is agnostic to the type of bridge created between the base distribution and true data. Notably, I-EBMs and the associated I-R-NCE objective are well-defined for any type of interpolating/diffusion bridge.

### 5.2.3 Advantages

With the I-EBM framework in place, we now describe the key advantages behind the formulation compared with other state-of-the-art generative models. As we will soon show experimentally (Section 7), it is important for generative policies to be able to both generate *and* score samples efficiently. I-EBMs meet both requirements in ways that many other state-of-the-art models cannot, as we discuss below.

*Comparison to stochastic interpolants:* As a stochastic interpolant model is as its core a continuous normalizing flow, computing sample likelihoods require solving the standard log-probability ODE (cf. Section 6), which is computationally bottlenecked by vector-Jacobian products. While we do employ various tricks to speed this procedure up (as discussed in Section 6), I-EBMs completely avoid this ODE computation for likelihood computation, and only require one forward pass through the network $\mathcal{E}_\theta(x, y, 1)$.

*Comparison to diffusion models:* A similar computation issue arises when scoring samples with ODE-based diffusion models (Song et al., 2021b; Karras et al., 2022) as with interpolant models. This is because denoiser models are typically parameterized as unconstrained vector-valued functions, and hence cannot be analytically integrated. Therefore, the most straightforward way to compute log-probability is to treat a diffusion model as a normalizing flow and use the log-probability ODE, inheriting the associated computational bottlenecks. An alternative to the log-probability ODE is to use Girsanov's theorem to derive a lower bound on the log-likelihood, which can then be estimated via Monte-Carlo sampling (Song et al., 2021a). A similar sampling-based approach to likelihood computation also applies to DDPM-style diffusion models (Sohl-Dickstein et al., 2015; Ho et al., 2020), where Monte-Carlo sampling of the evidence lower bound can be used to estimate a lower bound on the log-likelihood. However, in addition to introducing an extra hyperparameter (the number of Monte-Carlo samplings used) which is necessary to tune the best trade-off between accurancy and performance, comparing ELBO scores across samples is actually quite problematic as described next.

*Comparison to conditional VAE models:* Conditional variational autoencoders (CVAE) (Doersch, 2016) have been recently proposed by several authors (Zhao et al., 2023; Gomez-Gonzalez et al., 2020; Ivanovic et al., 2021) as a model for generative policies. Unfortunately, the ELBO loss which CVAEs optimize is insufficient to support ranking samples. To see this, first recall that CVAEs utilize the following decomposition of the log-probability (Doersch, 2016):

$$\log p(y \mid x) - \mathrm{KL}(q(z \mid x, y) \parallel p(z \mid x, y)) = \mathbb{E}_{z \sim q(\cdot \mid x, y)}[\log p(y \mid x, z)] - \mathrm{KL}(q(z \mid x, y) \parallel p(z \mid x)). \qquad (5.9)$$

The RHS of (5.9) lower bounds the log-probability $\log p(y \mid x)$, and is the ELBO loss used during training; it only references the encoder $q(z \mid x, y)$, the decoder $p(y \mid x, z)$, and the prior $p(z \mid x)$, and is therefore possible to optimize over. The LHS of (5.9) contains the desired $\log p(y \mid x)$, plus an extra error term $\phi(x, y) := -\mathrm{KL}(q(z \mid x, y) \parallel p(z \mid x, y))$, which cannot be computed due to its dependence on the true posterior $p(z \mid x, y)$. From this decomposition, we see that for a given shared context $x$ and a set of samples $y_1, \ldots, y_k$, the RHS of (5.9) cannot serve as a reliable ranking score function, since the unknown error terms $\phi(x, y_i)$ for $i = 1, \ldots, k$ are in general all different and not comparable. Note that this issue persists even in the limit of infinite samples used to compute the RHS of (5.9).

*Efficient sampling from I-EBMs:* We have thus far focused on the advantages of I-EBMs over other generative models when it comes to likelihood computation. We now discuss the sampling advantages of I-EBMs. Combining the vector field and energy function within the SDE sampling step (cf. Line 2 in Algorithm 4) significantly reduces the burden of MCMC-based sampling, which can be difficult to converge when starting far from the stationary distribution of the EBM. In this respect, I-EBMs bring the advantages of learning over multiple noise resolutions/annealed sampling to EBMs.

## 6 Efficient Log-Probability Computation

In this section, we discuss efficient computation of log-probability for continuous normalizing flows. We first discuss the generic setup, and then describe where the setup is instantiated in our work.

**Setup.** Let $v_t(x, z)$ denote a vector field, and consider the following continuous flow:

$$\frac{\mathrm{d}z}{\mathrm{d}t}(t) = v_t(x, z(t)), \quad z(0) \sim p_0(\cdot \mid x). \tag{6.1}$$

For any $T > 0$, a standard computation (Grathwohl et al., 2019; Chen et al., 2018) shows that the random variable $z(T)$ has log-probability given by the following formula:

$$\log p(z(T) \mid x) = \log p_0(z(0) \mid x) - \int_0^T \mathrm{div}_z(v_t(x, z(t))) \, \mathrm{d}t. \tag{6.2}$$

Recall that for a vector field $f : \mathbb{R}^d \mapsto \mathbb{R}^d$, we have $\mathrm{div}(f) = \mathrm{tr}(\partial f) = \sum_{i=1}^d \frac{\partial f}{\partial x_i}$.

**Applications.** We make use of the computation described in (6.2) in several key places for our work. First, we use this formula to compute $\log p_\xi(y \mid x)$ inside the R-NCE objective (cf. (3.3)). Second, we use this formula to compute $\log p_\theta(y \mid x)$ for a diffusion model where the denoiser is directly parameterized (instead of indirectly via the gradient of a scalar function); for diffusion models, this log-probability calculation is necessary for ranking samples in next action selection (cf. Section 7.3). Here, we apply (6.2) by interpreting the reverse probability flow ODE as a continuous normalizing flow of the form (6.1).

**Efficient Computation.** The main bottleneck of (6.2) is in computing the integrand, which involves computing the divergence of the vector field $v_t(x, z)$ w.r.t. the flow variable $z$. Assuming the flow variable $z \in \mathbb{R}^d$, the computational complexity of a single divergence computation scales *quadratically* in $d$, i.e., $O(d^2)$. The quadratic scaling arises due to the fact that for exact divergence computation, one needs to separately materialize each column of the Jacobian, and hence $O(d)$ computation is repeated $d$ times.[6] The typical presentation of CNFs advocates solving an augmented ODE which simultaneously performs sampling and log-probability computation:

$$\frac{\mathrm{d}}{\mathrm{d}t} \begin{bmatrix} z \\ \psi \end{bmatrix}(t) = \begin{bmatrix} v_t(x, z(t)) \\ -\mathrm{div}_z(v_t(x, z(t))) \end{bmatrix}, \quad z(0) \sim p_0(\cdot \mid x), \quad \psi(0) = \log p_0(z(0) \mid x).$$

This augmented ODE suffers from one key drawback: it forces both the sample variable $z$ and the log-probability variable $\psi$ to be integrated at the same resolution. Due to the quadratic complexity of computing the divergence, this integration becomes prohibitively slow for fine time resolutions. However, high sample quality depends on having a relatively small discretization error.

We resolve this issue by using a two time-scale approach. In our implementation, we separately integrate the sample variable $z$ and the log-probability variable $\psi$. We first integrate $z$ at a fine resolution. However, instead of computing the divergence of $v$ at every integration timestep, we compute and save the divergence values at a much coarser subset of timesteps. Then, once the sample variable is finished integrating, we finish off the computation by integrating the $\psi$ variable with one-dimensional trapezoidal integration (e.g., using `numpy.trapz`). Splitting the computation in this way allows one to decouple sample quality from log-probability accuracy; empirically, we have found that using at most 64 log-probability steps suffices even for the most challenging tasks we consider. A reference implementation is provided in Appendix A, Figure 5.

## 7 Experiments

In this section, we present experimental validation of R-NCE as a competitive model for generative policies. We implement our models in the `jax` (Bradbury et al., 2018) ecosystem, using `flax` (Heek et al., 2023) for training neural networks and `diffrax` (Kidger, 2021) for numerical integration. All models are trained using the Adam (Kingma & Ba, 2015) optimizer from `optax` (Babuschkin et al., 2020).

---

[6]Randomized methods such as the standard Hutchinson trace estimator are not applicable here, since we need to be able to compare and rank log-probabilities across a small batch of samples. Generically, unless the Hutchinson trace estimator is applied $\Omega(d)$ times (thus negating the computational benefits), the variance of the estimator will overwhelm the signal needed for ranking samples; this is a consequence of the Hanson-Wright inequality (Meyer et al., 2021).

### 7.1 Models Under Evaluation and Overview of Results

Here, we collect the types of generative models which we evaluate in our experiments. Note that the more complex experiments only evaluate a subset of these models, as not all of the following models work well for high dimensional problems.

- **NF** (normalizing flow): a CNF (Grathwohl et al., 2019), where for computational efficiency the vector field parameterizing the flow ODE is learned via the stochastic interpolant framework (Albergo et al., 2023) instead of maximum likelihood; see Section 5.1. The flow ODE is integrated with Heun. During training, we sample interpolant times from the push-forward measure of the uniform distribution on $[0, 1]$ transformed with the function $t \mapsto t^{1/\alpha}$, where $\alpha$ is a hyperparameter.

- **IBC** (implicit behavior cloning): the InfoNCE (van den Oord et al., 2018) inspired IBC objective of Florence et al. (2022), defined in (4.3), for training EBMs. Langevin sampling (cf. Algorithm 2) is used to produce samples.

- **R-NCE** and **I-R-NCE** (ranking noise contrastive estimation): the learning algorithm presented in Algorithm 1 for training EBMs, and its interpolating variation, Algorithm 3, presented in Section 5.2 for training I-EBMs; both leveraging the stochastic interpolant CNF as the learnable negative sampler. Sampling is performed using either Algorithm 2 for EBM or Algorithm 4 for I-EBM.

- **Diffusion-EDM**: A standard score-based diffusion model. We base our implementation heavily off the specific parameterization described in Karras et al. (2022), including the use of the reverse probability flow ODE for sampling in lieu of the reverse SDE, which we integrate via the Heun integrator. We also treat the reverse probability flow ODE as a CNF for the purposes of log-probability computation (cf. Section 6).

- **Diffusion-EDM-$\phi$**: Same as Diffusion-EDM, except instead of directly parameterizing the denoiser function, we represent the denoiser as the gradient of a parameterized energy model (Salimans & Ho, 2021; Du et al., 2023). This allows us to use identical architectures for models as IBC, R-NCE, and I-R-NCE. More details regarding recovering the relative likelihood model from the parameterization of Karras et al. (2022) are available in Appendix B.1. We note that this model has an extra hyperparameter, $\sigma_{\text{rel}}$, which indicates the noise level at which to utilize the energy model $\phi$ for relative likelihood scores.[7]

Note for both diffusion models, for brevity we often drop the "-EDM" label when it is clear from context that we are referring to this particular parameterization of diffusion models.

**Basic Principles for Model Comparisons.** As this list represents a wide variety of different methodologies for generative modeling, some care is necessary in order to conduct a fair comparison. Here, we outline some basic principles which we utilize throughout our experiments to ensure the fairest comparisons:

(a) *Comparable parameter counts*: We keep the parameter counts between different models in similar ranges, regardless of whether or not a particular model parameterizes vector fields (NF, Diffusion) or energy models (IBC, R-NCE, I-R-NCE, Diffusion-$\phi$). Furthermore, for R-NCE and I-R-NCE, we count the total number of parameters between both the negative sampler and the energy model.

(b) *Comparable functions evaluations for sampling*: We keep the number of function evaluations made to either vector fields or energy models in similar ranges regardless of the model, so that inference times are comparable.

(c) *Avoiding excessive hyperparameter tuning*: In order to limit the scope for comparison, we omit exploring many hyperparameter settings which apply equally to all models, but may give some

---

[7]We find that, much akin to how the diffusion reverse process for sampling is typically terminated at a small but non-zero time, this is also necessary for the numerical stability of the relative likelihood computation (cf. Appendix B.1, Equation (B.1)). Furthermore, we find that the best stopping time for the latter is typically an order of magnitude higher than the former. We leave further investigation into this for future work.

marginal performance increases. A few examples of the types of optimizations we omit include periodic activation functions (Sitzmann et al., 2020), pre vs. post normalization for self-attention (Xiong et al., 2020), and exponential moving average parameter updates (Yazıcı et al., 2019; Song & Ermon, 2020). While these types of optimizations have been reported in the literature to non-trivially improve performance, since they apply equally to all models, we omit them in the interest of limiting the scope of comparison.

**Overview of Results.** We provide a brief overview of the results to be presented. First, we consider synthetic two-dimensional conditional distributions (Section 7.2) as a basic sanity check. In these two-dimensional examples, we see that all models perform comparably except for IBC, which has visually worse sample quality. The next two examples feature robotic tasks where we learn policies that predict multiple actions into the future. This multistep horizon prediction is motivated by arguments given by Chi et al. (2023, Section IV.C), namely temporal consistency, robustness to idle actions, and eliciting multimodal behaviors.

The first robotic task is a high dimensional obstacle avoidance path planning problem (Section 7.3). Here, we see again that IBC yields policies with non-trivially higher collision rates and costs than the other models. More importantly, we start to see a separation between the other methods, with R-NCE ultimately yielding both the lowest collision rate and cost. The next and final task is our most challenging benchmark: a contact-rich block pushing task (Section 7.4). For this block pushing task, we utilize the full power of the interpolating EBM framework (cf. Section 5.2). Our experiments show that I-R-NCE yields the policy with the highest final goal coverage. Thus, the key takeaway from our experimental evaluation is that training EBMs with R-NCE does indeed yield high quality generative policies which are competitive with, and even outperform, diffusion and stochastic interpolant based policies.

## 7.2 Synthetic Two-Dimensional Examples

We first evaluate R-NCE and various baselines on several conditional two-dimensional problems, **Pinwheel** and **Spiral**.

- **Pinwheel**: The context space is a uniform distribution over $\mathsf{X} = \{4, 5, 6, 7\}$, denoting the number of spokes of a planar pinwheel. We apply a 10 dimensional sinusoidal positional embedding to embed $\mathsf{X}$ into $\mathbb{R}^{10}$. The event space $\mathsf{Y} = \mathbb{R}^2$, denoting the planar coordinates of the sample. The distribution is visualized in the upper left row of Table 1, with only $x \in \{4, 5, 6\}$ visualized for clarity.

- **Spiral**: The context space is a uniform distribution over $\mathsf{X} = [400, 800]$, denoting the length of a parametric curve representing a planar spiral. This range is normalized to the interval $[-1, 1]$ before being passed into the models. The event space $\mathsf{Y} = \mathbb{R}^2$, again denoting the planar coordinates of the sample. The distribution is visualized in the upper right row of Table 1, with only $x \in \{400, 500, 600\}$ visualized for clarity.

The performance of the models listed in Section 7.1 is shown in Table 1 (samples) and Table 2 (KDE plots). The specific details of model architectures, training details, and hyperparameter values are given in Appendix B.2. For all models, we restrict the parameter count of models to not exceed $\sim$22k; for R-NCE, this number is a limit on the sum of the NF+EBM parameters. We report the Bhattacharyya coefficient (BC) between the sampling distribution and the true distribution, computed as follows. First, recall that the BC between true conditional distribution $p(y \mid x)$ and a learned conditional distribution $p_\theta(y \mid x)$ is:

$$\mathrm{BC}(p(y \mid x), p_\theta(y \mid x)) = \int_\mathsf{Y} \sqrt{p(y \mid x)p_\theta(y \mid x)}\, \mathrm{d}\mu_Y \quad \forall x \in \mathsf{X}. \tag{7.1}$$

Note that the BC lies between $[0, 1]$, with a value of one indicating that $p(y \mid x) = p_\theta(y \mid x)$. We estimate both $p(y \mid x)$ and $p_\theta(y \mid x)$ via the following procedure, which we apply for each context $x$ that we evaluate separately. First, we compute a Gaussian KDE using 8,192 samples with `scipy.stats.gaussian_kde`, invoked with the default parameters. Next, we discretize the square $[-4, 4] \times [-4, 4]$ into $256 \times 256$ grid

**Table 1:** A two-dimensional histogram plot of the samples generated by each of the models listed in Section 7.1. The samples are binned at a resolution of 64 bins per coordinate. Each column represents a different context value $x$; see Section 7.2 for the specific values for each problem. The Bhattacharyya coefficient (BC) is computed as described in Section 7.2.

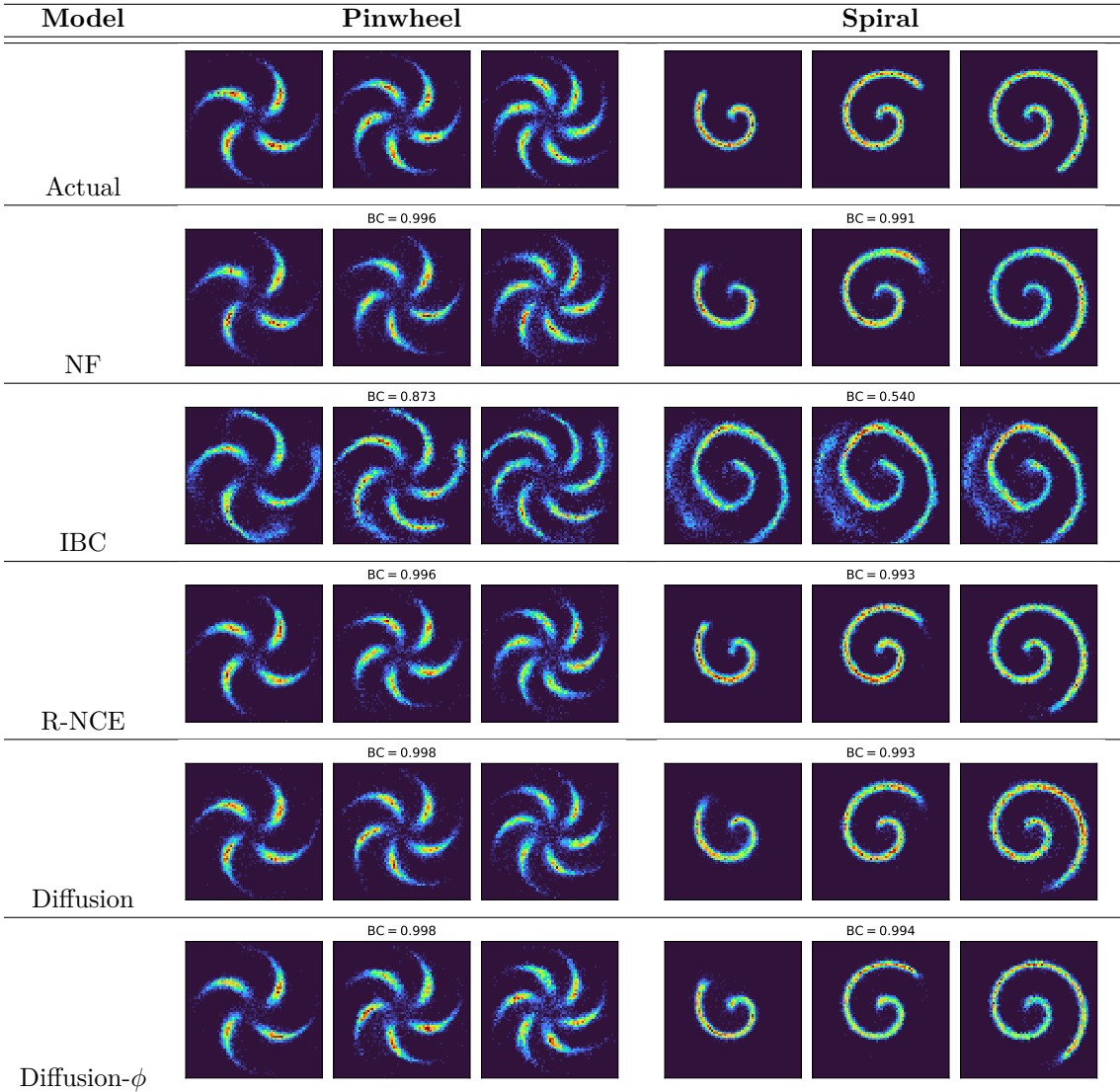

points, and compute (7.1) via numerical integration using the KDE estimates of the density. Finally, we report the minimum of the BC estimate over all $x$ values that we consider.

The main findings in Table 1 and Table 2 are that the highest quality samples are generated by the R-NCE, Diffusion, and NF models, between which the quality is indistinguishable, followed by a noticeable drop off in quality for IBC. Note that for IBC we use a uniform noise distribution to generate negative samples. While relatively simple, this example is sufficient to validate our implementations, albeit still illustrating the limitations of IBC.

### 7.3 Path Planning

We next turn to a higher dimensional problem, one of optimal path planning around a set of obstacles. We study this task due to its inherent multi-modality. The environment is a planar environment with 10 randomly sampled spherical obstacles, a random goal location, and a random starting point. The objective

**Table 2:** A Gaussian KDE estimate generated via 8,192 samples for each of the models listed in Section 7.1. The KDE is computed via `scipy.stats.gaussian_kde` with default values for the `bw_method` and `weights` parameters. Each column represents a different context value $x$; see Section 7.2 for the specific values for each problem. The Bhattacharyya coefficient (BC) is computed as described in Section 7.2.

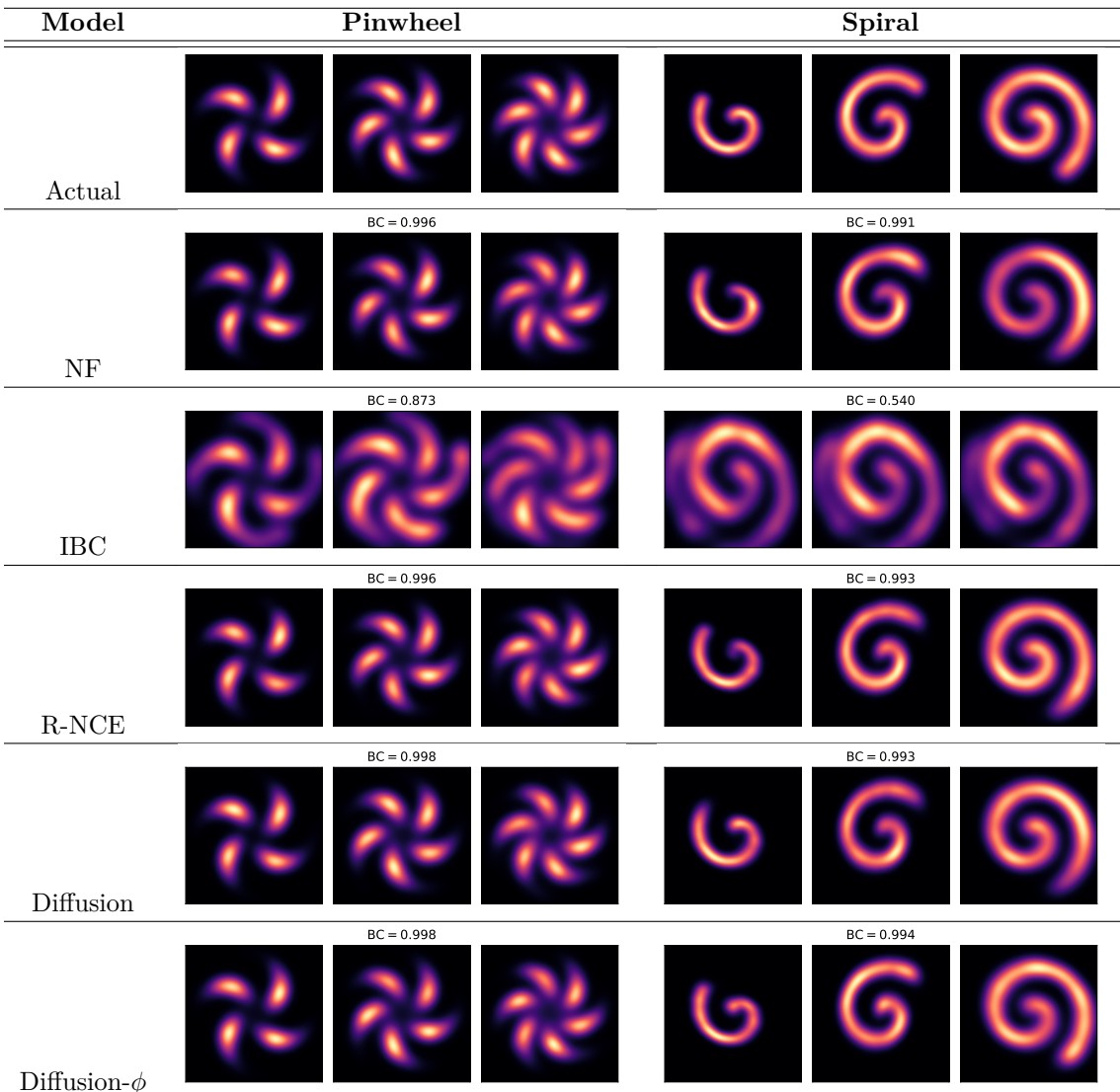

is to learn an autoregressive navigation policy which, when conditioned on the scene (i.e., obstacles and goal) and the last $N_{\text{ctx}}$ positions of the agent, models the distribution over the next $N_{\text{pred}}$ positions.

**Dataset Generation.** In order to generate the demonstration optimal trajectories, we use the stochastic Gaussian process motion planner (StochGPMP) of Urain et al. (2022b), in which we optimize samples generated from an endpoints conditioned Gaussian process (GP) prior with an obstacle avoidance cost. The GP prior captures dynamic feasibility (without considering the obstacles) and the following key properties: (i) closeness to the specified start-state with weight $\sigma_s > 0$, (ii) closeness to the specified goal-state with weight $\sigma_g > 0$, and (iii) trajectory smoothness with weight $\sigma_v > 0$; see Urain et al. (2022b); Mukadam et al. (2018); Barfoot et al. (2014) for details on the computation of the prior. Roughly speaking, given a trajectory $\{y[i]\}_{i=1}^{N}$, a start location $s$, and a goal location $g$, the primary components of the effective "cost

function" associated with the GP prior scale as:

$$\frac{1}{\sigma_s^2}\|y[1]-s\|^2 + \frac{1}{\sigma_g^2}\|y[N]-g\|^2 + \frac{1}{\sigma_v^2}\sum_{i=1}^{N-1}\|y[i+1]-y[i]\|^2.$$

In generating optimal trajectories from StochGPMP, we set the trajectory length to $N=50$. The obstacle cost $c_{\text{obs}}$ is parameterized by the obstacle set $O=\{(c_k,r_k)\}_{k=1}^{10}$, where $c_k\in\mathbb{R}^2$ and $r_k\in\mathbb{R}$ are the centers and radii of the spherical obstacles, respectively:

$$c_{\text{obs}}(\{y[i]\}_{i=1}^{N},O) = \sum_{i=1}^{N-1}\sum_{k=1}^{10}\sqrt{r_k^2 - \text{dist}^2(c_k,(y[i],y[i+1]))}\mathbf{1}\{\text{dist}(c_k,(y[i],y[i+1])) \leqslant r_k\}.$$

The contribution to the StochGPMP planner from the obstacle cost scales as $c_{\text{obs}}/\sigma_o^2$. Following the generation of global trajectories with the planner, we recursively split the trajectories into $(N_{\text{ctx}}+N_{\text{pred}})$-length snippets, and refine these snippets further with the StochGPMP planner, using the endpoints of the snippets as the "start" and "goal" locations. The collection of these snippets form the training and evaluation datasets.

**Policy and Rollouts.**   The autoregressive policy is then defined as the following conditional distribution:

$$p(y[i+1],\dots,y[i+N_{\text{pred}}] \mid y[i],\dots,y[i-N_{\text{ctx}}+1],g,O).$$

For simplicity, we assume perfect state observation and perform this modeling in position space, i.e., $y[i]\in\mathbb{R}^2$ denotes the planar coordinates of the trajectory at time $i$. Furthermore, we assume perfect knowledge of the goal $g$ and obstacles $O$. We leave relaxations of these assumptions to future work. We set $N_{\text{ctx}}=3$ and $N_{\text{pred}}=10$, yielding a 20-dimensional event space. In order to generate a full length trajectory, we autoregressively sample and rollout the policy. Given a prediction horizon of $N_{\text{pred}}=10$ and a total episode length of $N=50$ steps, we sample the policy 5 times over the course of an episode. In order to encourage more optimal trajectories from sampling, for each context/policy-step we sample 48 future prediction snippets, and take the one with the highest (relative) log-likelihood. For R-NCE and Diffusion-$\phi$, this computation is a straightforward forward pass through the energy model. On the other hand, for NF and Diffusion, this computation relies on the differential form of the evolution of log-probability described in Section 6.

**Evaluation.**   For evaluation, we compute the cost of the entire trajectory resulting from the autoregressive rollout as the summation of three terms: (i) goal-reaching, (ii) smoothness, and (iii) obstacle collision cost. To appropriately scale these costs in a manner consistent with the demonstration data, the evaluation cost is modeled after the primary cost terms in the GP cost plus the obstacle avoidance cost, used to generate the data:

$$\text{Obj}(y,g,O) = \frac{1}{\bar\sigma_g^2}\|y[N]-g\|^2 + \frac{1}{\bar\sigma_o^2}c_{\text{obs}}(\{y[i]\}_{i=1}^{N},O) + \frac{1}{\bar\sigma_v^2}\sum_{i=1}^{N-1}\|y[i]-y[i-1]\|^2. \tag{7.2}$$

Here, $\bar\sigma_g$, $\bar\sigma_o$, and $\bar\sigma_v$ are derived by normalizing the corresponding weights from the StochGPMP planner.

**Architecture and Training.**   For both energy models and vector fields, we design architectures based on encoder-only transformers (Vaswani et al., 2017). Specifically, we lift all conditioning and event inputs to a common embedding space, apply a positional embedding to the lifted tokens, pass the tokens through multiple transformer encoder layers, and conclude with a final dense accumulation MLP. However, as previously discussed in Section 4.3.1, for computational reasons we make the vector field parameterizing the proposal model for R-NCE a dense MLP, which is significantly simpler compared to the EBM. Indeed, the proposal CNF on its own is insufficient to solve this task, but is powerful enough as a negative sampler. More specific architecture, training, and hyperparameter tuning details are found in Appendix B.3. Note that for IBC, we do not use a fixed proposal distribution, as doing so results in poor quality energy models. Instead, we use the same NF proposal distribution as used for R-NCE, trained jointly as outlined in Algorithm 1.

For all models considered, we limit the parameter count to a maximum of ∼500k (for IBC/R-NCE, this is a limit on the EBM+NF parameters). Finally, for all models, during training, we perturbed the context and

prediction snippet, i.e., $\{y[i - N_{\text{ctx}} + 1], \ldots, y[i + N_{\text{pred}}]\}$, using Gaussian noise with variance $\sigma_{\text{pert}}^2$, where $\sigma_{\text{pert}}$ is annealed starting from 0.01 down to 0.005 over half the training steps, and held fixed thereafter. We found that this technique significantly boosted the performance of *all* models, and provide some intuition in the discussion of the results.

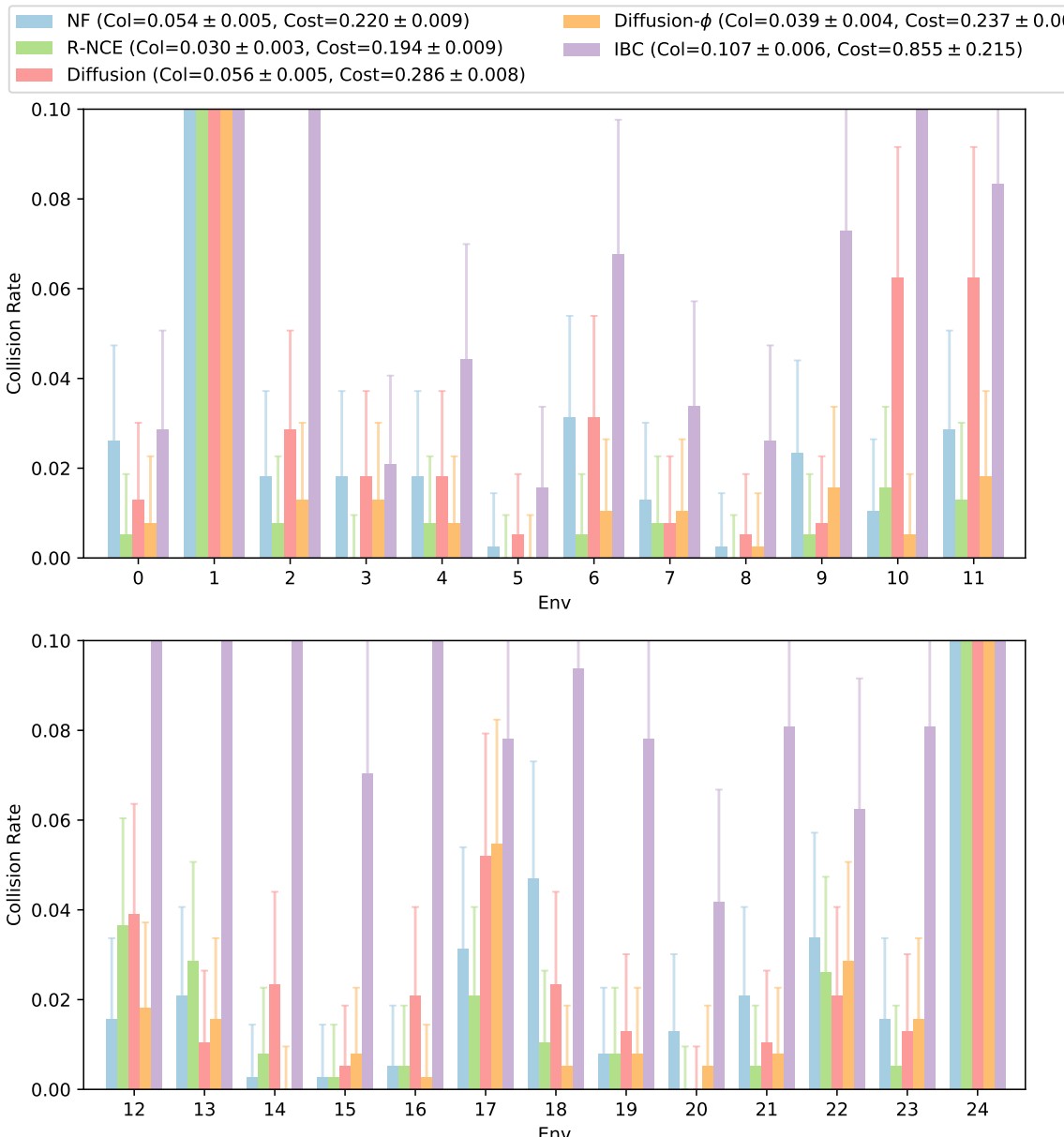

**Figure 4:** A bar plot comparing the collision rates amongst different models, for each of the 25 test environments. For each environment, a (environment conditioned) 95% confidence interval for the collision rate is computed via the Clopper-Pearson method and shown via an error bar. Furthermore, a symmetric 95% confidence interval for both the collision rate and the cost is shown for each model in the legend, computed via the standard normal approximation. Each color corresponds to a different model, with the color legend shown in the top plot. Note that for several of the environments, the collision rate extends beyond the limits of the displayed plots. For all models, Environments 1 and 24 are particularly challenging; we illustrate the qualitative behavior of this failure case in Table 3.

**Results.** We focus our evaluation on two metrics: (a) collision rate, and (b) the objective cost (7.2). Our main finding is that R-NCE yields the sampler with both the lowest collision rate and the lowest obstacle

**Table 3:** Sampled trajectories for three separate path planning test environments (out of 25). The specific environments are shown to highlight multi-modal solutions. Each environment contains 384 trajectories sampled autoregressively, 10 obstacles, and one goal location indicated by the orange marker. The collision rate and cost values are shown for each environment, along with a symmetric 95% confidence interval computed via a standard normal approximation. The "Env 1" column highlights the failure cases for the best performing models (cf. Figure 4), where collisions arise due to the models generating trajectories which attempt to pass through narrow passageways.

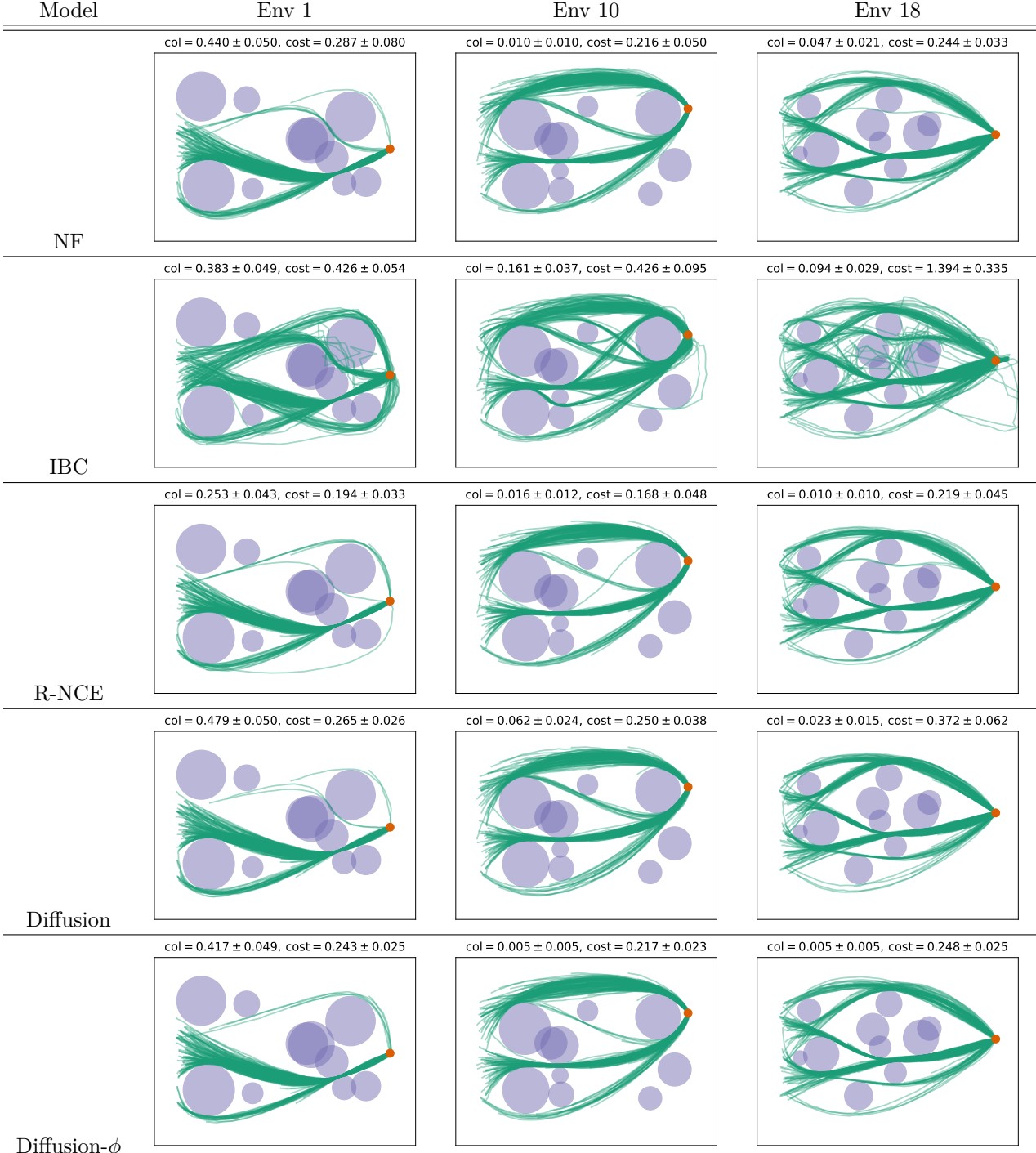

cost. Figure 4 provides a quantitative evaluation, showing the collision rates for each model on all 25 test environments. Furthermore, in Table 3, we show the performance of the various models on three test environments. Observe that in Figure 4, there are two environments (specifically Environments 1 and 24) for which the collision rate exceeds 10% on every model. Example trajectories from Environment 1 are shown in Table 3 to illustrate this failure scenario (note that Environment 24 exhibits similar behavior). Here, the failures arise from the policy attempting to aggressively cut through a very narrow slot between two obstacles.

For R-NCE, we additionally experimented with leveraging a pre-trained proposal model; all other relevant training hyperparameters were held the same as joint training, other than holding $\sigma_{\text{pert}}$ fixed at 0.005. The resulting model's performance yielded an average collision rate of 5.8% and cost 0.24, notably worse that the joint training equivalent. We hypothesize that data noising improves the conditioning of the generative learning problem. Indeed, the trajectories in the training dataset arguably live on a much smaller sub-manifold of the ambient event space. By annealing the perturbation noise, we are able to *guide* the convergence of the learned distributions towards this sub-manifold. This is not possible, however, if the proposal model is pre-trained and the perturbation noise is annealed during EBM training.

In Table 4, we additionally ablated the number of negative samples $K$ within the R-NCE objective. While higher $K$ yields better overall closed-loop performance, likely due to better generative models, the drop-off in performance is relatively mild for this problem.

**Table 4:** Ablation over the number of negative samples $K$ within the R-NCE objective. Increasing $K$ yields higher quality policies w.r.t. closed-loop collision rate and cost. However, the drop-off in performance is mild, suggesting some inherent robustness in the learning problem.

| $K$ | Col | Cost |
|---|---|---|
| 7 | $0.061 \pm 0.005$ | $0.20 \pm 0.009$ |
| 31 | $0.052 \pm 0.004$ | $0.212 \pm 0.01$ |
| 63 | $0.03 \pm 0.003$ | $0.194 \pm 0.009$ |

Finally, in Table 5, we show that generating multiple ($\ell = 48$) samples per step and selecting the one with highest log-probability has a non-trivial effect on the quality of the resulting trajectories, thus demonstrating the importance of ranking samplings.

**Table 5:** A study to illustrate the necessity of generating multiple samples per timestep, and selecting the sample with the highest score. Here, we let the variable $\ell$ denote the number of samples which are generated at every timestep. Symmetric 95% confidence interval for both the collision rate and the cost are computed via the standard normal approximation. Uniformly across all models, both the collision rate and trajectory cost rise significantly going from $\ell = 48$ samples per step down to $\ell = 1$. Recall that for Diffusion, the log-probability is computed by treating the model as a CNF (cf. Section 6).

| Model | Col ($\ell = 48$) | Col ($\ell = 1$) | Cost ($\ell = 48$) | Cost ($\ell = 1$) |
|---|---|---|---|---|
| NF | $0.054 \pm 0.005$ | $0.101 \pm 0.006$ | $0.220 \pm 0.009$ | $0.500 \pm 0.016$ |
| R-NCE | $0.030 \pm 0.003$ | $0.116 \pm 0.006$ | $0.194 \pm 0.009$ | $2.287 \pm 1.089$ |
| Diffusion | $0.056 \pm 0.005$ | $0.099 \pm 0.006$ | $0.286 \pm 0.008$ | $0.569 \pm 0.017$ |
| Diffusion-$\phi$ | $0.039 \pm 0.004$ | $0.087 \pm 0.006$ | $0.237 \pm 0.006$ | $0.489 \pm 0.015$ |
| IBC | $0.107 \pm 0.006$ | $0.203 \pm 0.008$ | $0.855 \pm 0.215$ | $8.734 \pm 1.677$ |

## 7.4 Contact-Rich Block Pushing (Push-T)

Our final task features contact-rich multi-modal planar manipulation which involves using a circular end effector to push a T-shaped block into a goal configuration (Chi et al., 2023; Florence et al., 2022). In this environment, the initial pose of the T-shaped block is randomized. The agent receives both an RGB image observation of the environment (which also contains a rendering of the target pose) and its current end effector position, and is tasked with outputting a sequence of position coordinates for the end effector. The

target position coordinates are then tracked via a PD controller. This task is the most challenging of the tasks we study, due to (a) contact-rich behavior and (b) reliance on visual feedback. Indeed, we find that successfully solving this task with R-NCE requires the I-R-NCE machinery introduced in Section 5.2.

Our specific simulation environment and training data comes from Chi et al. (2023), which uses the `pygame`[8] physics engine for contact simulation, and provides a dataset of 136 expert human teleoperated demonstrations.

**Policy and Rollouts.** Similar to path planning (cf. Section 7.3), we use an autoregressive policy to perform rollouts:

$$p(y[i+1], \ldots, y[i+N_{\text{pred}}] \mid y[i], o[i], \ldots, y[i-N_{\text{ctx}}+1], o[i-N_{\text{ctx}}-1]). \tag{7.3}$$

Here, $y[i]$ denotes the end effector position at timestep $i$, and $o[i]$ denotes the corresponding RGB image observation. Following Chi et al. (2023), we set $N_{\text{pred}} = 16$ and $N_{\text{ctx}} = 2$, yielding a 32-dimensional event space. Furthermore, during policy execution, even though we predict $N_{\text{pred}} = 16$ positions into the future, we only play the first 8 predictions in open loop, followed by replanning using (7.3); Chi et al. (2023, Figure 6) showed that this ratio of prediction to action horizon yielded the optimal performance on this task. As with the path planning example, for each policy step, we sample 8 predictions and execute the sequence with the highest (relative) log-likelihood.

**Evaluation.** Each rollout is scored with the following protocol. At each timestep $i$, the score of the current configuration is determined by first computing the ratio $r[i]$ of the area of the intersection between the current block pose and the target pose to the area of the block. The score $s[i]$ is then set to $s[i] = \min(r[i]/0.95, 1)$. The episode terminates when either (a) the score $s[i] = 1$, or (b) 200 timesteps have elapsed. The final score assigned to the episode is the maximum score over all the episode timesteps. Our final evaluation is done by sampling 256 random seeds (i.e., randomized initial configurations), and rolling out each policy 32 times within each environment (with different policy sampling randomness seeds).

**Architecture and Training.** We design two-stage architectures, which first encode the visual observations $\{o[i], \ldots, o[i-N_{\text{ctx}}+1]\}$ into latent representations via a convolutional ResNet, with coordinates appended to the channel dimension of the input (Liu et al., 2018), and spatial softmax layers at the end (Levine et al., 2016). To further improve the parameter efficiency of R-NCE and I-R-NCE, both the EBM and NF models *share* this visual encoder. Updates to the encoder are then allowed to be propagated by either the EBM or the NF optimization step through an appropriate stop-gradient operator. For the EBM, the spatial features are flattened and combined with both the agent positions and time index $t$ (the time index of the generative model, not the trajectory timestep index $i$), and passed through encoder-only transformers similar to the architectures used in path planning (cf. Section 7.3). For the interpolant NF which forms the negative sampler for I-R-NCE, special care is needed to design an architecture which is simultaneously expressive and computationally efficient; a dense MLP as used in path planning for the proposal distribution led to posterior collapse during training (cf. Section 4.3). We detail our design in Appendix B.4, in addition to various training and hyperparameter tuning details. For all models, we limit the parameter count to ∼3.3M (for I-R-NCE, this limit applies to the sum of the NF+EBM parameters).[9] Finally, similar to the training procedure for path planning, we employed the same annealed data perturbation schedule to guide training, with noise applied only to the sequence of end effector positions.

**Results.** We show our results in Table 6 and Table 7. Note that we do not evaluate IBC here due to its previous poor performance both in path planning, and poor performance shown for this task in Chi et al. (2023). Table 6 contains, for each model, visualizations of policy trajectories in four different evaluation environments. In Table 7, we show the final evaluation score achieved by each of the models; we additionally include results for I-R-NCE paired with the simpler two-stage sampling method from Algorithm 2, and the simpler non-interpolating R-NCE (Algorithm 1). We note that the best performance is achieved by

---

[8]Package website: `https://github.com/pygame/pygame`
[9]We find that models which are several orders of magnitude smaller than the models used in Chi et al. (2023) suffice.

**Table 6:** A visualization of the policy trajectories of each model in four different initial conditions. The trajectory is color coded using time, with the beginning of the trajectory represented using the lightest shade ▨, and the end of the trajectory with the darkest shade ■. The desired goal configuration is show in the green color ▨. Above each environment, the score assigned to the rollout is shown (see the evaluation paragraph in Section 7.4 for the definition of the score).

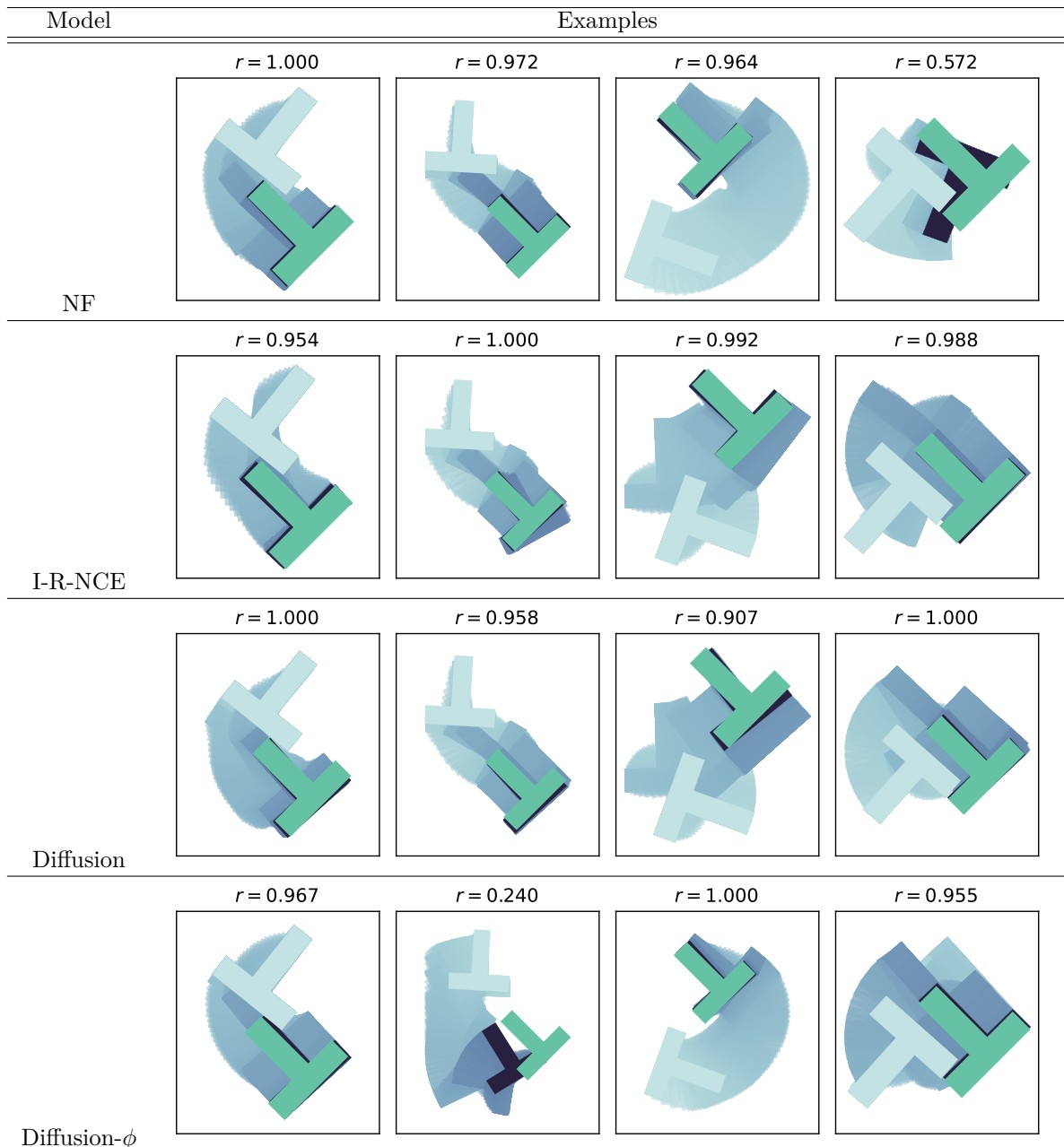

I-R-NCE, paired with the three-stage sampling method from Algorithm 4. This is closely followed by I-R-NCE with the two-stage sampling method from Algorithm 2; both I-R-NCE results notably outperform non-interpolating R-NCE. This suggests that training via I-R-NCE yields significantly superior models, even when the EBM is only used at $t = 1$ during sampling. A reasonable hypothesis would be that training via I-R-NCE yields *multi-scale variance reduction* whereby the shared network $\mathcal{E}_\theta$ benefits from learning across all times within $(0, 1]$, similar to the benefits attributed to time-varying score modeling. We anticipate that the gap between the two- and three-stage sampling algorithms should widen with even more challenging

distributions. Meanwhile, the non-interpolating variant of R-NCE does not inherit any multi-scale learning benefits. Finally, as with path planning, I-R-NCE (with or without the SDE sampling stage) outperforms the NF, Diffusion, and Diffusion-$\phi$ models.

**Table 7:** The final evaluation score across 256 randomly sampled initial configurations, and 32 rollouts within each environment configuration. The reported score is a scalar between $[0, 1]$ which describes the maximum coverage of the target achieved by the policy; the precise definition is given in the evaluation paragraph of Section 7.4. A 95% symmetric confidence interval computed via a normal approximation is shown for each score. Here, we see that the model trained with I-R-NCE outperforms the other models. The difference between I-R-NCE and I-R-NCE (Alg. 2) corresponds to the use of Algorithm 4 for the former and Algorithm 2 for the latter during sampling.

| NF | R-NCE | I-R-NCE (Alg. 2) | I-R-NCE | Diffusion | Diffusion-$\phi$ |
|---|---|---|---|---|---|
| $0.866 \pm 0.006$ | $0.824 \pm 0.006$ | $0.880 \pm 0.005$ | $0.884 \pm 0.005$ | $0.864 \pm 0.006$ | $0.860 \pm 0.006$ |

## 8 Conclusion

We showed that in the context of policy representation, energy-based models are a competitive alternative to other state-of-the-art generative models such as diffusion models and stochastic interpolants. This was done through several key ideas: (i) optimizing the ranking NCE objective in lieu of the InfoNCE objective proposed in the IBC work (Florence et al., 2022), (ii) employing a learnable negative sampler which is jointly, but non-adversarially, trained with the energy model, and (iii) training a family of energy models indexed by a scale variable to learn a stochastic process that forms a continuous bridge between the data distribution and latent noise.

This work naturally raises several broader points for further investigation. First, while we focused on theoretical foundations in addition to a comprehensive evaluation of R-NCE trained EBMs compared with other generative models, in future work we plan to deploy these EBMs on real hardware platforms. Second, this work investigated the use of R-NCE for training EBMs in the supervised setting, i.e., behavioral cloning. A promising future direction would be to adapt our R-NCE algorithm and insights for training EBM policies via reinforcement learning (Levine, 2018). Finally, despite our motivations being targeted towards robotic applications, many of our technical contributions actually apply to generative modeling more broadly. Another direction for future research is to apply our techniques to other domains such as text-to-image generation, super-resolution, and inverse problems in medical imaging, and see if the benefits transfer over to other domains.

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

# Appendix

## Table of Contents

## A    Reference Two Time-Scale Implementation of Joint Sample and Log-Probability Computation

Figure 5 provides a simple reference implementation of the two time-scale approach to jointly sampling and computing log-probability described in Section 6.

```python
from typing import Callable

import jax
import jax.numpy as jnp
import jax.scipy as jsp
import diffrax as dfx

Array, KeyArray = jax.Array, jax.Array
Time, Context, Event = float, Array, Array

VectorField = Callable[[Time, Event, Context], Event]

EVENT_DIM = ...

def latent_sample_and_log_prob(key: KeyArray) -> tuple[Event, float]:
    # Isotropic Gaussian base distribution for simplicity.
    z0 = jax.random.normal(key, shape=(EVENT_DIM,))
    return z0, jnp.sum(jsp.stats.norm.logpdf(z0))

def sample_and_log_prob(
    vf: VectorField, key: KeyArray, ctx: Context, ode_ts: Array, lp_ts: Array
) -> tuple[Event, float]:
    def d_dt_psi(t, z, x):
        fn = lambda z: vf(t, z, x)
        _, vjp_fn = jax.vjp(fn, z)
        (dfdz,) = jax.vmap(vjp_fn)(jnp.eye(EVENT_DIM))
        return -jnp.trace(dfdz)
    traj_subsaveat = dfx.SubSaveAt(ts=lp_ts, fn=d_dt_psi)

    z0, psi0 = latent_sample_and_log_prob(key)
    (z,), d_dt_psis = dfx.diffeqsolve(
        dfx.ODETerm(vf), solver=dfx.Heun(), t0=ode_ts[0], t1=ode_ts[-1],
        dt0=None, y0=z0, args=ctx, stepsize_controller=dfx.StepTo(ode_ts),
        saveat=dfx.SaveAt(subs=[dfx.SubSaveAt(t1=True), traj_subsaveat]))
    return z, psi0 + jnp.trapz(d_dt_psis, lp_ts)
```

**Figure 5:** A reference implementation of the two time-scale approach described in Section 6 for jointly computing samples and log-probabilities from a continuous normalizing flow.

## B    Experimental Details

### B.1    Recovering the Energy Model in Diffusion-$\phi$

The score function $\nabla \log p(y \mid x, t)$ in the Diffusion-EDM parameterization (Karras et al., 2022) is not represented directly by a neural network, but rather indirectly through a sequence of affine transformations. In order to recover (up to a constant) the corresponding function $\phi(x, y, t) = \log p(y \mid x, t)$, we need to

integrate the score across the affine transformations. Specifically, letting $f(x, y, t)$ denote the raw vector-valued network, Diffusion-EDM parameterizes the score as (Karras et al., 2022, Eq. 3 and Eq. 7):

$$\nabla \log p(y \mid x, t) = (d(x, y, \sigma(t)) - y)/\sigma(t)^2,$$
$$d(x, y, t) = c_{\text{skip}}(t)y + c_{\text{out}}(t)f(c_{\text{in}}(t)y, c_{\text{noise}}(t)).$$

Now, if the network $f(x, y, t)$ has special gradient structure $f(x, y, t) = \nabla_y \phi(x, y, t)$, then a straightforward computation yields:

$$\log p(y \mid x, t) = r(x, y, t) + \text{const}(x, t),$$
$$r(x, y, t) = \frac{1}{2t^2}(c_{\text{skip}}(t) - 1)\|y\|^2 + \frac{c_{\text{out}}(t)}{t^2 c_{\text{in}}(t)}\phi(x, c_{\text{in}}(t)y, c_{\text{noise}}(t)). \tag{B.1}$$

where $\text{const}(x, t)$ is a function that is unknown, but only depends on $x$ and $t$. Hence, for a fixed $x \in \mathsf{X}$ and events $\{y_i\} \subset \mathsf{Y}$, we can compute the *relative* likelihoods $\{r(x, y_t, \varepsilon)\}$, where $\varepsilon$ is a value close to zero (but not equal to zero as (B.1) becomes degenerate at $t = 0$). These relative likelihood values can be used to compare the likelihood of samples for a given context, and can be hence used to perform fast action selection.

## B.2 Synthetic Two-Dimensional Examples

For each problem (i.e., pinwheel and spiral), the training data consists of 50,000 samples from the joint distribution $\mathsf{P}_{X,Y}$.

**Architectures and EBM Sampling.** For all energy models (IBC, Diffusion-$\phi$, and R-NCE-EBM, as we do not use the interpolating variant of R-NCE for this benchmark), we model the energy function $\mathcal{E}(x, y)$ by first concatenating $(x, y)$ and then passing through 8 dense layers with residual connections. Similarly for the NF vector field and Diffusion denoiser, which is also a vector-valued map $v_t(x, y)$, we first embed the time argument through 2 fully connected MLP layers of width 10, and then pass the concatenated arguments $(x, y, \text{MLP}(t))$ into 8 dense layers of width 64 with residual connections. Finally for parameter efficiency reasons, the R-NCE-NF vector field first concatenates $z = (x, y)$ and then passes the inputs $(z, t)$ into 2 ConcatSquash[10] layers. Both here and throughout our experiments, we utilize the `swish` activation function (Ramachandran et al., 2017). EBM sampling (i.e., for IBC and R-NCE) is done using Algorithm 2 with Langevin MCMC. Table 8 and Table 9 contains the hyperparameters we search over.

**Table 8:** List of hyperparameters for the EBMs. LR denotes learning rate, WD denotes weight decay, $T_{\text{mcmc}}$ refers to the number of Langevin steps taken, $\eta$ refers to the Langevin step size, and $K$ denotes the number of negative examples. Note that $K$ is odd here since it was more computationally efficient to sample $K+1$ negative samples and replace one with the positive sample. Note that all models considered have at most ∼11k NF+EBM parameters in total.

| Model | Width | LR | WD | $T_{\text{mcmc}}$ | $\eta$ | $K$ |
|---|---|---|---|---|---|---|
| IBC | $\{64, 128\}$ | $\{10^{-3}, 5 \cdot 10^{-4}\}$ | $\{0, 10^{-4}\}$ | $\{250, 500, 750, 1000\}$ | $\{10^{-3}, 5 \cdot 10^{-4}\}$ | $\{9, 63, 127, 255\}$ |
| R-NCE-EBM | 64 | $\{10^{-3}, 5 \cdot 10^{-4}\}$ | $\{0, 10^{-4}\}$ | $\{500, 1000\}$ | $\{10^{-3}, 5 \cdot 10^{-4}\}$ | 9 |
| R-NCE-NF | $\{32, 64\}$ | $\{10^{-3}, 5 \cdot 10^{-4}\}$ | $\{0, 10^{-4}\}$ | | | |

**Training.** We use a constant learning rate for all models, a batch size of 128 examples, and train the models for 20,000 total batch steps. For R-NCE, 20,000 R-NCE batch steps are taken, but the negative sampler varies between 20,000 or 50,000 interpolant loss batch steps depending on the specific hyperparameter setting. In particular, we tune both the number of negative sampler updates ($T_{\text{samp}}$ in Algorithm 1, Line 5) in addition to the R-NCE updates ($T_{\text{rnce}}$ in Algorithm 1, Line 9) in the main loop. Specifically, recalling that $T_{\text{outer}}$ denotes the total number of outer loop iterations (Algorithm 1, Line 4), we search over the grid:

$$\{(T_{\text{outer}}=4000, T_{\text{samp}}=5, T_{\text{rnce}}=5), (T_{\text{outer}}=10000, T_{\text{samp}}=5, T_{\text{rnce}}=2)\}.$$

---

[10]https://docs.kidger.site/diffrax/examples/continuous_normalising_flow/

**Table 9:** List of hyperparameters for diffusion and NF models. The parameter $\alpha$ is described in Section 7.1, and the remaining parameters $\sigma_{\text{data}}$, $\sigma_{\text{max}}$, $\sigma_{\text{min}}$, and $\rho$ refer to various Diffusion-EDM parameters from Karras et al. (2022, Table 1). All models considered have at most $\sim$22k parameters. For sampling, all models use 768 Heun steps (1,024 total function evaluations).

| Model | Width | LR | WD | $\sigma_{\text{data}}$ | $\sigma_{\text{max}}$ | $\sigma_{\text{min}}$ | $\rho$ | $\alpha$ |
|---|---|---|---|---|---|---|---|---|
| Diffusion Diffusion-$\phi$ | $\{64, 128\}$ | $\{10^{-3}, 5 \cdot 10^{-4}\}$ | $\{0, 10^{-4}\}$ | $\{0.5, 1\}$ | $\{60, 80\}$ | $0.002$ | $\{7, 9\}$ | - |
| NF | | | | - | - | - | - | $\{2, 3, 4\}$ |

## B.3 Path Planning

**Architectures and EBM Sampling.** As discussed in Section 7.3, we train architectures based on encoder-only transformers. Here, we give a more detailed description. The inputs to our energy models and vector fields are interpolation/noise-scale time[11] $t \in \mathbb{R}$, obstacles $\mathbf{o} = \mathbb{R}^{10 \times 3}$, a goal location $g \in \mathbb{R}^2$, previous positions $\boldsymbol{\tau}_- = (y[i - N_{\text{ctx}} + 1], \ldots, y[i]) \in \mathbb{R}^{N_{\text{ctx}} \times 2}$, and future positions $\boldsymbol{\tau}_+ = (y[i + 1], \ldots, y[i + N_{\text{pred}}]) \in \mathbb{R}^{N_{\text{pred}} \times 2}$. We fix an embedding dimension $d_{\text{emb}}$, and denote the combined past and future snippets as $\boldsymbol{\tau} = (\boldsymbol{\tau}_-, \boldsymbol{\tau}_+)$. The architecture is depicted in Figure 6, where $d_{\text{out}}$ is equal to the dimensionality of the event space ($2N_{\text{pred}}$ for path planning) for vector fields and is equal to one for energy models.

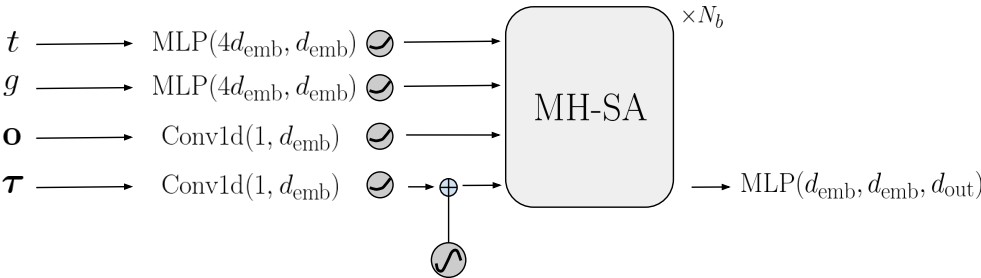

**Figure 6:** The encoder-only transformer architecture we utilize for path planning models. Here, ⬯ refers to the `swish` activation function (which is also the activation function used inside the MLP blocks), ⬯ is the standard sinusoidal positional embedding, and the first argument to Conv1d denotes the filter size. MH-SA denotes a standard Multi-Head Self-Attention encoder block (Vaswani et al., 2017); we use 4 self-attention heads in each block. The $\boldsymbol{\tau}$ tokens after the terminal block are flattened and passed through the output MLP.

In addition, we also consider architectures which utilize a convolutional "prior" which operates only on the trajectory $\boldsymbol{\tau}$ tokens. Namely, after $\boldsymbol{\tau}$ is embedded into $\mathbb{R}^{d_{\text{emb}}}$ and positionally encoded, we pass the resulting trajectory tokens through the following two blocks:

$$\text{Conv1d}(3, d_{\text{emb}}) \to \text{GroupNorm}(8) \to \sigma \to \text{Dense}(d_{\text{emb}}) \to \sigma \qquad \text{(First block)} \qquad \text{(B.2a)}$$

$$\text{Conv1d}(3, d_{\text{emb}}) \to \text{GroupNorm}(8) \to \sigma \to \text{Dense}(d_{\text{out}}) \to \sigma \qquad \text{(Second block).} \qquad \text{(B.2b)}$$

The output of the second block is then summed along the temporal axis, to produce the output of the prior. This is then added to the output of the final MLP, shown in Figure 6.

Furthermore, we remark that for computational and parameter efficiency, both IBC-NF and R-NCE-NF do not use the transformer encoder architecture described in Figure 6. Instead, all obstacle, goal, and trajectory inputs are concatenated together and fed into several ConcatSquash layers. While this produces a proposal distribution which is nowhere near optimal, it turns out to be a sufficiently powerful negative sampler for this problem. Sampling during inference from the EBM models is done using Algorithm 2 with Langevin sampling for the MCMC stage.

**Training.** We use a train batch size of 128. For both IBC and R-NCE, we use $T_{\text{outer}} = 250{,}000$ main loop steps; each loop consists of either $T_{\text{samp}} \in \{2, 4\}$ negative sampler batch steps, and 2 IBC/R-NCE updates

---

[11] For the IBC-EBM and R-NCE-EBM models, $t$ is fixed to a constant value of 1 as we do not use the I-EBM formulation here.

**Table 10:** List of hyperparameters for the IBC and R-NCE path planning models. Width denotes the width of the ConcatSquash layers, $d_{\text{emb}}$ and $N_{\text{b}}$ are described in Figure 6, $T_{\text{mcmc}}$ refers to the number of Langevin steps taken, and $\eta$ refers to the Langevin step size. The number of ConcatSquash layers for IBC-NF and R-NCE-NF is set equal to the number of self-attention blocks $N_{\text{b}}$ within the EBM; the constant $\alpha$ for the proposal NFs was fixed at 2. Note that in our hyperparameter sweeps, we exclude combinations of Width and $N_{\text{b}}$ which yield total (NF + EBM) parameter accounts exceeding $\sim$500k.

| Model | Width | $d_{\text{emb}}$ | $N_{\text{b}}$ | LR | WD | $T_{\text{mcmc}}$ | $\eta$ |
|---|---|---|---|---|---|---|---|
| IBC-EBM | - | 64 | $\{3, 4, 5\}$ | $5 \cdot 10^{-4}$ | $\{0, 10^{-4}\}$ | $\{500, 750\}$ | $10^{-4}$ |
| IBC-NF | $\{64, 128, 256\}$ | - | - | $10^{-3}$ | | | |
| R-NCE-EBM | - | 64 | $\{3, 4, 5\}$ | $5 \cdot 10^{-4}$ | $\{0, 10^{-4}\}$ | $\{500, 750\}$ | $10^{-4}$ |
| R-NCE-NF | $\{64, 128, 256\}$ | - | - | $10^{-3}$ | | | |

steps. The IBC-NF/R-NCE-NF optimizers use a cosine decay learning rate without warmup, whereas the IBC-EBM/R-NCE-EBM optimizers use a warmup cosine decay rate with 5,000 warmup steps. For both IBC and R-NCE, we use a fixed $K = 63$ negative samples. More hyperparameters are listed in Table 10. On the other hand, for Diffusion, Diffusion-$\phi$, and NF, we train our models for 500,000 batch steps and use a warmup cosine decay learning rate schedule,[12] with 5,000 warmup steps. The remaining hyperparameters are listed in Table 11.

**Table 11:** List of hyperparameters for diffusion and NF path planning models. Note that in our hyperparameter sweeps, we exclude combinations of $d_{\text{emb}}$ and $N_{\text{b}}$ which yield parameter accounts exceeding $\sim$500k. For sampling, all models use 375 Heun steps (750 total function evaluations), and 64 log-probability ODE steps (cf. Section 6).

| Model | $d_{\text{emb}}$ | $N_{\text{b}}$ | LR | WD | $\sigma_{\text{data}}$ | $\sigma_{\text{max}}$ | $\sigma_{\text{min}}$ | $\sigma_{\text{rel}}$ | $\rho$ |
|---|---|---|---|---|---|---|---|---|---|
| Diffusion | $\{64, 80\}$ | $\{3, 4, 5\}$ | $\{10^{-3}, .0005\}$ | $\{0, 10^{-4}\}$ | $\{.5, 1\}$ | $\{60, 80\}$ | .002 | - | $\{7, 9\}$ |
| Diffusion-$\phi$ | | | | | | | | $\{.01, .02, .03\}$ | |

| Model | $d_{\text{emb}}$ | $N_{\text{b}}$ | LR | WD | $\alpha$ |
|---|---|---|---|---|---|
| NF | $\{64, 80\}$ | $\{3, 4, 5\}$ | $\{10^{-3}, .0005\}$ | $\{0, 10^{-4}\}$ | $\{2, 3\}$ |

## B.4 Push-T

**Architectures and EBM Sampling.** For all models, we use a similar encoder-only transformer architecture as for path planning. The inputs to the energy models and vector fields consist of: interpolation/noise-scale time $t \in \mathbb{R}$,[13] past image observations $\mathbf{o} \in \mathbb{R}^{N_{\text{ctx}} \times 96 \times 96 \times 3}$, past agent positions $\boldsymbol{\tau}_- = (y[i - N_{\text{ctx}} + 1], \ldots, y[i]) \in \mathbb{R}^{N_{\text{ctx}} \times 2}$, and future positions $\boldsymbol{\tau}_+ = (y[i + 1], \ldots, y[i + N_{\text{pred}}]) \in \mathbb{R}^{N_{\text{pred}} \times 2}$.

The image observations are first featurized into a sequence of embeddings $\mathbb{R}^{N_{\text{ctx}} \times d_o}$ via a convolutional ResNet, with coordinates appended to the channel dimension of the images (Liu et al., 2018), and spatial softmax layers at the end (Levine et al., 2016). We use a 4-layer ResNet, with width $d_{\text{res}}$, and 4 spatial-softmax output filters, giving $d_o = 8$. The rest of the architecture resembles the self-attention architecture from Appendix B.3 (see Figure 7 for completeness). As before, we choose an embedding dimension $d_{\text{emb}}$; the output dimension $d_{\text{out}}$ is equal to the dimensionality of the event space, $2N_{\text{pred}}$, for vector fields, and is equal to one for energy models. As with path planning, we also add the output from a convolution prior, defined in (B.2), which operates only on the prediction tokens $\boldsymbol{\tau}_+$ post positional embedding. The dimensions $d_{\text{res}}$ and $d_{\text{emb}}$ were held fixed at 64 and 128 respectively, for all models.

The negative sampler NF model's architecture for I-R-NCE needed to be designed differently since the self-attention layers in Figure 7 were computationally prohibitive for negative sampling and log-probability computations, both of which are necessary for defining the I-R-NCE objective. As a result, we devised a novel Conv1D-Multi-Head-FiLM (CMHF) block, inspired from Perez et al. (2018), which, while inferior to the general self-attention block, was sufficiently powerful for training via I-R-NCE and computationally cheaper. The overall EBM+NF architecture and CMHF block are shown in Figures 8 and 9, respectively. Finally, drawing inspiration from the pre-conditioning function $c_{\text{noise}}(t)$ presented in Karras et al. (2022), we

---

[12]https://optax.readthedocs.io/en/latest/api.html#optax.warmup_cosine_decay_schedule
[13]Note that interpolation time $t \in [0, 1]$ is also an input to the EBM as we use the I-R-NCE formulation.

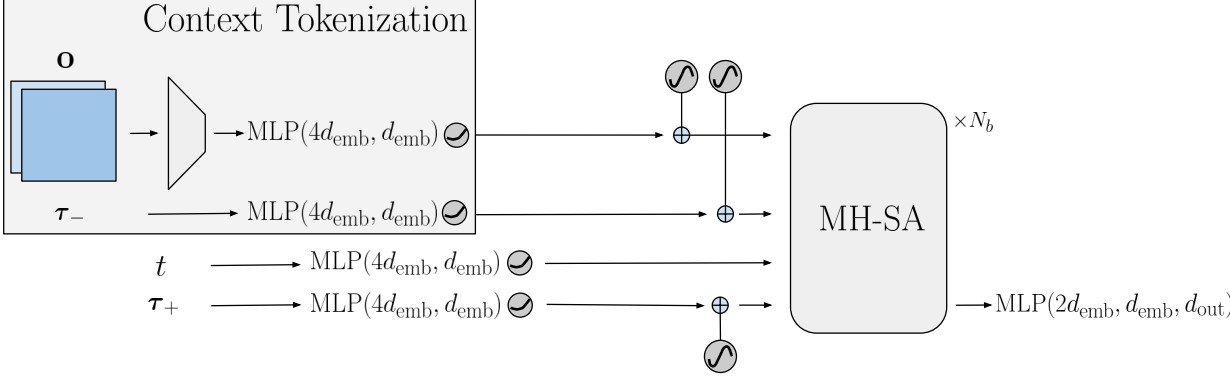

**Figure 7:** The encoder-only transformer architecture for Push-T. The $\boldsymbol{\tau}_+$ tokens after the terminal block are flattened and passed through the output MLP. Each MH-SA block used 4 attention heads.

applied the non-linear transform $t \mapsto -\log(1.1 - t)$ on the interpolation time $t$ used within I-R-NCE-EBM and I-R-NCE-NF. The advantage of this transform is to counteract the effect of the decreasing step-sizes from the transform $t \mapsto t^{1/\alpha}$ defined in Section 7.1.

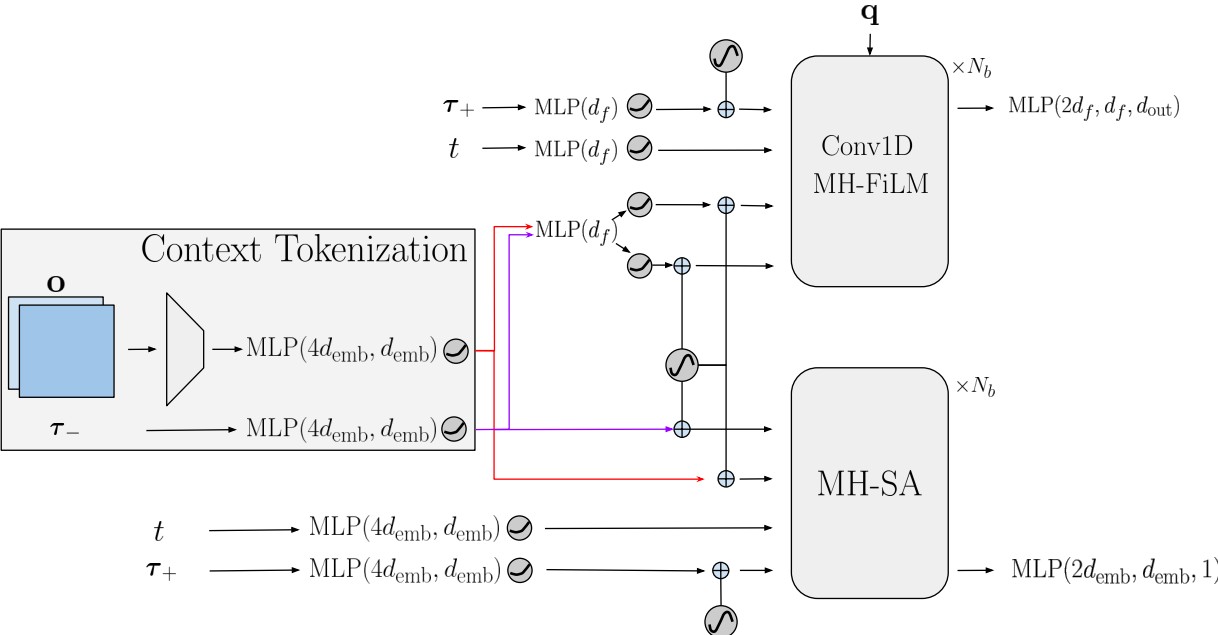

**Figure 8:** Combined NF (top) + EBM (bottom) architecture for I-R-NCE. To eliminate parameter redundancy, both models share the context tokenizer. However, only the sampler (cf. Algorithm 3, Line 9) or the EBM (cf. Algorithm 3, Line 14) is allowed to update the context tokenizer; we swept over both choices. Post context tokenization, the NF blocks use a smaller feature embedding, $d_f = d_{\text{emb}}/2 = 64$. The design of the Conv1D-Multi-Head-FiLM block is shown in Figure 9. We use 4 heads within the EBMs self-attention blocks and 2 heads within the NFs CMHF blocks. The additional input $\boldsymbol{q}$ to the CMHF blocks is explained in Figure 9.

Sampling during inference from the EBM models is done via Algorithm 4 with HMC sampling for the MCMC stage. We split the net amount of evaluations of $\nabla_y \mathcal{E}_\theta(x, y, t)$ between the SDE and MCMC stages using the interpolation time lower-bound $\underline{t}$; in particular, $\lfloor \underline{t} \cdot T_{\text{mcmc}} \rfloor$ evaluations are dedicated to the MCMC stage, while $\lfloor (1-\underline{t}) \cdot T_{\text{mcmc}} \rfloor$ evaluations are given to the SDE stage. This allows us to control the net inference-time computational fidelity.

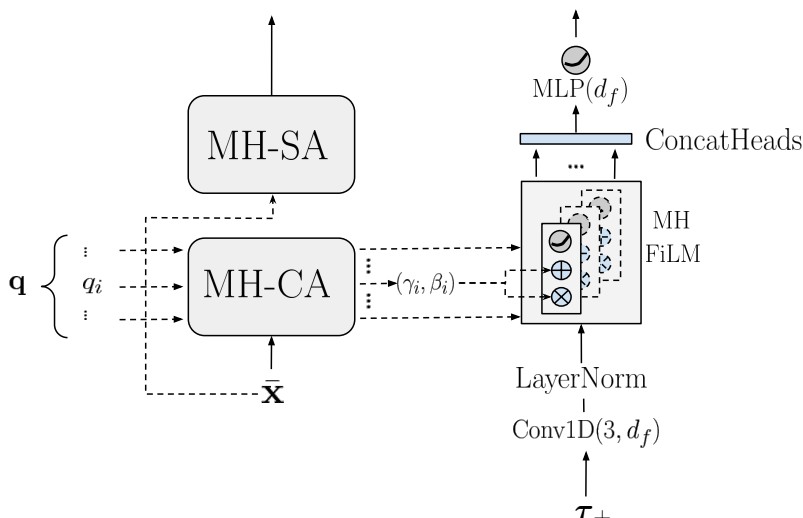

**Figure 9:** The Conv1D-MH-FiLM block. The input $\bar{\mathbf{x}}$ is the combined set of context tokens from $\mathbf{o}, \boldsymbol{\tau}_-$, and $t$. Each block has its own independent set of set of query tokens $\mathbf{q} \in \mathbb{R}^{n_q \times d_f}$, treated as learnable parameters, that cross-attend (MH-CA) to the composite context tokens $\bar{\mathbf{x}}$ to produce FiLM weights $\{(\gamma_i, \beta_i)\}_{i=1}^{n_q}$, where $\gamma_i, \beta_i \in \mathbb{R}^{d_f}$. The number of queries in each CMHF block $n_q$ was set to the number of heads (2). Each set of FiLM weights $(\gamma_i, \beta_i)$ are broadcasted across the temporal dimension of the prediction tokens $\boldsymbol{\tau}_+$ and affinely transform the prediction tokens (see Perez et al. (2018)) to yield $n_q$ copies of transformed prediction tokens. These are then concatenated (similar to a multi-head attention block) along the head-dimension, and projected back down to yield a set of output prediction tokens $\boldsymbol{\tau}_+$. Separately, the composite context tokens $\bar{\mathbf{x}}$ are propagated through a standard MH-SA block.

**Table 12:** List of hyperparameters for I-R-NCE. The constant $N_\mathrm{b}$ is described in Figure 8. $T_\mathrm{mcmc}$ refers to the *total* number of $\nabla_y \mathcal{E}_\theta$ evaluations, split between the SDE sampler and HMC leapfrog integrator steps, and $\eta$ refers to both the SDE noise-scale (see (5.7)) and HMC leapfrog integrator step size; we used 50 leapfrog steps per HMC momentum sampling step. The value $\underline{t}$ is the initial CNF sample time; see Algorithm 4. The constant $\alpha$ for the proposal NFs was fixed at 2.5. Note that in our sweeps, we exclude combinations of $N_\mathrm{b}$ for the EBM and NF which yield total parameter accounts exceeding $\sim$3.3M.

| Model | $N_\mathrm{b}$ | LR | WD | $T_\mathrm{mcmc}$ | $\eta$ | $\underline{t}$ |
|---|---|---|---|---|---|---|
| I-R-NCE-EBM | $\{6, 8\}$ | $2 \cdot 10^{-4}$ | $\{10^{-5}, 5 \cdot 10^{-6}, 10^{-6}\}$ | $\{750, 1000, 1250\}$ | $\{5 \cdot 10^{-4}, 10^{-3}\}$ | $\{0.45, 0.5\}$ |
| I-R-NCE-NF | $\{3, 4\}$ | $10^{-3}$ | $10^{-6}$ | - | - | - |

**Training.** We use a train batch size of 128 for the diffusion, NF, and I-R-NCE-NF models, and 64 for I-R-NCE-EBM. For Diffusion, Diffusion-$\phi$, and NF, we train our models for 200,000 batch steps and use a warmup cosine decay learning rate schedule with 5,000 warmup steps. For I-R-NCE, we use $T_\mathrm{outer} = 100,000$ main loop steps; each loop consists of $T_\mathrm{samp} = 4$ negative sampler batch steps, and $T_\mathrm{rnce} = 2$ R-NCE batch steps. Both models used a cosine decay learning rate schedule with a warmup of 1,500 main loop steps. We fix the number of negative samples for I-R-NCE to $K = 31$ and the number of interpolating times $m = 5$. For I-R-NCE, we also leveraged the computation trick from Section 6 by relying on 150 time-steps with a Heun integrator for sampling from the proposal model, and sub-sampling at only 15 time-steps for computing the log probability via trapezoidal integration. The remaining set of hyperparameters for I-R-NCE are outlined in Table 12.

Finally, we conclude by discussing the remaining set of hyperparameters for NF, Diffusion, and Diffusion-$\phi$. For all three models we hold the following hyperparameters fixed: (a) WD $= 10^{-6}$, (b) the number of sampling ODE steps equals 512 (so 1024 total number of function evaluations), and (c) the number of log probability ODE steps equal to 48 (cf. Section 6). For Diffusion and Diffusion-$\phi$, we use the default settings of $\sigma_\mathrm{data}$, $\sigma_\mathrm{min}$, $\sigma_\mathrm{max}$, and $\rho$ from Karras et al. (2022). We also fix $\sigma_\mathrm{rel} = 0.02$ for Diffusion-$\phi$. Finally, for NF, we fix $\alpha = 2$. The remaining hyperparameter values we sweep over are listed in Table 13.

**Table 13:** List of hyperparameters for NF, Diffusion, and Diffusion-$\phi$. The constant $N_{\mathrm{b}}$ is described in Figure 7. The parameter $\sigma_{\mathrm{pert}}$ governs the data noising, as discussed in the main text. All models considered have parameter counts not exceeding $\sim 3.3\mathrm{M}$.

| Model | $N_{\mathrm{b}}$ | LR | $\sigma_{\mathrm{pert}}$ |
|---|---|---|---|
| NF | | | |
| Diffusion | $\{4, 6, 8\}$ | $\{10^{-4}, 2 \cdot 10^{-4}, 5 \cdot 10^{-4}\}$ | $\{0, 0.01\}$ |
| Diffusion-$\phi$ | | | |

## C Proofs

### C.1 Optimality

Before we proceed, we state the following claim which will be used in the proof of Theorem 4.4.

**Claim C.1.** *Let $(\gamma_i)_{i=1}^d$ be a strictly positive vector in $\mathbb{R}^d$, and let $\mathcal{S}_{d-1}$ denote the $(d-1)$-dimensional simplex on $\mathbb{R}^d$. The function $F(q) = \sum_{i=1}^d \gamma_i \log q_i$ is uniquely maximized over $\mathcal{S}_{d-1}$ by $q = \gamma / \sum_{i=1}^d \gamma_i$.*

*Proof.* Follows from a standard Lagrange multiplier argument combined with the strict concavity of $\log x$ on the positive reals. $\square$

We now re-state and prove Theorem 4.4.

**Theorem 4.4** (Optimality)**.** *Suppose that $\mathscr{F}$ is realizable. Then, for any $\xi \in \Xi$, we have that:*

$$\Theta_\star = \underset{\theta \in \Theta}{\mathrm{argmax}}\, L(\theta, \xi). \tag{4.2}$$

*Proof.* As noted in Section 4.1, the proof strategy follows Ma & Collins (2018, Theorem 4.1), but includes the measure-theoretic arguments necessary in our setting. First, we note our assumptions ensure that $L(\theta, \xi)$ is finite for every $\theta, \xi \in \Theta \times \Xi$. Second, since the index $k$ does not affect the value in (3.8), we can take the average

$$L(\theta, \xi) = \frac{1}{K+1} \sum_{k=1}^{K+1} \mathbb{E}_x \mathbb{E}_{\bar{\mathbf{y}}|x;\xi,k}\ \log q_{\theta,\xi}(k \mid x, \bar{\mathbf{y}}).$$

We now use the existence of regular conditional densities to perform a change of measure argument as follows:[14]

$$
\begin{aligned}
L(\theta, \xi) &= \frac{1}{K+1} \mathbb{E}_x \left[ \sum_{k=1}^{K+1} \mathbb{E}_{\bar{\mathbf{y}}|x;\xi,k}\ \log q_{\theta,\xi}(k \mid x, \bar{\mathbf{y}}) \right] \\
&= \frac{1}{K+1} \mathbb{E}_x \left[ \sum_{k=1}^{K+1} \int \log q_{\theta,\xi}(k \mid x, \bar{\mathbf{y}}) p(y_k \mid x) \prod_{j \neq k} p_\xi(y_j \mid x)\, \mathrm{d}y_{1:K+1} \right] \\
&= \frac{1}{K+1} \mathbb{E}_x \left[ \sum_{k=1}^{K+1} \int \log q_{\theta,\xi}(k \mid x, \bar{\mathbf{y}}) \frac{p(y_k \mid x)}{p_\xi(y_k \mid x)} \prod_{j=1}^{K+1} p_\xi(y_j \mid x)\, \mathrm{d}y_{1:K+1} \right] \\
&= \frac{1}{K+1} \mathbb{E}_x \mathbb{E}_{\bar{\mathbf{y}} \sim \mathsf{P}^{K+1}_{\bar{\mathbf{y}}|x;\xi}} \left[ \sum_{k=1}^{K+1} \log q_{\theta,\xi}(k \mid x, \bar{\mathbf{y}}) \frac{p(y_k \mid x)}{p_\xi(y_k \mid x)} \right]. \tag{C.1}
\end{aligned}
$$

As a reminder, the notation $\bar{\mathbf{y}} \sim \mathsf{P}^{K+1}_{\bar{\mathbf{y}}|x;\xi}$ indicates that the $K+1$ samples in $\bar{\mathbf{y}}$ are drawn (conditionally on $x$) from the $K+1$ product distribution of $p_\xi(\cdot \mid x)$. Now, the term in the brackets is the (negative,

---

[14]Note that in the integral expressions that follow, we slightly abuse the measure-theoretic notation to imply the equivalence $\mathrm{d}y_{1:K+1} \equiv \mu_Y^{\times(K+1)}(\mathrm{d}y_{1:K+1})$, where $\mu_Y^{\times(K+1)}$ is the product measure.

un-normalized) cross entropy between the following distributions over $\{1, \ldots, K+1\}$: the (un-normalized) distribution $\{\frac{p(y_k|x)}{p_\xi(y_k|x)}\}_{k=1}^{K+1}$ and the distribution $\{q_{\theta,\xi}(k \mid x, \bar{\mathbf{y}})\}_{k=1}^{K+1}$.

Let $\theta_\star \in \Theta_\star$. By the definition of realizability, we have for a.e. $(x, \bar{\mathbf{y}})$:

$$q_{\theta_\star,\xi}(k \mid x, \bar{\mathbf{y}}) = \frac{p(y_k \mid x)/p_\xi(y_k \mid x)}{\sum_{j=1}^{K+1} p(y_j \mid x)/p_\xi(y_j \mid x)} \quad \forall k \in \{1, \ldots, K+1\}.$$

Thus, we see by Claim C.1 that the choice of $\theta_\star \in \Theta_\star$ pointwise maximizes the cross-entropy term in the brackets in (C.1), and hence $\Theta_\star \subseteq \arg\max_{\theta \in \Theta} L(\theta, \xi)$.

Now, suppose that $\bar{\theta} \in \arg\max_{\theta \in \Theta} L(\theta, \xi)$. Let $\theta_\star \in \Theta_\star$ be arbitrary. Since $L(\bar{\theta}, \xi) = L(\theta_\star, \xi)$ and since $\theta_\star$ pointwise maximizes (C.1), by Claim C.1 we must have that for a.e. $(x, \bar{\mathbf{y}})$:

$$q_{\bar{\theta},\xi}(k \mid x, \bar{\mathbf{y}}) = q_{\theta_\star,\xi}(k \mid x, \bar{\mathbf{y}}) \quad \forall k \in \{1, \ldots, K+1\}. \tag{C.2}$$

Note that when (C.2) holds, we have:

$$\frac{\exp(\mathcal{E}_{\bar{\theta}}(x, y_k))}{\exp(\mathcal{E}_{\theta_\star}(x, y_k))} = \frac{\sum_{y \in \bar{\mathbf{y}}} \exp(\mathcal{E}_{\bar{\theta}}(x, y) - \log p_\xi(y \mid x))}{\sum_{y \in \bar{\mathbf{y}}} \exp(\mathcal{E}_{\theta_\star}(x, y) - \log p_\xi(y \mid x))} \quad \forall k \in \{1, \ldots, K+1\}.$$

Because the expression on the RHS above is independent of $k$, then:

$$\exp(\mathcal{E}_{\bar{\theta}}(x, y_k) - \mathcal{E}_{\bar{\theta}}(x, y_\ell)) = \exp(\mathcal{E}_{\theta_\star}(x, y_k) - \mathcal{E}_{\theta_\star}(x, y_\ell)) \quad \forall k, \ell \in \{1, \ldots, K+1\}. \tag{C.3}$$

Thus, defining the set $A := \{(x, \bar{\mathbf{y}}) \mid (C.3) \text{ holds}\}$, we have:

$$(\mu_X \times \mu_Y^{\times(K+1)})(A) = 1. \tag{C.4}$$

We now define the following set on $\mathsf{X} \times \mathsf{Y}^2$:

$$B := \{(x, y, y') \mid \exp(\mathcal{E}_{\bar{\theta}}(x, y) - \mathcal{E}_{\bar{\theta}}(x, y')) \neq \exp(\mathcal{E}_{\theta_\star}(x, y) - \mathcal{E}_{\theta_\star}(x, y'))\}.$$

Notice that $(x, y, y') \in B$ if and only if $(x, y, y') \bigcup \mathsf{Y}^{K-1} \in A^c$. Thus, letting $c = \int d\mu_Y^{\times K-1}$ (here we use the finiteness of the measure space), it follows from Tonelli's theorem and (C.4):

$$\begin{aligned}
(\mu_X \times \mu_Y^{\times 2})(B) &= \int \mathbf{1}\{(x, y_1, y_2) \in B\} \, d(\mu_X \times \mu_Y^{\times 2}) \\
&= c^{-1} \int \left[ \int \mathbf{1}\{(x, y_1, y_2) \in B\} \, d(\mu_X \times \mu_Y^{\times 2}) \right] d\mu_Y^{\times(K-1)} \\
&= c^{-1} \int \mathbf{1}\{(x, y_1, y_2) \in B\} \, d(\mu_X \times \mu_Y^{\times(K+1)}) \\
&= c^{-1} \int \mathbf{1}\{(x, \bar{\mathbf{y}}) \in A^c\} \, d(\mu_X \times \mu_Y^{\times(K+1)}) = 0.
\end{aligned}$$

Next, we have that $\mu_\mathsf{X} \times \mu_\mathsf{Y}$ measure of the set

$$B' := \{(x, y) \mid \text{the cross-section } B''(x, y) := \{y' \mid (x, y, y') \in B\} \text{ is not } \mu_\mathsf{Y}\text{-null}\}$$

is also zero, due to the fact that if $\nu_1$ and $\nu_2$ are measures, then cross-sections of $\nu_1 \times \nu_2$ null sets (holding the first variable fixed) are $\nu_2$ null sets, $\nu_1$ almost everywhere (see e.g. Folland, 1999, Ch. 2, Ex. 49). Hence

for any $(x, y) \in (B')^c$:

$$
\begin{aligned}
p_{\bar{\theta}}(y \mid x) &= \frac{\exp(\mathcal{E}_{\bar{\theta}}(x, y))}{\int \exp(\mathcal{E}_{\bar{\theta}}(x, y')) \, \mu_Y(\mathrm{d}y')} \\
&= \frac{1}{\int \exp(\mathcal{E}_{\bar{\theta}}(x, y') - \mathcal{E}_{\bar{\theta}}(x, y)) \, \mu_Y(\mathrm{d}y')} \\
&= \frac{1}{\int_{B''(x,y)^c} \exp(\mathcal{E}_{\bar{\theta}}(x, y') - \mathcal{E}_{\bar{\theta}}(x, y)) \, \mu_Y(\mathrm{d}y')} \\
&= \frac{1}{\int_{B''(x,y)^c} \exp(\mathcal{E}_{\theta_\star}(x, y') - \mathcal{E}_{\theta_\star}(x, y)) \, \mu_Y(\mathrm{d}y')} \\
&= \frac{1}{\int \exp(\mathcal{E}_{\theta_\star}(x, y') - \mathcal{E}_{\theta_\star}(x, y)) \, \mu_Y(\mathrm{d}y')} \\
&= p_{\theta_\star}(y \mid x).
\end{aligned}
$$

This shows that $p_{\bar{\theta}}(y \mid x) = p_{\theta_\star}(y \mid x)$ holds $\mu_X \times \mu_Y$ almost everywhere, and hence $\bar{\theta} \in \Theta_\star$. $\qquad \square$

### C.2 Asymptotic Convergence

### C.2.1 Consistency

In order to prove Theorem 4.8, we require the following technical lemma.

**Lemma C.2.** *Let $B_2(r)$ denote the closed Euclidean ball centered at the origin with radius $r$; the dimension will be implicit from context. Let $\Theta$ be a compact set, and let $f : \Theta \to \mathbb{R}$ be a continuous function. Denote the maximization set*

$$
\Theta_\star := \left\{ \theta \in \Theta \mid f(\theta) = \sup_{\theta' \in \Theta} f(\theta') \right\}.
$$

*Suppose that $\Theta_\star$ is contained in the interior of $\Theta$. There exists a $\varepsilon_0 > 0$ such that for all $0 < \varepsilon \leqslant \varepsilon_0$,*

$$
\sup_{\theta \in \mathrm{cl}(\Theta \setminus (\Theta_\star + B_2(\varepsilon)))} f(\theta) < f^\star := \sup_{\theta \in \Theta} f(\theta),
$$

*where $\mathrm{cl}(\cdot)$ denotes the closure of a set.*

*Proof.* For $\varepsilon > 0$, define $\Theta^\varepsilon := \mathrm{cl}(\Theta \setminus (\Theta_\star + B_2(\varepsilon)))$. Since $\Theta_\star$ is assumed to be in the interior of $\Theta$, there exists an $\varepsilon_0 > 0$ such that for all $0 < \varepsilon \leqslant \varepsilon_0$, $\Theta^\varepsilon$ is non-empty. It is also compact by boundedness of $\Theta$. Let $\{\theta_k\}_k$ be a sequence within $\Theta^\varepsilon$ such that $f(\theta_k) \to \sup_{\theta \in \Theta^\varepsilon} f(\theta)$. Suppose for a contradiction that $\sup_{\theta \in \Theta^\varepsilon} f(\theta) = f^\star$. By compactness of $\Theta^\varepsilon$, there exists a convergent subsequence $\{\theta_{k_j}\}_j$ with limit $\theta^\varepsilon \in \Theta^\varepsilon$. By continuity, $f(\theta^\varepsilon) = f^\star$, and thus $\theta^\varepsilon \in \Theta_\star$. But, this means that $\Theta^\varepsilon \cap \Theta_\star$ is non-empty, a contradiction. $\quad \square$

We now restate and prove Theorem 4.8.

**Theorem 4.8** (General Consistency)**.** *Suppose that $\mathscr{F}$ is realizable, $\Theta_\star$ is contained in the interior of $\Theta$, and Assumptions 4.1 and 4.2 hold. Let $\hat{\theta}_n \in \mathrm{argmax}_{\theta \in \Theta} L_n(\theta, \hat{\xi}_n)$ denote an arbitrary empirical risk maximizer from $n$ samples. Then, $d(\hat{\theta}_n, \Theta_\star) \overset{n \to \infty}{\longrightarrow} 0$ almost surely.*

*Proof.* Let $\theta_\star \in \Theta_\star$ be arbitrary. For $\theta \in \Theta$, define the function $g(\theta)$ as:

$$
g(\theta) := \sup_{\xi \in \Xi} [L(\theta, \xi) - L(\theta_\star, \xi)].
$$

By the continuity of $L$ and the envelope theorem, $g(\theta)$ is continuous. Also, we have:

$$
\sup_{\theta \in \Theta} g(\theta) = \sup_{\theta \in \Theta} \sup_{\xi \in \Xi} [L(\theta, \xi) - L(\theta_\star, \xi)] = \sup_{\xi \in \Xi} \sup_{\theta \in \Theta} [L(\theta, \xi) - L(\theta_\star, \xi)] = 0,
$$

by Theorem 4.4. By Lemma C.2, since $\Theta_\star$ is contained in the interior of $\Theta$, there exists $\varepsilon_0 > 0$ such that for all $0 < \varepsilon \leqslant \varepsilon_0$:

$$\theta \in \mathrm{cl}(\Theta \setminus (\Theta_\star + B_2(\varepsilon))) \implies g(\theta) < 0.$$

Next, by optimality of $\hat{\theta}_n$, $L_n(\hat{\theta}_n, \hat{\xi}_n) \geqslant L_n(\theta_\star, \hat{\xi}_n)$. Thus,

$$
\begin{aligned}
L(\theta_\star, \hat{\xi}_n) - L(\hat{\theta}_n, \hat{\xi}_n) &= L(\theta_\star, \hat{\xi}_n) - L_n(\hat{\theta}_n, \hat{\xi}_n) + L_n(\hat{\theta}_n, \hat{\xi}_n) - L(\hat{\theta}_n, \hat{\xi}_n) \\
&\leqslant L(\theta_\star, \hat{\xi}_n) - L_n(\theta_\star, \hat{\xi}_n) + L_n(\hat{\theta}_n, \hat{\xi}_n) - L(\hat{\theta}_n, \hat{\xi}_n) \\
&\leqslant 2 \sup_{\theta \in \Theta, \xi \in \Xi} |L(\theta, \xi) - L_n(\theta, \xi)|.
\end{aligned}
$$

Therefore, we obtain the implication:

$$
\begin{aligned}
d(\hat{\theta}_n, \Theta_\star) > \varepsilon &\implies g(\hat{\theta}_n) < 0 \\
&\implies L(\hat{\theta}_n, \hat{\xi}_n) - L(\theta_\star, \hat{\xi}_n) < 0 \\
&\implies 2 \sup_{\theta \in \Theta, \xi \in \Xi} |L(\theta, \xi) - L_n(\theta, \xi)| > 0.
\end{aligned}
$$

Now, let $(\theta_0, \xi_0)$ be as indicated in Assumption 4.2, and define $U(\theta, \xi) := L(\theta, \xi) - L(\theta_0, \xi_0)$ and similarly $U_n(\theta, \xi) := L_n(\theta, \xi) - L_n(\theta_0, \xi_0)$. By triangle inequality:

$$\sup_{\theta \in \Theta, \xi \in \Xi} |L_n(\theta, \xi) - L(\theta, \xi)| \leqslant \sup_{\theta \in \Theta, \xi \in \Xi} |U_n(\theta, \xi) - U(\theta, \xi)| + |L_n(\theta_0, \xi_0) - L(\theta_0, \xi_0)|.$$

Hence, we have the following event inclusion:

$$
\begin{aligned}
&\left\{ \limsup_{n \to \infty} d(\hat{\theta}_n, \Theta_\star) > \varepsilon \right\} \\
&\subset \left\{ \limsup_{n \to \infty} \sup_{\theta \in \Theta, \xi \in \Xi} |U_n(\theta, \xi) - U(\theta, \xi)| > 0 \right\} \bigcup \left\{ \limsup_{n \to \infty} |L_n(\theta_0, \xi_0) - L(\theta_0, \xi_0)| > 0 \right\}.
\end{aligned}
$$

By the compactness of $\Theta \times \Xi$, continuity of $(\theta, \xi) \mapsto \bar{\ell}_{\theta, \xi}(x, y)$, and Assumption 4.2, the first event on the RHS has measure zero by the uniform converge result of Ferguson (2017, Theorem 16(a)).

On the other, the second event on the RHS also has measure zero by the strong law of large numbers (SLLN), which simply requires the measurability of $\bar{\ell}_{\theta, \xi}$ and the existence of $(\theta_0, \xi_0)$. Hence, $\left\{ \limsup_{n \to \infty} d(\hat{\theta}_n, \Theta_\star) > \varepsilon \right\}$ is a measure zero set for every $\varepsilon > 0$ sufficiently small, establishing the result. $\qquad \square$

### C.2.2 Asymptotic Normality

The following technical lemma will be necessary for what follows.

**Lemma C.3** (Symmetrization lemma, (cf. Ma & Collins, 2018, Lemma B.4)). *Let $M(x, y)$ be a measurable function. For any $k \in \{1, \ldots, K+1\}$, define:*

$$\mathcal{Z}_k[M](\theta, \xi) := \mathbb{E}_x \mathbb{E}_{\bar{\mathbf{y}}|x; \xi, k} \left[ \sum_{i=1}^{K+1} q_{\theta, \xi}(i \mid x, \bar{\mathbf{y}}) M(x, y_i) \right].$$

*For any $\theta_\star \in \Theta_\star$ and $\xi \in \Xi$, we have:*

$$\mathcal{Z}_k[M](\theta_\star, \xi) = \mathbb{E}_{x, y}[M(x, y)].$$

*Proof.* Wlog we can take $k = 1$. By symmetry:

$$\mathcal{Z}_1[M](\theta, \xi) = \frac{1}{K+1} \sum_{k=1}^{K+1} \mathcal{Z}_k[M](\theta, \xi)$$

$$= \frac{1}{K+1} \sum_{k=1}^{K+1} \mathbb{E}_x \mathbb{E}_{\bar{\mathbf{y}} \sim \mathsf{P}_{\bar{y}|x;\xi}^{K+1}} \left[ \left( \sum_{i=1}^{K+1} q_{\theta,\xi}(i \mid x, \bar{\mathbf{y}}) M(x, y_i) \right) \frac{p(y_k \mid x)}{p_\xi(y_k \mid x)} \right]$$

$$= \frac{1}{K+1} \mathbb{E}_x \mathbb{E}_{\bar{\mathbf{y}} \sim \mathsf{P}_{\bar{y}|x;\xi}^{K+1}} \left[ \sum_{i=1}^{K+1} q_{\theta,\xi}(i \mid x, \bar{\mathbf{y}}) M(x, y_i) \left( \sum_{k=1}^{K+1} \frac{p(y_k \mid x)}{p_\xi(y_k \mid x)} \right) \right].$$

For any $\theta \in \Theta_\star$:

$$q_{\theta_\star, \xi}(i \mid x, \bar{\mathbf{y}}) = \frac{p(y_i \mid x)/p_\xi(y_i \mid x)}{\sum_{k=1}^{K+1} p(y_k \mid x)/p_\xi(y_k \mid x)}.$$

Hence:

$$\mathcal{Z}_1[M](\theta_\star, \xi) = \frac{1}{K+1} \mathbb{E}_x \mathbb{E}_{\bar{\mathbf{y}} \sim \mathsf{P}_{\bar{y}|x;\xi}^{K+1}} \left[ \sum_{i=1}^{K+1} M(x, y_i) \frac{p(y_i \mid x)}{p_\xi(y_i \mid x)} \right]$$

$$= \frac{1}{K+1} \sum_{i=1}^{K+1} \mathbb{E}_x \mathbb{E}_{y_i \sim p(\cdot|x)} [M(x, y_i)]$$

$$= \mathbb{E}_{x,y}[M(x, y)].$$

$\square$

We prove Theorem 4.10 in two parts, starting with the following proposition.

**Proposition C.4.** *Let $\theta_\star \in \Theta_\star$ and $\xi_\star \in \Xi$. Put $\gamma_\star := (\theta_\star, \xi_\star)$. We have that:*

$$\mathrm{Var}_z \left[ \nabla_\theta \bar{\ell}_{\gamma_\star}(z) \right] = -\nabla_\theta^2 L(\theta_\star, \xi_\star).$$

*Proof.* We first observe that since $\mathbb{E}_z[\nabla_\theta \bar{\ell}_{\gamma_\star}(z)] = 0$ by optimality, the following identity holds:

$$\mathrm{Var}_z \left[ \nabla_\theta \bar{\ell}_{\gamma_\star}(z) \right] = \mathbb{E}_z \left[ \nabla_\theta \bar{\ell}_{\gamma_\star}(z) \nabla_\theta \bar{\ell}_{\gamma_\star}(z)^\mathsf{T} \right].$$

Now, define the following shorthand notation (with arguments $x$ and $\bar{\mathbf{y}}$ implicit):

$$g_k := \nabla_\theta \mathcal{E}_{\theta_\star}(x, y_k), \quad q_k := q_{\theta_\star, \xi_\star}(k \mid x, \bar{\mathbf{y}}), \quad k \in \{1, \dots, K+1\}.$$

we have by symmetry and the definition of $q_k$:

$$\mathbb{E}_x \mathbb{E}_{\bar{\mathbf{y}}|x, \xi_\star, k} \left[ \sum_{i=1}^{K+1} q_i g_i g_k^\mathsf{T} \right] = \frac{1}{K+1} \sum_{k=1}^{K+1} \mathbb{E}_x \mathbb{E}_{\bar{\mathbf{y}} \sim \mathsf{P}_{\bar{y}|x;\xi}^{K+1}} \left[ \sum_{i=1}^{K+1} q_i g_i g_k^\mathsf{T} \frac{p(y_k \mid x)}{p_{\xi_\star}(y_k \mid x)} \right]$$

$$= \frac{1}{K+1} \sum_{k=1}^{K+1} \mathbb{E}_x \mathbb{E}_{\bar{\mathbf{y}} \sim \mathsf{P}_{\bar{y}|x;\xi}^{K+1}} \left[ \sum_{i=1}^{K+1} q_i g_i g_k^\mathsf{T} q_k \left( \sum_{j=1}^{K+1} \frac{p(y_j \mid x)}{p_{\xi_\star}(y_j \mid x)} \right) \right]$$

$$= \frac{1}{K+1} \sum_{j=1}^{K+1} \mathbb{E}_x \mathbb{E}_{\bar{\mathbf{y}} \sim \mathsf{P}_{\bar{y}|x;\xi}^{K+1}} \left[ \frac{p(y_j \mid x)}{p_{\xi_\star}(y_j \mid x)} \sum_{i=1}^{K+1} \sum_{k=1}^{K+1} q_i g_i g_k^\mathsf{T} q_k \right]$$

$$= \frac{1}{K+1} \sum_{j=1}^{K+1} \mathbb{E}_x \mathbb{E}_{\bar{\mathbf{y}}|x, \xi_\star, j} \left[ \sum_{i=1, k=1}^{K+1} q_i g_i q_k g_k^\mathsf{T} \right]$$

$$= \mathbb{E}_x \mathbb{E}_{\bar{\mathbf{y}}|x, \xi_\star, k} \left[ \sum_{i=1, j=1}^{K+1} q_i g_i q_j g_j^\mathsf{T} \right] = \mathbb{E}_x \mathbb{E}_{\bar{\mathbf{y}}|x, \xi_\star, k} \left[ \left( \sum_{i=1}^{K+1} q_i g_i \right)^{\otimes 2} \right],$$

where $v^{\otimes 2} = v v^{\mathsf{T}}$ denotes the outer product of a vector $v$. With this calculation, we have that:

$$\mathrm{Var}_z \left[ \nabla_\theta \bar{\ell}_{\gamma_\star}(z) \right]$$

$$= \mathbb{E}_x \mathbb{E}_{\bar{\mathbf{y}}|x,\xi_\star,k} \left[ \left( \nabla_\theta \mathcal{E}_{\theta_\star}(x, y_k) - \sum_{i=1}^{K+1} q_{\theta_\star,\xi_\star}(i \mid x, \bar{\mathbf{y}}) \nabla_\theta \mathcal{E}_{\theta_\star}(x, y_i) \right)^{\otimes 2} \right]$$

$$= \mathbb{E}_{x,y}[\nabla_\theta \mathcal{E}_{\theta_\star}(x, y) \nabla_\theta \mathcal{E}_{\theta_\star}(x, y)^{\mathsf{T}}] - \mathbb{E}_x \mathbb{E}_{\bar{\mathbf{y}}|x,\xi_\star,k} \left[ \left( \sum_{i=1}^{K+1} q_{\theta_\star,\xi_\star}(i \mid x, \bar{\mathbf{y}}) \nabla_\theta \mathcal{E}_{\theta_\star}(x, y_i) \right)^{\otimes 2} \right]. \tag{C.5}$$

It remains to show that the expression (C.5) is equal to $-\nabla_\theta^2 L(\theta_\star, \xi_\star)$, from which we conclude the result. To do this, we next recall the following identity:

$$\nabla_\theta q_{\theta,\xi}(k \mid x, \bar{\mathbf{y}}) = q_{\theta,\xi}(k \mid x, \bar{\mathbf{y}}) \nabla_\theta \log q_{\theta,\xi}(k \mid x, \bar{\mathbf{y}}).$$

Hence for any $k \in \{1, \ldots, K+1\}$, letting $\partial_\theta(\cdot)$ denote the Jacobian of the argument,

$$\partial_\theta(q_{\theta,\xi}(k \mid x, \bar{\mathbf{y}}) \nabla_\theta \mathcal{E}_\theta(x, y_k))$$

$$= q_{\theta,\xi}(k \mid x, \bar{\mathbf{y}}) \left[ \nabla_\theta^2 \mathcal{E}_\theta(x, y_k) + \nabla_\theta \mathcal{E}_\theta(x, y_k) \nabla_\theta \log q_{\theta,\xi}(k \mid x, \bar{\mathbf{y}})^{\mathsf{T}} \right]$$

$$= q_{\theta,\xi}(k \mid x, \bar{\mathbf{y}}) \left[ \nabla_\theta^2 \mathcal{E}_\theta(x, y_k) + \nabla_\theta \mathcal{E}_\theta(x, y_k) \nabla_\theta \mathcal{E}_\theta(x, y_k)^{\mathsf{T}} - \sum_{i=1}^{K+1} q_{\theta,\xi}(i \mid x, \bar{\mathbf{y}}) \nabla_\theta \mathcal{E}_\theta(x, y_k) \nabla_\theta \mathcal{E}_\theta(x, y_i)^{\mathsf{T}} \right].$$

Recalling the following expression for the gradient $\nabla_\theta L(\theta, \xi)$:

$$\nabla_\theta L(\theta, \xi) = \mathbb{E}_{x,y} \nabla_\theta \mathcal{E}_\theta(x, y) - \mathbb{E}_x \mathbb{E}_{\bar{\mathbf{y}}|x;\xi} \left[ \sum_{i=1}^{K+1} q_{\theta,\xi}(i \mid x, \bar{\mathbf{y}}) \nabla_\theta \mathcal{E}_\theta(x, y_i) \right],$$

we have that the Hessian $\nabla_\theta^2 L(\theta, \xi)$ is:

$$\nabla_\theta^2 L(\theta, \xi) = \mathbb{E}_x \mathbb{E}_{\bar{\mathbf{y}}|x;\xi} \left[ \nabla_\theta^2 \mathcal{E}_\theta(x, y_k) - \sum_{i=1}^{K+1} q_{\theta,\xi}(i \mid x, \bar{\mathbf{y}}) \left\{ \nabla_\theta^2 \mathcal{E}_\theta(x, y_i) + \nabla_\theta \mathcal{E}_\theta(x, y_i) \nabla_\theta \mathcal{E}_\theta(x, y_i)^{\mathsf{T}} \right\} \right]$$

$$+ \mathbb{E}_x \mathbb{E}_{\bar{\mathbf{y}}|x;\xi} \left[ \sum_{i=1}^{K+1} \sum_{i'=1}^{K+1} q_{\theta,\xi}(i \mid x, \bar{\mathbf{y}}) q_{\theta,\xi}(i' \mid x, \bar{\mathbf{y}}) \nabla_\theta \mathcal{E}_\theta(x, y_i) \nabla_\theta \mathcal{E}_\theta(x, y_{i'})^{\mathsf{T}} \right]$$

$$= \mathbb{E}_x \mathbb{E}_{\bar{\mathbf{y}}|x;\xi} \left[ \nabla_\theta^2 \mathcal{E}_\theta(x, y_k) - \sum_{i=1}^{K+1} q_{\theta,\xi}(i \mid x, \bar{\mathbf{y}}) \left\{ \nabla_\theta^2 \mathcal{E}_\theta(x, y_i) + \nabla_\theta \mathcal{E}_\theta(x, y_i) \nabla_\theta \mathcal{E}_\theta(x, y_i)^{\mathsf{T}} \right\} \right]$$

$$+ \mathbb{E}_x \mathbb{E}_{\bar{\mathbf{y}}|x;\xi} \left[ \left( \sum_{i=1}^{K+1} q_{\theta,\xi}(i \mid x, \bar{\mathbf{y}}) \nabla_\theta \mathcal{E}_\theta(x, y_i) \right)^{\otimes 2} \right].$$

Let us now simplify this by evaluating the Hessian at $(\theta, \xi) = (\theta_\star, \xi_\star)$. Applying the symmetrization lemma (Lemma C.3) and the identity (C.5), we have:

$$\nabla_\theta^2 L(\theta_\star, \xi_\star) = -\mathbb{E}_{x,y}[\nabla_\theta \mathcal{E}_{\theta_\star}(x, y) \nabla_\theta \mathcal{E}_{\theta_\star}(x, y)^{\mathsf{T}}] + \mathbb{E}_x \mathbb{E}_{\bar{\mathbf{y}}|x;\xi_\star} \left[ \left( \sum_{i=1}^{K+1} q_{\theta_\star,\xi_\star}(i \mid x, \bar{\mathbf{y}}) \nabla_\theta \mathcal{E}_{\theta_\star}(x, y_i) \right)^{\otimes 2} \right] \tag{C.6}$$

$$= -\mathrm{Var}_z \left[ \nabla_\theta \bar{\ell}_{\gamma_\star}(z) \right].$$

$\square$

We now proceed to restate and prove Theorem 4.10.

**Theorem 4.10** (Asymptotic Normality)**.** *Suppose that $\mathscr{F}$ is realizable and Assumptions 4.1, 4.2, and 4.9 hold. Let $\{\hat\theta_n\}$ be a sequence of optimizers as defined via (4.5). Denote the joint parameter $\gamma := (\theta, \xi)$ and joint r.v. $z := (x, y)$, and assume the following properties hold:*

1. *The negative proposal distribution parameters $\{\xi_n\}$ are $\sqrt{n}$-consistent about a fixed point $\xi_\star$, i.e., $\sqrt{n}(\hat\xi_n - \xi_\star) = O_p(1)$.*

2. *The set $\Theta_\star = \{\theta_\star\}$ is a singleton, and $\gamma_\star := (\theta_\star, \xi_\star)$ lies in the interior of $\Theta \times \Xi$.*

3. *The Hessian of $\bar\ell_\gamma$ is Lipschitz-continuous, i.e.,*

$$\|\nabla_\gamma^2 \bar\ell_{\gamma_1}(z) - \nabla_\gamma^2 \bar\ell_{\gamma_2}(z)\|_{\mathrm{op}} \leqslant M(z)\|\gamma_1 - \gamma_2\|, \quad \mathbb{E}_z[M(z)] < \infty, \ \forall \gamma_1, \gamma_2 \in \Theta \times \Xi.$$

4. *The $(\theta, \theta)$ block of the Hessian $\nabla_\gamma^2 L(\theta_\star, \xi_\star)$, denoted $\nabla_\theta^2 L(\theta_\star, \xi_\star)$, satisfies $\nabla_\theta^2 L(\theta_\star, \xi_\star) \prec 0$.*

*Then, $\sqrt{n}(\hat\theta_n - \theta_\star) \rightsquigarrow \mathcal{N}(0, V_\theta)$, where*

$$V_\theta = -\nabla_\theta^2 L(\theta_\star, \xi_\star)^{-1} = -\mathbb{E}_z\left[\nabla_\theta^2 \bar\ell_{\gamma_\star}(z)\right]^{-1}.$$

*Proof.* We first require the following remainder version of the mean-value theorem, which is a consequence of Taylor's theorem.

**Claim C.5.** *By the $\mathcal{C}^2$ smoothness and Lipschitz Hessian properties of $\gamma \mapsto \bar\ell_\gamma(z)$, for any $z \in \mathsf{X} \times \mathsf{Y}$, and $\gamma_1, \gamma_2 \in \Theta \times \Xi$, it holds:*

$$\nabla_\gamma \bar\ell_{\gamma_2}(z) = \nabla_\gamma \bar\ell_{\gamma_1}(z) + \nabla_\gamma^2 \bar\ell_{\gamma_1}(z)(\gamma_2 - \gamma_1) + R(\gamma_2 - \gamma_1),$$

*where $R$ is a remainder matrix dependent upon $z, \gamma_1, \gamma_2$, and satisfies: $\|R\|_{\mathrm{op}} \leqslant \frac{1}{2} M(z)\|\gamma_2 - \gamma_1\|$.*

Applying Claim C.5 with $(\gamma_1, \gamma_2) = (\gamma_\star, \hat\gamma_n)$, where $\hat\gamma_n := (\hat\theta_n, \hat\xi_n)$, we obtain:

$$\nabla_\theta \bar\ell_{\hat\gamma_n}(z) = \nabla_\theta \bar\ell_{\gamma_\star}(z) + \nabla_\theta^2 \bar\ell_{\gamma_\star}(z)(\hat\theta_n - \theta_\star) + \nabla_{\theta\xi}^2 \bar\ell_{\gamma_\star}(z)(\hat\xi_n - \xi_\star) + \underbrace{\begin{bmatrix} I_{|\theta|} \\ O_{|\xi|} \end{bmatrix}^{\mathsf{T}}}_{:= B^{\mathsf{T}}} R(\hat\gamma_n - \gamma_\star),$$

where $I_d$ and $O_d$ are the identity and zero matrices of dimensions $d \times d$ respectively. Now, since $\hat\theta_n$ converges to $\theta_\star$ a.s., for $n$ sufficiently large, the optimizer $\hat\theta_n \in \arg\max_{\theta \in \Theta} \hat{\mathbb{E}}_z \bar\ell_{(\theta, \hat\xi_n)}(z)$ also lies in the interior of $\Theta$, and thus, $\nabla_\theta \hat{\mathbb{E}}_z \bar\ell_{\hat\gamma_n}(z) = 0$. Applying the empirical expectation on all terms above yields:

$$\begin{aligned}
0 &= \nabla_\theta \hat{\mathbb{E}}_z \bar\ell_{\hat\gamma_n}(z) \\
&= \hat{\mathbb{E}}_z[\nabla_\theta \bar\ell_{\gamma_\star}(z)] + \hat{\mathbb{E}}_z[\nabla_\theta^2 \bar\ell_{\gamma_\star}(z)](\hat\theta_n - \theta_\star) + \hat{\mathbb{E}}_z[\nabla_{\theta\xi}^2 \bar\ell_{\gamma_\star}(z)](\hat\xi_n - \xi_\star) \\
&\quad + B^{\mathsf{T}} \hat{\mathbb{E}}_z[R(\hat\gamma_n, \gamma_\star, z)](\hat\gamma_n - \gamma_\star).
\end{aligned}$$

Now, notice that:

$$\|\hat{\mathbb{E}}_z[R(\hat\gamma_n, \gamma_\star, z)]\|_{\mathrm{op}} = \left\|\frac{1}{n}\sum_{i=1}^n R(\hat\gamma_n, \gamma_\star, z^{(i)})\right\|_{\mathrm{op}} \leqslant \frac{1}{n}\sum_{i=1}^n \|R(\hat\gamma_n, \gamma_\star, z^{(i)})\|_{\mathrm{op}}$$

$$\leqslant \underbrace{\frac{1}{n}\sum_{i=1}^n \frac{1}{2}M(z^{(i)})}_{\xrightarrow{\text{a.s.}} \frac{1}{2}\mathbb{E}_z[M(z)]} \underbrace{\|\hat\gamma_n - \gamma_\star\|}_{\xrightarrow{p} 0} = o_p(1).$$

Then, the remainder term takes the form $o_p(1)(\hat{\theta}_n - \theta_\star) + o_p(1)(\hat{\xi}_n - \xi_\star)$. Thus,

$$
\begin{aligned}
0 &= \hat{\mathbb{E}}_z[\nabla_\theta \bar{\ell}_{\gamma_\star}(z)] + \left(\hat{\mathbb{E}}_z[\nabla_\theta^2 \bar{\ell}_{\gamma_\star}(z)] + o_p(1)\right)(\hat{\theta}_n - \theta_\star) + \left(\hat{\mathbb{E}}_z[\nabla_{\theta\xi}^2 \bar{\ell}_{\gamma_\star}(z)] + o_p(1)\right)(\hat{\xi}_n - \xi_\star) \\
&= \hat{\mathbb{E}}_z[\nabla_\theta \bar{\ell}_{\gamma_\star}(z)] + \left(\mathbb{E}_z[\nabla_\theta^2 \bar{\ell}_{\gamma_\star}(z)] + (\hat{\mathbb{E}}_z - \mathbb{E}_z)[\nabla_\theta^2 \bar{\ell}_{\gamma_\star}(z)] + o_p(1)\right)(\hat{\theta}_n - \theta_\star) \\
&\quad + \left(\mathbb{E}_z[\nabla_{\theta\xi}^2 \bar{\ell}_{\gamma_\star}(z)] + (\hat{\mathbb{E}}_z - \mathbb{E}_z)[\nabla_{\theta\xi}^2 \bar{\ell}_{\gamma_\star}(z)] + o_p(1)\right)(\hat{\xi}_n - \xi_\star).
\end{aligned}
$$

By the SLLN, $(\hat{\mathbb{E}}_z - \mathbb{E}_z)[\nabla_\theta^2 \bar{\ell}_{\gamma_\star}(z)] \xrightarrow{\text{a.s.}} 0$ and $(\hat{\mathbb{E}}_z - \mathbb{E}_z)[\nabla_{\theta\xi}^2 \bar{\ell}_{\gamma_\star}(z)] \xrightarrow{\text{a.s.}} 0$, and thus are both $o_p(1)$. Furthermore, notice from Theorem 4.4 that since $\theta_\star$ maximizes $\theta \mapsto L(\theta, \xi)$ for any fixed $\xi$, it follows that $\nabla_{\theta\xi}^2 L(\theta_\star, \xi_\star) = \mathbb{E}_z[\nabla_{\theta\xi}^2 \bar{\ell}_{\gamma_\star}(z)] = 0$. Thus,

$$
\begin{aligned}
-\hat{\mathbb{E}}_z[\nabla_\theta \bar{\ell}_{\gamma_\star}(z)] &= \left(\mathbb{E}_z[\nabla_\theta^2 \bar{\ell}_{\gamma_\star}(z)] + o_p(1)\right)(\hat{\theta}_n - \theta_\star) + o_p(1)(\hat{\xi}_n - \xi_\star) \\
\iff -\sqrt{n}\hat{\mathbb{E}}_z[\nabla_\theta \bar{\ell}_{\gamma_\star}(z)] &= \sqrt{n}\left(\mathbb{E}_z[\nabla_\theta^2 \bar{\ell}_{\gamma_\star}(z)] + o_p(1)\right)(\hat{\theta}_n - \theta_\star) + \sqrt{n} \cdot o_p(1)(\hat{\xi}_n - \xi_\star).
\end{aligned}
$$

Now, from Theorem 4.4 and Assumption 4.9, we have that $\mathbb{E}_z[\nabla_\theta \bar{\ell}_{\gamma_\star}(z)] = 0$. Thus, by the CLT, $\sqrt{n}\hat{\mathbb{E}}_z[\nabla_\theta \bar{\ell}_{\gamma_\star}(z)] \rightsquigarrow \mathcal{N}(0, \text{Var}_z[\nabla_\theta \bar{\ell}_{\gamma_\star}(z)])$. Next, by the hypothesis (1) that $\sqrt{n}(\hat{\xi}_n - \xi_\star)$ is $O_p(1)$, the last term above is $o_p(1)$. Finally, by hypothesis (4), for sufficiently large $n$, $\mathbb{E}_z[\nabla_\theta^2 \bar{\ell}_{\gamma_\star}(z)] + o_p(1)$ is invertible. Thus, applying Slutsky's theorem yields $\sqrt{n}(\hat{\theta}_n - \theta_\star) \rightsquigarrow \mathcal{N}(0, V_\theta)$, where:

$$
V_\theta = \mathbb{E}_z\left[\nabla_\theta^2 \bar{\ell}_{\gamma_\star}(z)\right]^{-1} \text{Var}_z\left[\nabla_\theta \bar{\ell}_{\gamma_\star}(z)\right] \mathbb{E}_z\left[\nabla_\theta^2 \bar{\ell}_{\gamma_\star}(z)\right]^{-1}.
$$

Applying Proposition C.4 gives the desired result.

$\square$

## C.3  Learning the Proposal Distribution

We now restate and prove Theorem 4.12.

**Theorem 4.12.** *Consider the setting of Theorem 4.10, and suppose that $p_{\xi_\star}(y \mid x) = p(y \mid x)$. Then, the asymptotic variance matrix $V_\theta$ is:*

$$
V_\theta = \left(1 + \frac{1}{K}\right) \mathbb{E}_x[I(\theta_\star \mid x)]^{-1}.
$$

*Proof.* From (C.6) in the proof of Proposition C.4, we have the identity:

$$
\nabla_\theta^2 L(\theta_\star, \xi_\star) = -\mathbb{E}_{x,y}[\nabla_\theta \mathcal{E}_{\theta_\star}(x,y)\nabla_\theta \mathcal{E}_{\theta_\star}(x,y)^\mathsf{T}] + \mathbb{E}_x \mathbb{E}_{\bar{\mathbf{y}}|x;\xi_\star}\left[\left(\sum_{i=1}^{K+1} q_{\theta_\star,\xi_\star}(i \mid x, \bar{\mathbf{y}})\nabla_\theta \mathcal{E}_{\theta_\star}(x,y_i)\right)^{\otimes 2}\right]. \quad (C.7)
$$

With the assumption that $\theta_\star \in \Theta_\star$ and $p_{\xi_\star}(y \mid x) = p(y \mid x)$, we make two observations:

1. $\bar{\mathsf{P}}_{\bar{\mathbf{y}}|x;\xi_\star} = p(\cdot \mid x)^{\otimes K+1}$, and

2. $q_{\theta_\star,\xi_\star}(i \mid x, \bar{\mathbf{y}}) = \frac{1}{K+1}$ for all $i \in \{1, \ldots K+1\}$.

Hence, we have the simplifications:

$$
\mathbb{E}_x \mathbb{E}_{\bar{\mathbf{y}}|x;\xi_\star}\left[\left(\sum_{i=1}^{K+1} q_{\theta_\star,\xi_\star}(i \mid x, \bar{\mathbf{y}})\nabla_\theta \mathcal{E}_{\theta_\star}(x,y_i)\right)^{\otimes 2}\right] = \frac{1}{K+1}\mathbb{E}_x[I(\theta_\star \mid x)] + \mathbb{E}_{x,y}[\nabla_\theta \mathcal{E}_{\theta_\star}(x,y)]^{\otimes 2}.
$$

From this, we conclude that:

$$
\nabla_\theta^2 L(\theta_\star, \xi_\star) = \frac{1}{K+1}\mathbb{E}_x[I(\theta_\star \mid x)] - \mathbb{E}_x[I(\theta_\star \mid x)] = -\left(1 - \frac{1}{K+1}\right)\mathbb{E}_x[I(\theta_\star \mid x)].
$$

From Theorem 4.10, the asymptotic variance is $V_\theta = -[\nabla_\theta^2 L(\theta_\star, \xi_\star)]^{-1}$, from which the claim follows. $\square$

### C.3.1 Adversarial R-NCE

To prove Proposition 4.13, we require the following technical lemma.

**Lemma C.6** (Saddle Optimality). *Fix an integer $d \in \mathbb{N}_+$ and $k \in \{1, \ldots, d\}$. Define the function $F_k : \mathbb{R}^d_{\geqslant 0} \to \mathbb{R}$ as:*

$$F_k(\eta) := \log\left(\frac{\langle \eta, e_k \rangle}{\langle \eta, \mathbf{1} \rangle}\right) \langle \eta, e_k \rangle.$$

*The function $F_k$ is convex on the domain $\mathbb{R}^d_{\geqslant 0}$.*

*Proof.* A straightforward computation shows that:

$$\nabla^2 F_k(\eta) = \langle \eta, e_k \rangle \left(\frac{e_k}{\langle \eta, e_k \rangle} - \frac{\mathbf{1}}{\langle \eta, \mathbf{1} \rangle}\right)\left(\frac{e_k}{\langle \eta, e_k \rangle} - \frac{\mathbf{1}}{\langle \eta, \mathbf{1} \rangle}\right)^{\top}.$$

Since $\eta \in \mathbb{R}^d_{\geqslant 0}$, we have $\langle \eta, e_k \rangle \geqslant 0$ and hence $\nabla^2 F_k(\eta) \succcurlyeq 0$. $\qquad\square$

We now restate and prove Proposition 4.13.

**Proposition 4.13.** *Suppose that $\mathscr{F}$ is realizable, and furthermore, the family of proposal distributions $\mathscr{F}_n$ is realizable. That is, there exists some non-empty $\Xi_\star \subset \Xi$ s.t. for every $\xi_\star \in \Xi_\star$, $p_{\xi_\star}(y \mid x) = p(y \mid x)$ for a.e. $(x, y)$. Then:*

$$\max_{\theta \in \Theta} \min_{\xi \in \Xi} L(\theta, \xi) = L(\theta_\star, \xi_\star) = \log\left(\frac{1}{K+1}\right), \tag{4.8}$$

*where $(\theta_\star, \xi_\star) \in \Theta_\star \times \Xi_\star$.*

*Proof.* Fix $k \in \{1, \ldots, K+1\}$. From the proof of Theorem 4.4, we can write:

$$L(\theta_\star, \xi) = \mathbb{E}_x \mathbb{E}_{\bar{\mathbf{y}} \sim \mathsf{P}^{K+1}_{\bar{\mathbf{y}}|x;\xi}} \left[\log\left(\frac{p_{\theta_\star}(y_k|x)/p_\xi(y_k|x)}{\sum_{j=1}^{K+1} p_{\theta_\star}(y_j|x)/p_\xi(y_j|x)}\right) p_{\theta_\star}(y_k|x)/p_\xi(y_k|x)\right].$$

Define the mapping $\eta : (x, \bar{\mathbf{y}}) \to \mathbb{R}^{K+1}_{\geqslant 0}$ as $\eta_k := p_{\theta_\star}(y_k|x)/p_\xi(y_k|x)$. Notice that

$$\mathbb{E}_x \mathbb{E}_{\bar{\mathbf{y}} \sim \mathsf{P}^{K+1}_{\bar{\mathbf{y}}|x;\xi}} \eta_k(x, \mathbf{y}) = \mathbb{E}_x \mathbb{E}_{\bar{\mathbf{y}} \sim \bar{\mathsf{P}}_{\bar{\mathbf{y}}|x;\xi,k}}[1] = 1.$$

Thus, by Lemma C.6 and Jensen's inequality:

$$L(\theta_\star, \xi) = \mathbb{E}_x \mathbb{E}_{\bar{\mathbf{y}} \sim \mathsf{P}^{K+1}_{\bar{\mathbf{y}}|x;\xi}} F_k(\eta(x, \bar{\mathbf{y}})) \geqslant F_k(\mathbf{1}) = \log\left(\frac{1}{K+1}\right).$$

On the other hand, since $p_{\theta_\star}(y|x) = p_{\xi_\star}(y|x) = p(y|x)$, $L(\theta_\star, \xi_\star) = \log\left(\frac{1}{K+1}\right)$. So, we have the following chain of inequalities:

$$\log\left(\frac{1}{K+1}\right) = L(\theta_\star, \xi_\star) = \min_{\xi \in \Xi} L(\theta_\star, \xi) \leqslant \max_{\theta \in \Theta} \min_{\xi \in \Xi} L(\theta, \xi) \leqslant \max_{\theta \in \Theta} L(\theta, \xi_\star) = L(\theta_\star, \xi_\star) = \log\left(\frac{1}{K+1}\right).$$

The last inequality above is a consequence of Theorem 4.4. Thus, we conclude that $\Theta_\star \times \Xi_\star$ is a saddle-set of the game:

$$\max_{\theta \in \Theta} \min_{\xi \in \Xi} L(\theta, \xi).$$

$\qquad\square$

**Proposition 4.16.** *For any Stackelberg optimal pair $(\bar{\theta}, \bar{\xi})$ to the population game (4.8), it holds that $\bar{\theta} \in \Theta_\star$. Conversely, for any $\theta_\star \in \Theta_\star$, there exists an $\bar{\xi} \in \Xi$ such that $(\theta_\star, \bar{\xi})$ is Stackelberg optimal for the population game (4.8).*

*Proof.* We first prove the forward direction. By definition of Stackelberg optimality, it holds that:

$$\inf_{\xi' \in \Xi} L(\bar{\theta}, \xi') \geqslant \inf_{\xi' \in \Xi} L(\theta, \xi') \quad \forall \theta \in \Theta.$$

Suppose that $\bar{\theta} \notin \Theta_\star$. By Theorem 4.4, $L(\theta_\star, \xi) > L(\bar{\theta}, \xi)$ for all $\xi \in \Xi$. Taking the infimum on both sides, we have that:

$$\inf_{\xi \in \Xi} L(\theta_\star, \xi) > \inf_{\xi \in \Xi} L(\bar{\theta}, \xi),$$

where the strict inequality is preserved since $\Xi$ is compact and $\xi \mapsto L(\theta, \xi)$ is continuous for every $\theta \in \Theta$, so the infimums are attained. But this contradicts the Stackelberg optimality of $\bar{\theta}$.

Now for the backwards direction. Consider the chain of inequalities for any $\theta \in \Theta$:

$$\inf_{\xi' \in \Xi} L(\theta, \xi') \leqslant \sup_{\theta' \in \Theta} \inf_{\xi' \in \Xi} L(\theta', \xi') \leqslant \sup_{\theta' \in \Theta} L(\theta', \xi) = L(\theta_\star, \xi) \quad \forall \xi \in \Xi,$$

where the last inequality follows from optimality of $\theta_\star$ for any $\xi$, as per Theorem 4.4. Now, again by continuity of $\xi \mapsto L(\theta_\star, \xi)$ and compactness of $\Xi$, there exists an $\bar{\xi}$ such that $\inf_{\xi \in \Xi} L(\theta_\star, \xi) = L(\theta_\star, \bar{\xi})$. Hence, for any $(\theta, \xi) \in \Theta \times \Xi$:

$$\inf_{\xi' \in \Xi} L(\theta, \xi') \leqslant L(\theta_\star, \bar{\xi}) \leqslant L(\theta_\star, \xi),$$

which shows that $(\theta_\star, \bar{\xi})$ is Stackleberg optimal for (4.8). $\qquad\square$

**Theorem 4.17** (Adversarial Consistency)**.** *Suppose that Assumption 4.1 and Assumption 4.2 hold. Let $(\hat{\theta}_n, \hat{\xi}_n) \in \Theta \times \Xi$ be a sequence of Stackelberg optimal pairs, as defined in (4.10), for the finite sample game (4.9). Then, the following results hold:*

1. *Suppose that $\Theta_\star$ is contained in the interior of $\Theta$. We have that $d(\hat{\theta}_n, \Theta_\star) \stackrel{n \to \infty}{\longrightarrow} 0$ a.s.*

2. *Suppose furthermore that Assumption 4.9 holds. Let $\bar{\bar{\Xi}} := \{\xi \in \Xi \mid L(\theta_\star, \xi) = \inf_{\xi' \in \Xi} L(\theta_\star, \xi')\}$ for an arbitrary choice of $\theta_\star \in \Theta_\star$ (the set definition is independent of this choice), and suppose $\bar{\bar{\Xi}}$ is contained in the interior of $\Xi$. We have that $d(\hat{\xi}_n, \bar{\bar{\Xi}}) \stackrel{n \to \infty}{\longrightarrow} 0$ a.s.*

*Proof.* **Part (1).** We prove convergence of $\hat{\theta}_n$ to a maxmin Stackelberg optimal point for the population game, i.e., a global maximizer of the function $L^\theta$, defined as $\theta \mapsto \inf_{\xi \in \Xi} L(\theta, \xi)$. From Proposition 4.16, this is necessarily the set $\Theta_\star$. By our assumptions, we have that $L(\theta, \xi)$ is continuous. Since $\Xi$ is compact, by classic envelope theorems, we have that $L^\theta$ is continuous. By Lemma C.2, since $\Theta_\star$ is contained in the interior of $\Theta$, there exists $\varepsilon_0 > 0$ such that for all $0 < \varepsilon \leqslant \varepsilon_0$:

$$\theta \in \mathrm{cl}(\Theta \setminus (\Theta_\star + B_2(\varepsilon))) \Longrightarrow L^\theta < L^\star := \sup_{\theta \in \Theta} L^\theta.$$

Put $L_n^\theta$ as the function $\theta \mapsto \inf_{\xi \in \Xi} L_n(\theta, \xi)$. Since $(\hat{\theta}_n, \hat{\xi}_n)$ is a Stackleberg equilibrium, we have that $\sup_{\theta \in \Theta} L_n^\theta = L_n^{\hat{\theta}_n}$. Now, observe that if $\hat{\theta}_n \in \Theta \setminus (\Theta_\star + B_2(\varepsilon))$:

$$
\begin{aligned}
L^{\hat{\theta}_n} < L^\star &\Longleftrightarrow L^\star - L_n^{\hat{\theta}_n} + L_n^{\hat{\theta}_n} - L^{\hat{\theta}_n} > 0 \\
&\Longleftrightarrow \sup_{\theta \in \Theta} \inf_{\xi \in \Xi} L(\theta, \xi) - \sup_{\theta \in \Theta} \inf_{\xi \in \Xi} L_n(\theta, \xi) + \inf_{\xi \in \Xi} L_n(\hat{\theta}_n, \xi) - \inf_{\xi \in \Xi} L(\hat{\theta}_n, \xi) > 0.
\end{aligned}
$$

Note that for any two real-valued functions $f(z), g(z)$ and domain $\mathcal{Z}$, it holds that:

$$\left| \sup_{z \in \mathcal{Z}} f(z) - \sup_{z \in \mathcal{Z}} g(z) \right| \leqslant \sup_{z \in \mathcal{Z}} |f(z) - g(z)|.$$

Thus,

$$\left| \sup_{\theta \in \Theta} \inf_{\xi \in \Xi} L_n(\theta, \xi) - \sup_{\theta \in \Theta} \inf_{\xi \in \Xi} L(\theta, \xi) \right| \leqslant \sup_{\theta \in \Theta} \left| \inf_{\xi \in \Xi} L_n(\theta, \xi) - \inf_{\xi \in \Xi} L(\theta, \xi) \right|$$

$$= \sup_{\theta \in \Theta} \left| \sup_{\xi \in \Xi} [-L(\theta, \xi)] - \sup_{\xi \in \Xi} [-L_n(\theta, \xi)] \right|$$

$$\leqslant \sup_{\theta \in \Theta, \xi \in \Xi} |L_n(\theta, \xi) - L(\theta, \xi)|. \tag{C.8}$$

Similarly,

$$\left| \inf_{\xi \in \Xi} L_n(\hat{\theta}_n, \xi) - \inf_{\xi \in \Xi} L(\hat{\theta}_n, \xi) \right| = \left| \sup_{\xi \in \Xi} [-L(\hat{\theta}_n, \xi)] - \sup_{\xi \in \Xi} [-L_n(\hat{\theta}_n, \xi)] \right|$$

$$\leqslant \sup_{\xi \in \Xi} |L_n(\hat{\theta}_n, \xi) - L(\hat{\theta}_n, \xi)|$$

$$\leqslant \sup_{\theta \in \Theta, \xi \in \Xi} |L_n(\theta, \xi) - L(\theta, \xi)|.$$

This yields the implication:

$$d(\hat{\theta}_n, \Theta_\star) > \varepsilon \implies \sup_{\theta \in \Theta, \xi \in \Xi} |L_n(\theta, \xi) - L(\theta, \xi)| > 0.$$

Following the proof of Theorem 4.8, the conclusion of part (1) is now straightforward.

**Part (2).** Let $\theta_\star \in \Theta_\star$ be arbitrary. The function $\xi \mapsto L(\theta_\star, \xi)$ is continuous, so by Lemma C.2, since $\bar{\Xi}$ is assumed to be in the interior of $\Xi$, there exists $\varepsilon_0 > 0$ such that for $0 < \varepsilon \leqslant \varepsilon_0$:

$$\xi \in \mathrm{cl}(\Xi \setminus (\bar{\Xi} + B_2(\varepsilon))) \implies L(\theta_\star, \xi) > L^\star := \inf_{\xi \in \Xi} L(\theta_\star, \xi). \tag{C.9}$$

Let $\delta_\varepsilon > 0$ denote this gap:

$$\delta_\varepsilon := \inf_{\xi \in \mathrm{cl}(\Xi \setminus (\bar{\Xi} + B_2(\varepsilon)))} L(\theta_\star, \xi) - L^\star.$$

By Assumption 4.9, we have that $L$ is $\mathcal{C}^1$ on $\Theta \times \Xi$. Hence, the following constant $B_L$ is well defined and finite:

$$B_L := \sup_{\theta \in \Theta, \xi \in \Xi} \|\nabla_\theta L(\theta, \xi)\|.$$

Next, let $\hat{\theta}_\star \in \Theta_\star$ satisfy $\|\hat{\theta}_\star - \hat{\theta}_n\| \leqslant d(\hat{\theta}_n, \Theta_\star) + \frac{\delta_\varepsilon}{2B_L}$, and let $\bar{\xi} \in \bar{\Xi}$ be arbitrary. Consider the following chain of inequalities:

$$|L(\hat{\theta}_\star, \hat{\xi}_n) - L(\hat{\theta}_\star, \bar{\xi})| \leqslant |L(\hat{\theta}_\star, \hat{\xi}_n) - L(\hat{\theta}_n, \hat{\xi}_n)| + |L(\hat{\theta}_n, \hat{\xi}_n) - L_n(\hat{\theta}_n, \hat{\xi}_n)| + |L_n(\hat{\theta}_n, \hat{\xi}_n) - L(\hat{\theta}_\star, \bar{\xi})|$$

$$\leqslant \sup_{\xi \in \Xi} |L(\hat{\theta}_\star, \xi) - L(\hat{\theta}_n, \xi)| + \sup_{\theta \in \Theta, \xi \in \Xi} |L(\theta, \xi) - L_n(\theta, \xi)|$$

$$+ |L_n(\hat{\theta}_n, \hat{\xi}_n) - L(\hat{\theta}_\star, \bar{\xi})|$$

$$=: T_1 + T_2 + T_3. \tag{C.10}$$

We first control $T_1$ by uniform Lipschitz continuity; in particular, by boundedness of $\nabla_\theta L$ and the mean-value theorem:

$$T_1 = \sup_{\xi \in \Xi} |L(\hat{\theta}_\star, \xi) - L(\hat{\theta}_n, \xi)| \leqslant \sup_{\theta \in \Theta, \xi \in \Xi} \|\nabla_\theta L(\theta, \xi)\| \|\hat{\theta}_\star - \hat{\theta}_n\| = B_L \|\hat{\theta}_\star - \hat{\theta}_n\| \leqslant B_L d(\hat{\theta}_n, \Theta_\star) + \frac{\delta_\varepsilon}{2}.$$

Next, we control $T_3$. Since $(\hat{\theta}_n, \hat{\xi}_n)$ is Stackelberg optimal for the finite sample game, we have $L_n(\hat{\theta}_n, \hat{\xi}_n) = \sup_{\theta \in \Theta} \inf_{\xi \in \Xi} L_n(\theta, \xi)$. Furthermore, by Proposition 4.16, the pair $(\hat{\theta}_\star, \bar{\xi})$ is Stackelberg optimal for the population game, and hence $L(\hat{\theta}_\star, \bar{\xi}) = \sup_{\theta \in \Theta} \inf_{\xi \in \Xi} L(\theta, \xi)$. By (C.8),

$$T_3 = |L_n(\hat{\theta}_n, \hat{\xi}_n) - L(\hat{\theta}_\star, \bar{\xi})| = \left| \sup_{\theta \in \Theta} \inf_{\xi \in \Xi} L_n(\theta, \xi) - \sup_{\theta \in \Theta} \inf_{\xi \in \Xi} L(\theta, \xi) \right|$$

$$\leqslant \sup_{\theta \in \Theta, \xi \in \Xi} |L(\theta, \xi) - L_n(\theta, \xi)| = T_2.$$

Combining the bounds on $T_1$ and $T_3$ with (C.10):

$$|L(\hat{\theta}_\star, \hat{\xi}_n) - L(\hat{\theta}_\star, \bar{\xi})| \leqslant 2T_2 + B_L d(\hat{\theta}_n, \Theta_\star) + \frac{\delta_\varepsilon}{2}.$$

This bound with (C.9) yields the implication:

$$d(\hat{\xi}_n, \bar{\Xi}) > \varepsilon \implies \delta_\varepsilon \leqslant L(\hat{\theta}_\star, \hat{\xi}_n) - L^\star$$

$$\implies \delta_\varepsilon \leqslant 2T_2 + B_L d(\hat{\theta}_n, \Theta_\star) + \frac{\delta_\varepsilon}{2}$$

$$\implies \delta_\varepsilon \leqslant 4T_2 + 2B_L d(\hat{\theta}_n, \Theta_\star).$$

Hence, we have:

$$\left\{ \limsup_{n \to \infty} d(\hat{\xi}_n, \bar{\Xi}) > \varepsilon \right\} \subset \left\{ \limsup_{n \to \infty} \sup_{\theta \in \Theta, \xi \in \Xi} |L(\theta, \xi) - L_n(\theta, \xi)| > 0 \right\} \bigcup \left\{ \limsup_{n \to \infty} d(\hat{\theta}_n, \Theta_\star) > 0 \right\}.$$

Part (2) now follows, since both events on the RHS were established in part (1) as measure zero events, and since $\varepsilon > 0$ is arbitrarily small. $\qquad\square$

**Theorem 4.18** (Adversarial Normality). *Suppose that $\mathscr{F}$ is realizable and Assumptions 4.1, 4.2, and 4.9 hold. Let $\hat{\gamma}_n := (\hat{\theta}_n, \hat{\xi}_n) \in \Theta \times \Xi$ be a sequence of Stackelberg optimal pairs, as defined in (4.10), for the finite sample game (4.9), and define $\bar{\Xi}$ as in Theorem 4.17. Assume then that the following properties hold:*

1. *The sets $\Theta_\star, \bar{\Xi}$ are singletons, i.e., $\Theta_\star = \{\theta_\star\}$, and $\bar{\Xi} = \{\bar{\xi}\}$, and $\gamma_\star := (\theta_\star, \bar{\xi})$ lies in the interior of $\Theta \times \Xi$.*

2. *The Hessian of $\bar{\ell}_\gamma$ is Lipschitz-continuous, i.e.,*

$$\|\nabla_\gamma^2 \bar{\ell}_{\gamma_1}(z) - \nabla_\gamma^2 \bar{\ell}_{\gamma_2}(z)\|_{\mathrm{op}} \leqslant M(z)\|\gamma_1 - \gamma_2\|, \quad \mathbb{E}_z[M(z)] < \infty, \ \forall \gamma_1, \gamma_2 \in \Theta \times \Xi.$$

3. *The $(\theta, \theta)$ block of the Hessian $\nabla_\gamma^2 L(\theta_\star, \bar{\xi})$, denoted $\nabla_\theta^2 L(\theta_\star, \bar{\xi})$, satisfies $\nabla_\theta^2 L(\theta_\star, \bar{\xi}) \prec 0$, and the $(\xi, \xi)$ block, denoted $\nabla_\xi^2 L(\theta_\star, \bar{\xi})$, satisfies $\nabla_\xi^2 L(\theta_\star, \bar{\xi}) \succ 0$.*

*Then, $\sqrt{n}(\hat{\gamma}_n - \gamma_\star) \rightsquigarrow \mathcal{N}(0, V_\gamma^a)$, where*

$$V_\gamma^a = \mathbb{E}_z \left[ \nabla_\gamma^2 \bar{\ell}_{\gamma_\star}(z) \right]^{-1} \mathrm{Var}_z \left[ \nabla_\gamma \bar{\ell}_{\gamma_\star}(z) \right] \mathbb{E}_z \left[ \nabla_\gamma^2 \bar{\ell}_{\gamma_\star}(z) \right]^{-1}. \tag{4.11}$$

*Proof.* Note first that as a consequence of Theorem 4.17, $\hat{\gamma}_n \overset{n \to \infty}{\longrightarrow} \gamma_\star$ almost surely. We now proceed in a manner similar to the proof for Theorem 4.10, with the following identity:

$$\nabla_\gamma \bar{\ell}_{\hat{\gamma}_n}(z) = \nabla_\gamma \bar{\ell}_{\gamma_\star}(z) + \nabla_\gamma^2 \bar{\ell}_{\gamma_\star}(z)(\hat{\gamma}_n - \gamma_\star) + R(\hat{\gamma}_n - \gamma_\star),$$

where $R$ is a remainder matrix dependent upon $z, \gamma_\star, \hat{\gamma}_n$, and satisfies: $\|R\|_{\mathrm{op}} \leqslant \frac{1}{2} M(z)\|\hat{\gamma}_n - \gamma_\star\|$. Applying the empirical expectation on all terms, and as in the proof for Theorem 4.10, upper-bounding $\|\hat{\mathbb{E}}_z R(\gamma_\star, \hat{\gamma}_n, z)\|_{\mathrm{op}}$ by an $o_p(1)$ matrix by leveraging the fact that $\|\hat{\gamma}_n - \gamma_\star\| \overset{\text{a.s.}}{\to} 0$, yields:

$$\hat{\mathbb{E}}_z[\nabla_\gamma \bar{\ell}_{\hat{\gamma}_n}(z)] = \hat{\mathbb{E}}_z[\nabla_\gamma \bar{\ell}_{\gamma_\star}(z)] + \left( \hat{\mathbb{E}}_z[\nabla_\gamma^2 \bar{\ell}_{\gamma_\star}(z)] + o_p(1) \right)(\hat{\gamma}_n - \gamma_\star)$$

$$= \hat{\mathbb{E}}_z[\nabla_\gamma \bar{\ell}_{\gamma_\star}(z)] + \left( \mathbb{E}_z[\nabla_\gamma^2 \bar{\ell}_{\gamma_\star}(z)] + (\hat{\mathbb{E}}_z - \mathbb{E}_z)[\nabla_\gamma^2 \bar{\ell}_{\gamma_\star}(z)] + o_p(1) \right)(\hat{\gamma}_n - \gamma_\star).$$

By SLLN, $(\hat{\mathbb{E}}_z - \mathbb{E}_z)[\nabla_\gamma^2 \bar{\ell}_{\gamma_\star}(z)] \overset{\text{a.s.}}{\to} 0$, and thus:

$$\hat{\mathbb{E}}_z[\nabla_\gamma \bar{\ell}_{\hat{\gamma}_n}(z)] = \hat{\mathbb{E}}_z[\nabla_\gamma \bar{\ell}_{\gamma_\star}(z)] + \left(\mathbb{E}_z[\nabla_\gamma^2 \bar{\ell}_{\gamma_\star}(z)] + o_p(1)\right)(\hat{\gamma}_n - \gamma_\star).$$

To complete the result, we need only establish the following: (i) $\hat{\mathbb{E}}_z[\nabla_\gamma \bar{\ell}_{\hat{\gamma}_n}(z)] = 0$, (ii) $\mathbb{E}_z[\nabla_\gamma \bar{\ell}_{\gamma_\star}(z)] = 0$, and (iii) $\mathbb{E}_z[\nabla_\gamma^2 \bar{\ell}_{\gamma_\star}(z)]$ is invertible. The result then follows by applying CLT to $\sqrt{n}\hat{\mathbb{E}}_z[\nabla_\gamma \bar{\ell}_{\gamma_\star}(z)]$ and Slutsky's theorem (see for instance, the proof for Theorem 4.10). We show (ii) and (iii) first.

For (ii), notice that by the definition of Stackelberg optimality, $\bar{\xi}$ minimizes $\xi \mapsto L(\theta_\star, \xi)$ over $\Xi$ and by assumption, is contained within the interior of $\Xi$. Thus, $\nabla_\xi L(\theta_\star, \bar{\xi}) = 0$. Further, by Theorem 4.4, $\theta_\star$ maximizes $\theta \mapsto L(\theta, \xi)$ over $\Theta$ for any $\xi \in \Xi$, and is also assumed to be within the interior of $\Theta$. Thus, $\nabla_\theta L(\theta_\star, \bar{\xi}) = 0$. Leveraging $\mathcal{C}^1$-regularity, we conclude that $\mathbb{E}_z[\nabla_\gamma \bar{\ell}_{\gamma_\star}(x)] = 0$.

To prove (iii), notice that the Hessian $\nabla_\gamma^2 L(\theta_\star, \bar{\xi})$ is block-diagonal, since by optimality of $\theta_\star$ *for any* $\xi$, it follows that $\nabla_{\theta\xi}^2 L(\theta_\star, \bar{\xi}) = 0$. Further, the blocks $\nabla_\theta^2 L(\theta_\star, \bar{\xi})$ and $\nabla_\xi^2 L(\theta_\star, \bar{\xi})$ are assumed invertible.

Finally, to prove stationarity for the finite-sample objective at $\hat{\gamma}_n$, we first require the following additional result.

**Claim C.7.** $\|\nabla_\gamma^2 L_n(\hat{\theta}_n, \hat{\xi}_n) - \nabla_\gamma^2 L(\theta_\star, \bar{\xi})\|_{\text{op}} \overset{n\to\infty}{\longrightarrow} 0$ a.s.

*Proof.* Let $d := \dim(\theta) + \dim(\xi)$, and let $\mathbb{S}^{d-1}$ denote the unit-sphere in $\mathbb{R}^d$. We first claim that:

$$\sup_{\theta\in\Theta, \xi\in\Xi} \|\nabla_\gamma^2 L_n(\theta, \xi) - \nabla_\gamma^2 L(\theta, \xi)\|_{\text{op}} \overset{n\to\infty}{\longrightarrow} 0 \quad \text{a.s.} \tag{C.11}$$

To see this, by the variational representation of the operator norm (Rudin, 2008, Theorem 4.4), the empirical process above is:

$$\sup_{\substack{\theta\in\Theta, \xi\in\Xi, \\ v_1, v_2\in\mathbb{S}^{d-1}}} v_1^\mathsf{T}(\nabla_\gamma^2 L_n(\theta, \xi) - \nabla_\gamma^2 L(\theta, \xi))v_2.$$

Assumption 4.9 combined with Ferguson (2017, Theorem 16(a)) yields (C.11).

Next, by $\mathcal{C}^2$-regularity, $\nabla_\gamma^2 L(\theta, \xi) = \mathbb{E}_z \nabla_\gamma^2 \bar{\ell}_\gamma(z)$, and therefore by Jensen's inequality and the Hessian Lipschitz assumption:

$$\begin{aligned}
\|\nabla_\gamma^2 L(\hat{\theta}_n, \hat{\xi}_n) - \nabla_\gamma^2 L(\theta_\star, \bar{\xi})\|_{\text{op}} &= \|\mathbb{E}_z \nabla_\gamma^2 \bar{\ell}_{\hat{\gamma}_n}(z) - \mathbb{E}_z \nabla_\gamma^2 \bar{\ell}_{\gamma_\star}(z)\|_{\text{op}} \\
&\leqslant \mathbb{E}_z \|\nabla_\gamma^2 \bar{\ell}_{\hat{\gamma}_n}(z) - \nabla_\gamma^2 \bar{\ell}_{\gamma_\star}(z)\|_{\text{op}} \\
&\leqslant \mathbb{E}_z M(z)\|\hat{\gamma}_n - \gamma_\star\|.
\end{aligned}$$

Finally, then

$$\begin{aligned}
\|\nabla_\gamma^2 L_n(\hat{\theta}_n, \hat{\xi}_n) - \nabla_\gamma^2 L(\theta_\star, \bar{\xi})\|_{\text{op}} &\leqslant \|\nabla_\gamma^2 L_n(\hat{\theta}_n, \hat{\xi}_n) - \nabla_\gamma^2 L(\hat{\theta}_n, \hat{\xi}_n)\|_{\text{op}} + \|\nabla_\gamma^2 L(\hat{\theta}_n, \hat{\xi}_n) - \nabla_\gamma^2 L(\theta_\star, \bar{\xi})\|_{\text{op}} \\
&\leqslant \sup_{\theta\in\Theta, \xi\in\Xi} \|\nabla_\gamma^2 L_n(\theta, \xi) - \nabla_\gamma^2 L(\theta, \xi)\|_{\text{op}} + \mathbb{E}_z M(z)\|\hat{\gamma}_n - \gamma_\star\|.
\end{aligned}$$

Then, (C.11) along with $\hat{\gamma}_n \overset{n\to\infty}{\longrightarrow} \gamma_\star$ a.s. proves the stated claim. $\qquad\square$

Returning now to (i), since we only assume that $(\hat{\theta}_n, \hat{\xi}_n)$ is a Stackelberg optimal point, it may not satisfy the stationarity condition $\hat{\mathbb{E}}_z[\nabla_\gamma \bar{\ell}_{\hat{\gamma}_n}(z)] = 0$ (Jin et al., 2020). However, since $\hat{\xi}_n$ globally minimizes $\xi \mapsto L_n(\hat{\theta}_n, \xi)$ and $\hat{\xi}_n \to \bar{\xi}$ a.s., which is at an interior point, for large enough $n$ we have $\nabla_\xi L_n(\hat{\theta}_n, \hat{\xi}_n) = 0$.

Furthermore, by Claim C.7 and the assumption that $\nabla_\xi^2 L(\theta_\star, \bar{\xi})$ is non-singular, for large enough $n$ we have that $\nabla_\xi^2 L_n(\hat{\theta}_n, \hat{\xi}_n)$ is also non-singular. By the implicit function theorem, there exists a continuously differentiable function $\hat{G}_n : \theta \in \Theta \to \Xi$, defined for $\theta$ in an open local neighborhood of $\hat{\theta}_n$, s.t. $\hat{G}_n(\hat{\theta}_n) = \hat{\xi}_n$ and $\nabla_\xi L_n(\theta, \hat{G}_n(\theta)) = 0$. Then, since $\hat{\theta}_n$ is a global interior maximum (for large enough $n$, since $\hat{\theta}_n \to \theta_\star$ a.s.) of $\theta \mapsto \inf_{\xi\in\Xi} L_n(\theta, \xi)$, which in turn equals $L_n(\theta, \hat{G}_n(\theta))$ for $\theta$ near $\hat{\theta}_n$, stationarity dictates that $\nabla_\theta L_n(\hat{\theta}_n, \hat{G}_n(\hat{\theta}_n)) = 0$. It follows then by the chain rule that $\nabla_\theta L_n(\hat{\theta}_n, \hat{\xi}_n) = 0$. $\qquad\square$

**Corollary 4.19** (Marginal Normality). *Under the assumptions of Theorem 4.18, the parameters $\{\theta_n\}$ satisfy $\sqrt{n}(\hat{\theta}_n - \theta_\star) \rightsquigarrow \mathcal{N}(0, V_\theta^a)$, where*

$$V_\theta^a = -\nabla_\theta^2 L(\theta_\star, \bar{\xi})^{-1}$$

*Proof.* From the proof of Theorem 4.18, the Hessian $\nabla_\gamma^2 L(\theta_\star, \bar{\xi})$ is block-diagonal since the cross-term $\nabla_{\theta\xi}^2 L(\theta_\star, \bar{\xi}) = 0$. Thus, the $(\theta, \theta)$ block for $V_\gamma^a$ in (4.11) reduces to:

$$\mathbb{E}_z[\nabla_\theta^2 \bar{\ell}_{\gamma_\star}(z)]^{-1} \mathrm{Var}_z[\nabla_\theta \bar{\ell}_{\gamma_\star}(z)] \mathbb{E}_z[\nabla_\theta^2 \bar{\ell}_{\gamma_\star}(z)]^{-1}.$$

Applying Proposition C.4 gives the desired result. $\qquad\square$

