# OpenReview forum: "Revisiting Energy Based Models as Policies: Ranking Noise Contrastive Estimation and Interpolating Energy Models"
_TMLR — Accepted by TMLR_

### Review · Reviewer_A1FS · 2024-05-08

**Summary Of Contributions:**

- The authors provide a detailed analysis and theoretical explanation as to why Energy-Based Models (EBMs) trained  as conditional policies p(a|s) with the InfoNCE loss from Implicit Behavioral Cloning (IBC) do not converge to optimal solutions. This is an issue observed in many prior work and limited the general usage of EBMs in the context of robot learning.
- The authors propose a novel training algorithm for EBMs called R-NCE, that uses ranked noise contrastive loss together with a negative sampling model. The loss for training conditional EBM policies improves several flaws over prior methods and achieves better convergence and performance. This approach addresses the shortcomings of traditional EBM training techniques by improving model training robustness and effectiveness.
- A new class of interpolant-EBM models is proposed, combining elements from diffusion models with EBMs under the R-NCE training framework, that show improved performance in challenging experiments.
- The contributions and claims made in the paper are thoroughly validated through a series of toy task experiments and extensive evaluations on robotic learning benchmarks with competitive baselines.

**Audience:**

Yes

**Broader Impact Concerns:**

The work tackles improving Generative models in the context of robotics, this could be used for potential harmfull task such as:
- improved generative models such as an EBM have the potential to be used to generate images and videos, which can be used with negative intent to create harmful content
- Training robot policies, that act autonomously in the real world could potentially harm humans without safety measurements and controlled testing.

**Claims And Evidence:**

Yes

**Requested Changes:**

- add a general overview at the beginning of the experiment section to summarize the following experiments
- ablation experiment in one of the robotic task to ablate the negative sampling method and it’s impact to the performance and other design decision
 - A table of content for the long appendix would also improve readability
 - A reference training curve and stability would be very insightful for the readers. Chi et al., 2023 contains a figure to showcase the evaluation results over training epochs for IBC and Diffusion Policy, which compares diffusion policies and ebm rollout performance over training time. Adding a similar plot for R-NCE would further strengthen the claims regarding training stability and provide new insights.
 - The papers claims, that learning multimodal policies is crucial for imitation learning, but no experiment quantifies the diversity of learned behavior of the proposed model. Some experiment to analyze that property compared to prior policies would be interesting. Recent Imitation Learning benchmarks like [1], that contain a entropy measurement to quantify multimodal behavior, would be suited for that.
  - Prior work such as IBC reported difficulties in using Transfomer architectures as an EBM architecture given the training instabilities of EBMs trained with InfoNCE. Thus, all prior work only use MLP models as an EBM backbone, which was another major drawback of the default IBC framework. In my opinion, this fact that R-NCE EBMs can be trained with a transformer backbone effectively is worth highlighting somewhere.

All requested changes are not cruical for my acceptance but would further strengthen the contributions of this work.

[1]: Jia, Xiaogang, et al. "Towards Diverse Behaviors: A Benchmark for Imitation Learning with Human Demonstrations." _The Twelfth International Conference on Learning Representations_. 2023.

**Strengths And Weaknesses:**

Strengths
- The paper is very well written and structured. The papers start with a general explanation of why EBMs training is so challenging and highlighted shortcomings and theoretical flaws of prior training methods before introducing their own contributions.
- The proposed algorithm and novel model class claims are supported with detailed derivations and illustrative experiments to provide strong theoretical and empirical evidence
- Detailed experimental evaluation in different benchmarks with relevant baselines that verify the theoretical contributions with additional empirical evaluations. The propsed EBM policy does improve upon prior EBM policies by a significant margin in all experiments

Weaknesses
 - given the amount of experiments and evaluations in the experiment section, I found it hard to follow the conclusion of each section. Adding some more structure and and overview for the reader would be valuable.

---

> ### Author Response · Authors · 2024-07-03
> **Response**
>
> Thank you for your considered feedback. Please find below a per-item response to 'Requested Changes'.
>
> * The beginning of the experiment section (Section 7.1) features a fairly extensive overview of the baselines and models to be tested and principles for comparison—including a discussion on expressivity of the different models. In addition, the paragraph titled “Overview of Results” at the end of Section 7.1 provides a short summary of each of the experiments to follow. We are happy to add further details to this overview for any specific portions of Section 7.1 that the reviewer believes could be improved.
> * We are working to add an additional experimental result for the path-planning task, where we ablate the number of negative samples $K$ for R-NCE. This is the most crucial hyperparameter for the algorithm. Other hyperparameters, such as optimizer learning rates, architecture depth/width, are fairly standard.
> * We have added a ToC for the appendix.
> * We will add the evaluation training curve for R-NCE/I-R-NCE for the Push-T task, in order to provide a direct comparison to the corresponding plot in Chi et al. Indeed, we observe stable improvement in the policy’s performance, similar to a Diffusion policy, and unlike the IBC curves in Chi et al.
> * Please refer to the figures in Table 3 and Table 5 for an illustration on the diversity of the learned policy in both the path-planning task, and the Push-T task. Regarding the proposed entropy measurement in [1], we agree in principle that this is a reasonable idea. However, the entropy measurement in [1] requires task-specific discretization of the action space, since the action spaces we are dealing with are inherently continuous (and differential entropy as a stand-alone quantity is not meaningful). However, both path-planning and Push-T use action spaces corresponding to absolute $(x, y)$ coordinates, naively discretizing this space will not result in a useful metric. We are open to any suggestions on what discretization would yield a more meaningful measurement.
> * We have highlighted the ability to train Transformer-based EBM architectures in the contributions statement in the Introduction.
>
> All edits in the manuscript are in red.

---

### Review · Reviewer_i6Db · 2024-06-15

**Summary Of Contributions:**

The paper presents several theoretical results regarding energy-based models (EBM) in the behavior cloning setting:
	- The authors demonstrate that a previous method (IBC) based on InfoNCE may be not consistent, while their proposed method based on R-NCE with a learned negative sampler is.
	- Inspired by diffusion models, the authors propose a method to train EBM models with multiple noise resolutions.
The propositions are experimentally validated on three domains.

**Audience:**

Yes

**Claims And Evidence:**

Yes

**Requested Changes:**

1. For better legibility, I would suggest the authors to explain right away why it's useful to define the R-NCE objective as in (3.8), which looks a bit strange since the LHS doesn't depend on k, but the RHS does.
2. In Remark 3.2, could the authors explain why Lines 11 and 12 can be performed independently? It seems that Line 12 depends on Line 11.
3. For the path planning domain, although the authors seem to follow the previous work by Urain et al. I was wondering why c_obs (which has a square root) is not in the same scale as the other terms, which are all squared distances. Could the authors comment on that?
4. I like the proposition of interpolating EBMs. However, could the authors discuss more clearly about the possible advantages/disadvantages of this model compared to alternative models like diffusion models or stochastic interpolants?
5. Could you comment about the sensitivitiy of the results to the hyperparameters?

Some minor issues:
Page 3: that easier
Page 4: to the their
Page 5: I find the notation to define \bar y not so great, since it mixes set and vector notations.
Page 10: suceed
Page 10: Please recall the definition of || . ||_op
Page 12: Is V_R defined?
Page 15: Stackleberg

**Strengths And Weaknesses:**

STRENGTHS

The paper presents several theoretical contributions, which allow notably to better understand previous work (IBC). Even if some of them (e.g., Theorem 4.4) are extension of existing ones, their presentation and discussion in the context of behavior cloning seem to be quite novel. Moreover, the proposition about interpolating EBM sounds promising and could lead to interesting novel developments.

Although the paper could be more self-contained and is not always easily accessible to non-experts, I find the exposition to be well-organized and well-written, with informative side remarks and connections to other results and works.


WEAKNESSES

The empirical validation seems to be quite limited. For instance, it is not clear to me how sensitive the experimental results are to the various chosen hyperparameters.

The paper could be written to be more accessible to non-experts.

---

> ### Author Response · Authors · 2024-07-03
> **Response**
>
> Thank you for your considered feedback. Please find below a per-item response to 'Requested Changes'.
>
> * We have added clarification to why we adopt the $q$ notation. Notably, it is to highlight the similarity to a multi-class classification objective (cf. the “Intuition” paragraph in Section 3.1 leading up to Equation (3.8)), and help ease the discussion within the analysis sections. For instance, consider the gradient for the R-NCE objective below Equation (3.9), which would be more notationally burdensome without the help of the multi-class probability notation.
>
> * We have clarified Remark 3.2. We wish to highlight that as the negative sampling distribution does not depend upon the EBM’s parameters, no reparameterization tricks are required and the batch augmentation in Line 11 does not introduce any gradient dependency during the optimization step in Line 12.
>
> * Regarding $c_{\mathrm{obs}}$, we replicated the cost function from Urain et al.'s codebase (https://github.com/anindex/stoch_gpmp/blob/8fe9c3b6be6ed028b1432631ccc373bbe8080ec0/stoch_gpmp/costs/cost_functions.py#L258C26-L258C31). While we agree that the scale of $c_{\mathrm{obs}}$ is not intuitive, we decided to keep it in the interest of being consistent with the StochGPMP implementation online. Finally, we remark that our algorithms and results are agnostic to whatever cost function is used to score the trajectories.
>
> * Regarding Interpolating EBMs, we have added Section 5.2.3 discussing its advantages. Specifically, given the fact that both sampling and scoring samples is critical to achieve the best closed-loop performance (cf. Table 4), the key advantage revolves around the ease with which I-EBMs can do both. In addition to the modifications in Section 5.2.3, we also provide some commentary on the baseline Diffusion-$\phi$, which leverages the gradient of a scalar potential to construct the diffusion denoiser: Much like I-EBMs, Diffusion-$\phi$, which uses the gradient of an EBM as a score model, does not require expensive log-probability ODE calculations to score samplings. However, despite comparable computation time, there are still a few disadvantages to relying on Diffusion-$\phi$. First, is that the correct way to train Diffusion-$\phi$ models remains an open research question [Salimans and Ho, 2021]. Indeed in [Salimans and Ho, 2021], they actually found that for image classification tasks, the standard gradient of scalar function parameterization that we use in Diffusion-$\phi$ is inferior to a modified energy model function proposed in that work. However, in our experiments, we found that the standard parameterization worked fairly well. Hence, at the very least this implies that Diffusion-$\phi$ requires an additional architectural hyperparameter to tune over for each task. Second, we found some numerical instability when querying our Diffusion-$\phi$ energy model, as described in Footnote 5 at the bottom of Page 23. Indeed, since when $\sigma \to 0$ the energy model becomes degenerate, we had to introduce yet another hyperparameter $\sigma_{\mathrm{rel}}$ indicating the noise scale at which we query the likelihood. In our experiments we found that the most accurate $\sigma_{\mathrm{rel}}$ was an order of magnitude more than the $\sigma$ at which we stop the sampler in the backwards process. This issue, in addition to adding another hyperparameter, is mathematically unsatisfying and needs further investigation (which is out of scope for this work). Furthermore, and most importantly, it is completely avoided in I-EBMs since I-EBMs build off of stochastic interpolations which define a valid probability bridge on the finite interval $[0, 1]$. Finally, we note that at a conceptual level, diffusion models and I-EBMs are not fundamentally incompatible. Indeed, we could replace the interpolating backend of I-EBM with a diffusion process, which we also leave to future exploration.
>
> * We are working to add an additional experimental result for the path-planning example where we ablate the number of negative samples $K$ for R-NCE. This is the most crucial hyperparameter for the algorithm.
>
> * Thanks for pointing out the missing notation definitions and various typos, we have now addressed these issues.
>
> All edits in the manuscript are in red.

---

### Review · Reviewer_jagM · 2024-06-21

**Summary Of Contributions:**

The authors propose the use of R-NCE as a training algorithm for Energy-Based Policies. The paper claims the limitations of the training algorithm of Implicit Behavioral Cloning (IBC) and show that those limitations can be solved using R-NCE.

Additionally, the authors introduce I-R-NCE. Inspired by the multiple time steps based generative models, this work extends R-NCE to train Energy-Based Models on multiple timesteps.

The paper presents a large theoretical evaluation comparing the training loss of IBC and R-NCE to validate their claims and show empirically the application of R-NCE Energy-Based Policies in a set of Behavioral Cloning experiments.

**Audience:**

Yes

**Broader Impact Concerns:**

There are no ethical implications in the work.

**Claims And Evidence:**

Yes

**Requested Changes:**

Regarding the contributions of the paper, the authors should check the highlighted weaknesses and fix them accordingly.

Regarding the experimental section, I suggest the authors to evaluate their proposed algorithm in a challenging robotics experiments such as Robomimic (https://arxiv.org/abs/2108.03298).

**Strengths And Weaknesses:**

**Strengths**
- The presented theoretical comparison between IBC and R-NCE loss is of high interest to understand the limitations of IBC and presents the clear benefit of using R-NCE instead.
- The authors show the usefulness of Energy-Based Models as Policies if the selected training loss is proper.

**Weaknesses**
- Some of the authors claim require additional details to be clear.
  - In page 1, 'EBM yields compact representations; model size is an important...'. Nevertheless, the authors are not only training EBM, but also a contrastive model, such as a CNF that would still be large.
  - In page 2, authors claim as contribution 'EBMs are compatible with multiple noise resolutions'. Nevertheless, in the paper 'How to Train Your Energy-Based Models' (https://arxiv.org/abs/2101.03288) it is already suggested this possibility, that authors fail to cite.
- Despite the authors explore the application of EBM's as policies for robotics, there are no experiments in a high dimensional robot.

---

> ### Author Response · Authors · 2024-07-03
> **Response**
>
> Thank you for your considered feedback. Please find below a per-item response to 'Weaknesses'.
>
> * We highlight that the first bullet point under ‘Basic Principles for Model Comparison’ in Section 7.1 states that the parameter counts between different models were held in similar ranges. Furthermore, for R-NCE and I-R-NCE, it is the total number of parameters between the EBM and CNF that is used for comparison. We have also added a remark in the Push-T example where the encoders for the observation histories (image + robot state) are shared between the EBM and CNF, Thus, only the two prediction heads differ in our architecture. Furthermore, only the R-NCE or the CNF training objective is allowed to update the encoder, enforced using a stop-gradient operator.
>
> * Regarding Song and Kingma’s work, we have indeed already cited the aforementioned paper. Nevertheless, we agree that the idea that EBMs are in principle compatible with multiple noise resolutions is not necessarily novel; in fact, this is how diffusion models actually came to be. However, what is not so clear is what is the best way to train EBMs at multiple noise resolutions. While diffusion models provide one approach, our work provides an alternative competitive approach based on quite different mathematical principles (ranking NCE vs. denoising score matching). Therefore, we have reworded the emphasis of our contribution to be around a novel Interpolating-EBMs architecture to train EBMs at multiple noise resolutions.
>
> * To complement our theoretical contributions, we have chosen to report a thorough empirical analysis on benchmarks recently studied in the Diffusion policies literature (cf. Chi et al.): path-planning in obstacle fields & contact-rich pushing tasks. We believe these tasks are highly representative of the type of multi-modality present in real robot control tasks. As the focus of our work is mostly theory and algorithms, we believe that additional hardware experiments are out-of-scope for the current publication.
>
> All edits in the manuscript are in red.

---

> ### Comment · Reviewer_jagM · 2024-07-22
> **No further comments**
>
> The authors made clear the claims, I have no further objections.
>
> I therefore agree the paper is in a publishable state.

---

### Decision · Action_Editor_pie6 · 2024-07-29

**Recommendation:** Accept as is

**Comment:**

The submission investigates the use of energy based models for behavioral cloning. Based on the observation that training EBMs using the InfoNCE loss is biased (unless when using uniform negative samples, which are ineffective in practice), an alternate, unbiased training objective based on ranking noise contrastive estimation (R-NCE) with a learned negative sampling distribution is proposed. Furthermore, inspired by the success of diffusion models, the paper proposed to train an interpolating EBM (I-EBM) at different noise levels. For that variant, the EBM gets the noise level as additional input, and a continuous normalizing flow is used for negative sampling with the corresponding noise level. The proposed method is evaluated on low-dimensional toy tasks and on higher-dimensional motion planning and block pushing tasks that were used in related work. In these experiments, training EBMs with R-NCE was shown to clearly outperform the INFO-NCE loss, reaching the performance--or even slightly outperforming--diffusion models and normalizing flows.

As the reviews did not raise major concerns that would need to be addressed, I recommend acceptance as is. Yet, I encourage the authors to add the additional ablations in the block pushing environment, which they mentioned in their rebuttal, to the camera-ready.

**Audience:**

The main finding that EBMs can achieve remarkable performance for behavioral cloning seems quite interesting for the community. Additional contributions of interest include the R-NCE training at different noise levels and the theoretical analysis.

**Claims And Evidence:**

All claims seem to be supported by sufficient evidence. The reviewers did also not raise major concerns in that regard. Some statements have been slightly rephrased in the revision to address minor concerns.